# MUBen: Benchmarking the Uncertainty of Molecular Representation Models

**Yinghao Li**[1], **Lingkai Kong**[1], **Yuanqi Du**[2], **Yue Yu**[1], **Yuchen Zhuang**[1], **Wenhao Mu**[1],
**Chao Zhang**[1]

[1]Georgia Institute of Technology, Atlanta, GA    [2]Cornell University, Ithaca, NY
[1]{yinghaoli,lkkong,yueyu,yczhuang,wmu30,chaozhang}@gatech.edu
[2]yd392@cornell.edu

## Abstract

Large molecular representation models pre-trained on massive unlabeled data have shown great success in predicting molecular properties. However, these models may tend to overfit the fine-tuning data, resulting in over-confident predictions on test data that fall outside of the training distribution. To address this issue, uncertainty quantification (UQ) methods can be used to improve the models' calibration of predictions. Although many UQ approaches exist, not all of them lead to improved performance. While some studies have included UQ to improve molecular pre-trained models, the process of selecting suitable backbone and UQ methods for reliable molecular uncertainty estimation remains underexplored. To address this gap, we present MUBEN, which evaluates different UQ methods for state-of-the-art backbone molecular representation models to investigate their capabilities. By fine-tuning various backbones using different molecular descriptors as inputs with UQ methods from different categories, we critically assess the influence of architectural decisions and training strategies. Our study offers insights for selecting UQ for backbone models, which can facilitate research on uncertainty-critical applications in fields such as materials science and drug discovery.

## 1 Introduction

The task of molecular representation learning is pivotal in scientific domains and bears the potential to facilitate research in fields such as chemistry, biology, and materials science [81]. Nonetheless, supervised training typically requires vast quantities of data, which may be challenging to acquire due to the need for expensive laboratory experiments [15]. With the advent of self-supervised learned Transformers [80] pioneered by BERT [13] and GPT [64], there has been a surge of interest in creating large-scale pre-trained molecular representation models from unlabeled datasets [82, 93, 65, 48, 96, 75]. Such models have demonstrated impressive representational capabilities, achieving state-of-the-art (SOTA) performance on a variety of molecular property prediction tasks [90].

However, in many applications, there is a pressing need for reliable predictions that are not only precise but also *uncertainty-aware*. The provision of calibrated uncertainty estimates allows us to distinguish "noisy" predictions and thus improve model robustness [23], or estimate data distributions, which can enhance downstream tasks such as active learning [20], high throughput screening [52, 60], or wet-lab experimental design [15, 71]. Unfortunately, large-scale models can easily overfit the fine-tuning data and exhibit misplaced confidence in their predictions [28, 42]. Several works have introduced various uncertainty quantification (UQ) methods in molecular property prediction [67, 95, 68, 34, 52, 31, 8, 71, 12, 27, 89] and in neighboring research areas such as protein engineering [26]. For instance, [8] extend a message passing neural network (MPNN) [24] with post-hoc recalibration [28] to address the overconfidence in molecular property prediction; [71] apply evidential

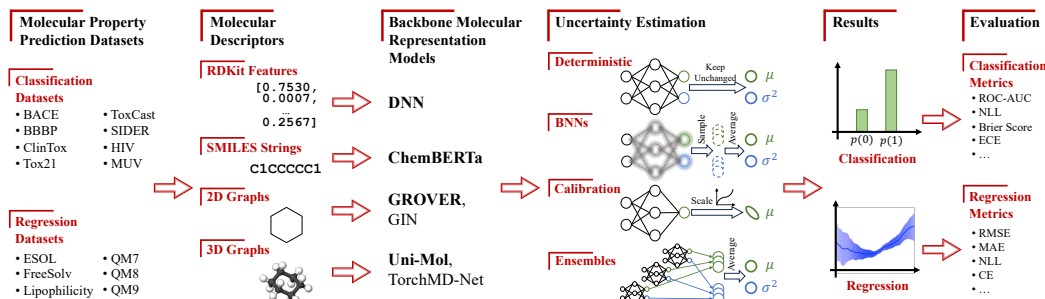

Figure 1: An overview of MUBEN with all datasets, backbone models, UQ methods, and metrics enumerated. Elements in green and blue are related to classification and regression tasks, respectively.

message passing networks [70, 2] to quantitative structure-activity relationship regression tasks; [12] utilize Bayesian optimization to conduct reliable predictions of Nanoporous material properties.

Despite their contributions, these studies exhibit several constraints: **1)** the variety of uncertainty estimation methods considered is limited with some effective strategies being overlooked; **2)** each research focuses on a limited range of properties such as quantum mechanics, failing to explore a broader spectrum of tasks; and **3)** none embraces the power of recent pre-trained backbone models, which have shown remarkable performance in prediction but may lead to different results in UQ. To the best of our knowledge, a comprehensive evaluation of UQ methods applied to pre-trained molecular representation models is currently lacking and warrants further investigation.

We present MUBEN, a benchmark designed to assess the performance of UQ methods applied to the molecular representation models for property prediction on various metrics, as illustrated in Figure 1. It encompasses UQ methods from different categories, including deterministic prediction, Bayesian neural networks, post-hoc calibration, and ensembles (§ 3.2), on top of a set of molecular representation (backbone) models, each relies on a molecular descriptor from a distinct perspective (§ 3.1). MUBEN delivers intriguing results and insights that may guide the selection of backbone model and/or UQ methods in practice (§ 4). We structure our code, available at https://github.com/paper-submit-account/MUBen, to be user-friendly, easily transferrable, and extendable, with the hope that this work will promote the future development of UQ methods, pre-trained models, or applications within the domains of materials science and drug discovery.

## 2 Problem Setup

Let $\boldsymbol{x}$ denote a descriptor of a molecule, such as SMILES or a 2D topological graph. We define $y$ as the label with domain $y \in \{1, \ldots, K\}$ for $K$-class classification or $y \in \mathbb{R}$ for regression. $\boldsymbol{\theta}$ denotes all parameters of the molecular representation network and the task-specific layer. We assume that a training dataset $\mathcal{D}$ consists of $N$ i.i.d. samples $\mathcal{D} = \{(\boldsymbol{x}_n, y_n)\}_{n=1}^{N}$. Our goal is to fine-tune the parameters $\boldsymbol{\theta}$ on $\mathcal{D}$ and learn a calibrated predictive probability model $p_{\boldsymbol{\theta}}(y|\boldsymbol{x}_n)$. The UQ metrics are briefly illustrated below with more details being provided in appendix E.

**Negative Log-Likelihood**   Negative Log-Likelihood (**NLL**) is often utilized to assess the quality of model uncertainty on holdout sets for both classification and regression tasks. Despite being a proper scoring rule as per Gneiting's framework [25], certain limitations such as overemphasizing tail probabilities [63] make it inadequate to serve as the only UQ metric.

**Brier Score**   Brier Score (**BS**) [7] is the mean squared error between the predicted probability and ground-truth label $\text{BS} = \frac{1}{K} \sum_y (p_{\boldsymbol{\theta}}(y|\boldsymbol{x}_n) - \mathbf{1}_{y=y_n})^2$, where $\mathbf{1}$ is the indicator function. It serves as another proper scoring rule for classification tasks.

**Calibration Errors**   Calibration Errors measure the correspondence between predicted probabilities and empirical accuracy. For classification, we use Expected Calibration Error (**ECE**) [58], which first divides the predicted probabilities $\{p_{\boldsymbol{\theta}}(y_n \mid \boldsymbol{x}_n)\}_{n=1}^{N}$ into $S$ bins $B_s = \{n \in \{1, \ldots, N\} \mid p_{\boldsymbol{\theta}}(y_n \mid \boldsymbol{x}_n) \in (\rho_s, \rho_{s+1}]\}$ with $s \in \{1, \ldots, S\}$ and then com-

pute the $L_1$ loss between the accuracy and predicted probabilities within each bin: ECE $=$ $\sum_{s=1}^{S} \frac{|B_s|}{N} \left| \frac{1}{|B_s|} \sum_{n \in B_s} \mathbf{1}_{y_n = \hat{y}_n} - \frac{1}{|B_s|} \sum_{n \in B_s} p_{\boldsymbol{\theta}}(\hat{y}_n \mid \boldsymbol{x}_n) \right|$, where $\hat{y}_n = \arg\max_y p_{\boldsymbol{\theta}}(y \mid \boldsymbol{x}_n)$ is the $n$-th prediction. Similarly, for regression, we use Regression Calibration Error (**CE**), [43], which calculates the true frequency of the predicted points lying in each confidence interval against the predicted fraction of points in that interval: CE $= \frac{1}{S} \sum_{s=1}^{S} (\frac{s}{S} - |\{n \in \{1, \ldots, N\} \mid F_{\boldsymbol{\theta}}(y_n \mid \boldsymbol{x}_n) \leq \frac{s}{S}\}|/N)^2$, where $\frac{s}{S}$ is the expected quantile and $F_{\boldsymbol{\theta}}(y_n | \boldsymbol{x}_n)$ represents the predicted quantile value of $y_n$, *e.g.*, Gaussian CDF $\Phi(y_n; \hat{y}_n, \hat{\sigma}_n)$ parameterized by the predicted mean $\hat{y}_n$ and variance $\hat{\sigma}_n$ if we assume the labels are subject to the Gaussian distribution.

## 3 Experiment Setup

### 3.1 Backbone Models

Molecular descriptors, such as fingerprints, SMILES strings, and graphs, package the structural data of a molecule into a format that computational algorithms can readily process. When paired with distinct model architectures carrying divergent inductive biases, these descriptors can offer varied benefits depending on the specific task. Therefore, we select 4 primary backbone models that accept distinct descriptors as input. We also include 2 supplementary backbones to cover other aspects such as model architecture and pre-training objectives.

For primary backbone models, we select various pre-trained Transformer encoder-based models, each is SOTA or close to SOTA for their respective input format: **1) ChemBERTa** [10, 1], which accepts SMILES strings as input; **2) GROVER** [65], pre-trained with 2D molecular graphs via a Transformer bolstered by a dynamic message-passing graph neural network; and **3) Uni-Mol** [96], which incorporates 3D molecular conformations into its input, and pre-trains specialized Transformer architectures to encode the invariant spatial positions of atoms and represent the edges between atom pairs. We also implement a **4)** fully connected deep neural network (**DNN**) that uses 200-dimensional RDKit features which are regarded as fixed outputs of a hand-crafted backbone feature generator. This simple model aims to highlight the performance difference between heuristic feature generators and the automatic counterparts that draw upon self-supervised learning processes for knowledge acquisition. In addition, we have two other backbones: **5) TorchMD-NET**, as introduced in [78] and pre-trained according to [94]. This model is designed to handle 3D inputs and leverages an equivariant Transformer architecture aimed at quantum mechanical properties. **6) GIN** from [91], which processes 2D molecular graphs. GIN is theoretically among the most powerful graph neural networks (GNNs) available, offering an additional randomly initialized backbone baseline with less sophisticated input features. Compared with the primary one, these models are constrained in pre-training: TorchMD-NET is exclusively pre-trained on quantum mechanical data, whereas GIN does not involve any pre-processed features. Please refer to appendix C for more details.

### 3.2 Uncertainty Quantification Methods

**Deterministic Uncertainty Prediction**    In standard practice, deep classification networks utilize SoftMax or Sigmoid outputs to distribute prediction weights among target classes. Such weights serve as an estimate of the model's uncertainty. In regression tasks, rather than one single output value, models often parameterize an independent Gaussian distribution with predicted means and variances for each data point, and the variance magnitude acts as an estimate of the model's uncertainty.

An alternate loss function for classification is the **Focal Loss** [47, 55], which minimizes a regularised KL divergence between the predicted values and the true labels. The regularization increases the entropy of the predicted distribution while the KL divergence is minimized, mitigating the model overconfidence and improving the uncertainty representation.

**Bayesian Learning and Inference**    BNNs estimate the probability distribution over the parameters of the network. Specifically, BNNs operate on the assumption that the network layer weights follow a specific distribution that is optimized through maximum a posteriori estimation during training. During inference, multiple network instances are randomly sampled from the learned distributions. Each instance then makes independent predictions, and the prediction distribution inherently captures the desired uncertainty information. Some notable methods that employ this approach include:

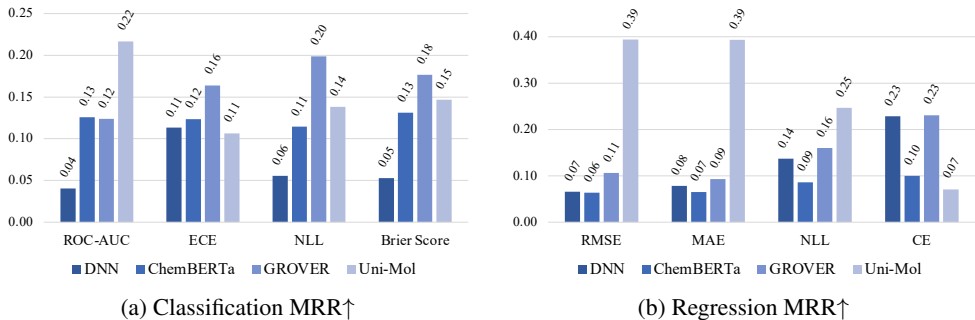

|  | (a) Classification MRR↑ | (b) Regression MRR↑ |

Figure 2: MRR of the backbone models for each metric, each is macro-averaged from the reciprocal ranks of the results of all corresponding UQ methods on all datasets. MRRs accentuate top results, offering a complementary perspective on the backbone performance.

- Bayes by Backprop (**BBP**) [6] first introduces Monte Carlo gradients, an extension of the Gaussian reparameterization trick [39], to learn the posterior distribution directly through backpropagation.
- Stochastic Gradient Langevin Dynamics (**SGLD**) [86] applies Langevin dynamics to infuse noise into the stochastic gradient descent training process, thereby mitigating the overfitting of network parameters. The samples generated via Langevin dynamics can be used to form Monte Carlo estimates of posterior expectations during the inference stage.
- **MC Dropout** [19] demonstrates that applying dropout is equivalent to a Bayesian approximation of the deep Gaussian process [11]. Uncertainty is derived from an ensemble of multiple stochastic forward passes with dropout enabled.
- SWA-Gaussian (**SWAG**) [51] provides an efficient way of estimating Gaussian posteriors over network weights by utilizing stochastic weight averaging (SWA) [35] combined with low-rank & diagonal Gaussian covariance approximations.

**Post-Hoc Calibration** Post-hoc calibration is proposed to address the over-confidence issue of deterministic classification models by adjusting the output logits after training. The most popular method is **Temperature Scaling** [62, 28], which adds a learned scaling factor to the Sigmoid or SoftMax output activation to control their "spikiness". [28] shows that this simple step improves model calibration in general, yielding better UQ performance.

**Deep Ensembles** The ensemble method has long been used in machine learning to improve model performance [14] but was first adopted to estimate uncertainty in [44]. It trains a deterministic network multiple times with different random seeds and combines their predictions at inference. The ensembles can explore training modes thoroughly in the loss landscape and thus are robust to noisy and out-of-distribution examples [18].

### 3.3 Datasets

We carry out our experiments on MoleculeNet [90], a collection of widely utilized datasets covering molecular properties such as quantum mechanics, solubility, and toxicity. For classification, MUBEN incorporates BBBP, ClinTox, Tox21, ToxCast, SIDER, BACE, HIV, and MUV with all labels being binary; the first 5 concern **physiological** properties and the last 3 **biophysics**. For regression, we select ESOL, FreeSolv, Lipophilicity, QM7, QM8, and QM9, of which the first 3 contain **physical chemistry** properties and the rest fall into the **quantum mechanics** category. In line with previous studies [16, 96], MUBEN divides all datasets with scaffold splitting to minimize the impact of dataset randomness and, consequently, enhances the reliability of the evaluation. Moreover, scaffold splitting alienates the molecular features in each dataset split, inherently creating a challenging out-of-distribution (OOD) setup that better reflects the real-world scenario. For comparison, we also briefly report the results of random splitting. More details are in appendix B.

Table 1: Classification results. "↑" and "↓" imply that better performance is indicated by a larger or smaller value, respectively. Text in bold signifies the top-performing uncertainty quantification method within each individual backbone model, and cells in blue indicate the best performance across all backbone-UQ combinations. "ROC" represents ROC-AUC.

| | Tox21[a] | | | | ToxCast[a] | | | | Average Ranking[b] | | | |
|---|---|---|---|---|---|---|---|---|---|---|---|---|
| | ROC↑ | ECE↓ | NLL↓ | BS↓ | ROC↑ | ECE↓ | NLL↓ | BS↓ | ROC↓ | ECE↓ | NLL↓ | BS↓ |
| DNN-RDKit | | | | | | | | | | | | |
| Deterministic | 0.7386 | 0.0417 | 0.2771 | 0.0779 | 0.6222 | 0.1168 | 0.4436 | 0.1397 | 25.75 | 17.25 | 24.13 | 22.88 |
| Temperature | 0.7386 | **0.0342** | 0.2723 | 0.0773 | 0.6220 | 0.1114 | 0.4882 | 0.1398 | 25.75 | 15.38 | 21.25 | 19.88 |
| Focal Loss | 0.7374 | 0.1058 | 0.3161 | 0.0871 | 0.6289 | 0.1264 | 0.4389 | 0.1396 | 25.88 | 24.38 | 22.38 | 24.00 |
| MC Dropout | 0.7376 | 0.0356 | 0.2727 | 0.0763 | 0.6248 | 0.1093 | 0.4319 | 0.1358 | 26.50 | 13.00 | 19.38 | 18.63 |
| SWAG | 0.7364 | 0.0438 | 0.2793 | 0.0790 | 0.6207 | 0.1175 | 0.4441 | 0.1400 | 26.38 | 18.50 | 25.63 | 23.50 |
| BBP | 0.7243 | 0.0422 | 0.2847 | 0.0814 | 0.6020 | 0.1443 | 0.4673 | 0.1510 | 22.75 | 19.13 | 19.38 | 21.38 |
| SGLD | 0.7257 | 0.1192 | 0.3455 | 0.0978 | 0.5319 | 0.3054 | 0.6685 | 0.2378 | 27.75 | 26.00 | 28.88 | 27.88 |
| Ensembles | **0.7540** | 0.0344 | **0.2648** | **0.0746** | **0.6486** | 0.0900 | 0.4008 | 0.1292 | **20.00** | 7.13 | **11.75** | **13.13** |
| ChemBERTa | | | | | | | | | | | | |
| Deterministic | 0.7542 | 0.0571 | 0.2962 | 0.0812 | 0.6554 | 0.1209 | 0.4313 | 0.1330 | 15.63 | 17.38 | 18.88 | 19.38 |
| Temperature | 0.7542 | 0.0424 | 0.2744 | 0.0792 | 0.6540 | 0.1067 | 0.4817 | 0.1313 | 15.88 | 12.00 | 13.88 | 15.38 |
| Focal Loss | 0.7523 | 0.0969 | 0.3052 | 0.0845 | 0.6442 | 0.1197 | 0.4243 | 0.1346 | 17.63 | 20.13 | 17.88 | 20.63 |
| MC Dropout | 0.7641 | 0.0423 | 0.2697 | **0.0744** | 0.6624 | 0.1069 | 0.4070 | 0.1276 | 12.50 | **10.75** | 10.63 | **10.00** |
| SWAG | 0.7538 | 0.0592 | 0.3008 | 0.0818 | 0.6556 | 0.1202 | 0.4305 | 0.1327 | 16.13 | 19.50 | 21.38 | 20.38 |
| BBP | 0.7433 | 0.0459 | 0.2765 | 0.0780 | 0.5814 | 0.1276 | 0.4545 | 0.1469 | 20.88 | 19.00 | 16.00 | 19.50 |
| SGLD | 0.7475 | 0.0504 | 0.2784 | 0.0795 | 0.5436 | 0.2238 | 0.5602 | 0.1881 | 21.13 | 19.88 | 19.13 | 18.63 |
| Ensembles | **0.7681** | **0.0440** | **0.2679** | 0.0750 | **0.6733** | **0.1037** | **0.3986** | **0.1258** | **12.38** | 13.00 | **11.63** | 12.25 |
| GROVER | | | | | | | | | | | | |
| Deterministic | 0.7808 | 0.0358 | 0.2473 | 0.0694 | 0.6587 | 0.1043 | 0.4091 | 0.1298 | 11.63 | 11.50 | 8.88 | 9.88 |
| Temperature | 0.7810 | **0.0291** | 0.2439 | 0.0686 | 0.6496 | 0.1424 | 0.4612 | 0.1424 | 12.63 | 8.88 | 7.50 | 9.25 |
| Focal Loss | 0.7779 | 0.1148 | 0.3052 | 0.0811 | 0.6359 | 0.1221 | 0.4365 | 0.1383 | 15.00 | 23.38 | 21.50 | 22.25 |
| MC Dropout | 0.7817 | 0.0346 | 0.2455 | 0.0689 | 0.6615 | **0.1009** | **0.4042** | **0.1288** | 11.25 | 10.50 | 7.63 | 8.63 |
| SWAG | 0.7837 | 0.0360 | 0.2482 | 0.0689 | 0.6603 | 0.1060 | 0.4114 | 0.1301 | 9.13 | 11.63 | 8.88 | 8.38 |
| BBP | 0.7697 | 0.0438 | 0.2552 | 0.0711 | 0.5995 | 0.1731 | 0.5090 | 0.1660 | 16.75 | 22.00 | 14.75 | 15.88 |
| SGLD | 0.7635 | 0.0402 | 0.2558 | 0.0716 | 0.5542 | 0.2712 | 0.6194 | 0.2139 | 18.75 | 18.63 | 15.25 | 15.63 |
| Ensembles | **0.7876** | 0.0316 | **0.2411** | **0.0675** | **0.6646** | 0.1034 | 0.4061 | 0.1290 | **8.50** | 8.13 | 5.38 | **6.88** |
| Uni-Mol | | | | | | | | | | | | |
| Deterministic | 0.7895 | 0.0454 | 0.2601 | 0.0716 | 0.6734 | 0.1020 | 0.3983 | 0.1274 | 11.50 | 15.50 | 16.13 | 13.88 |
| Temperature | 0.7896 | 0.0346 | 0.2483 | 0.0704 | 0.7028 | 0.1456 | 0.4566 | 0.1355 | 11.00 | 12.00 | 13.13 | 12.75 |
| Focal Loss | 0.7904 | 0.0972 | 0.2899 | 0.0785 | 0.6934 | 0.1227 | 0.4079 | 0.1284 | 10.00 | 23.50 | 19.38 | 20.50 |
| MC Dropout | 0.7891 | 0.0480 | 0.2628 | 0.0726 | 0.6833 | 0.1074 | 0.4015 | 0.1274 | 12.88 | 19.88 | 19.25 | 17.25 |
| SWAG | 0.7842 | 0.0593 | 0.2994 | 0.0728 | 0.6870 | 0.1085 | 0.4005 | 0.1271 | 10.13 | 22.13 | 22.25 | 18.25 |
| BBP | 0.7932 | 0.0396 | 0.2520 | 0.0703 | 0.6273 | 0.1296 | 0.4522 | 0.1456 | 12.13 | 17.00 | 15.50 | 16.25 |
| SGLD | 0.7887 | 0.0433 | 0.2569 | 0.0684 | 0.5700 | 0.1953 | 0.5207 | 0.1717 | 14.75 | 18.13 | 18.88 | 14.50 |
| Ensembles | 0.8052 | **0.0332** | **0.2389** | **0.0662** | 0.6841 | **0.0953** | **0.3877** | **0.1247** | **5.13** | 8.88 | **7.63** | **6.50** |
| TorchMD-NET[c] | 0.7793 | 0.0409 | 0.2614 | 0.0708 | 0.6540 | 0.1546 | 0.4424 | 0.1396 | - | - | - | - |
| GIN[c] | 0.6829 | 0.0634 | 0.3268 | 0.0840 | 0.5752 | 0.1381 | 0.4835 | 0.1477 | - | - | - | - |

[a] The "Tox21" and "ToxCast" columns present metric scores on representative exemplar datasets, highlighting the trends observable across all datasets.
[b] The "Average Ranking" columns provide the rank of each model's UQ metrics against all other backbone-UQ combinations averaged from all classification datasets; smaller number indicates better performance.
[c] We report the results from the best-performing UQ method—Deep Ensembles for TorchMD-NET and GIN. These backbones are not ranked together with the primary benchmark.

## 4 Results and Analysis

**Prediction Performance**   Theoretically, inserting UQ methods into the training pipeline does not guarantee better prediction on i.i.d. datasets. However, since we ensure OOD test points with scaffold splitting, UQ methods may mitigate the distribution gap, yielding better test results. The columns ROC-AUC, RMSE, and MAE in Tables 1, 2, and Figure 2 illustrate the predictive performance of each method. Examining from the lens of UQ methods, none provides a consistent guarantee of performance improvement over direct prediction, except for Deep Ensembles, and MC Dropout for regression. The randomness in the initialization and training trajectory of Deep Ensembles explores a broader range of loss landscapes, which partially addresses the distribution shift issue, as observed by [18]. MC Dropout samples multiple sets of model parameters from the Gaussian distributions centered at the fine-tuned deterministic net weights, which may flatten extreme regression abnormality triggered by OOD features. This phenomenon is less pronounced for classification due to the $(0, 1)$ output domain. However, other BNNs do not exhibit the same advantage. SWAG, while similar to MC Dropout *w.r.t.* training, might intensify training data overfitting due to the additional steps taken

Table 2: Regression results (lower is better) including two example datasets and the average ranking.

| | Lipophilicity | | | | QM9 | | | | Average Ranking | | | |
|---|---|---|---|---|---|---|---|---|---|---|---|---|
| | RMSE | MAE | NLL | CE | RMSE | MAE | NLL | CE | RMSE | MAE | NLL | CE |
| DNN-RDKit | | | | | | | | | | | | |
| Deterministic | 0.7575 | 0.5793 | 0.6154 | 0.0293 | 0.01511 | 0.01012 | -3.379 | 0.04419 | 16.67 | 15.67 | 11.33 | 10.50 |
| MC Dropout | 0.7559 | 0.5773 | 0.9071 | 0.0341 | 0.01480 | 0.01000 | -3.526 | 0.04327 | 15.17 | 15.17 | 11.33 | 10.67 |
| SWAG | 0.7572 | 0.5823 | 0.7191 | 0.0308 | 0.01524 | 0.01019 | -3.284 | 0.04495 | 18.00 | 17.67 | 14.00 | 12.50 |
| BBP | 0.7730 | 0.5938 | 0.7578 | 0.0305 | 0.01534 | 0.01025 | -3.347 | 0.04452 | 21.17 | 20.50 | 10.33 | 7.33 |
| SGLD | 0.7468 | 0.5743 | **0.2152** | **0.0090** | 0.01958 | 0.01437 | -3.335 | **0.00702** | 19.83 | 19.33 | 6.83 | **1.83** |
| Ensembles | **0.7172** | **0.5490** | 0.6165 | 0.0322 | **0.01430** | **0.00956** | **-3.602** | 0.04362 | **11.50** | **11.17** | **6.33** | 8.67 |
| ChemBERTa | | | | | | | | | | | | |
| Deterministic | 0.7553 | 0.5910 | 1.2368 | 0.0362 | 0.01464 | 0.00916 | -2.410 | 0.05468 | 17.83 | 16.67 | 18.83 | 14.67 |
| MC Dropout | **0.7142** | **0.5601** | 0.8178 | 0.0349 | 0.01412 | 0.00880 | -3.150 | 0.05133 | **14.33** | **13.83** | 15.00 | 13.67 |
| SWAG | 0.7672 | 0.5992 | 1.5809 | 0.0395 | 0.01477 | 0.00925 | -2.170 | 0.05535 | 19.33 | 18.50 | 20.33 | 15.83 |
| BBP | 0.7542 | 0.5869 | **0.4419** | **0.0279** | 0.01443 | 0.00928 | -2.593 | 0.05399 | 17.67 | 18.50 | 10.83 | 9.00 |
| SGLD | 0.7622 | 0.5982 | 0.8719 | 0.0355 | 0.01530 | 0.01012 | **-3.758** | **0.03378** | 19.83 | 20.50 | **9.50** | **8.50** |
| Ensembles | 0.7367 | 0.5763 | 0.9756 | 0.0360 | **0.01397** | **0.00868** | -2.876 | 0.05425 | 14.83 | **13.83** | 14.67 | 12.67 |
| GROVER | | | | | | | | | | | | |
| Deterministic | 0.6316 | 0.4747 | 2.1512 | 0.0478 | 0.01148 | 0.00678 | -0.787 | 0.06206 | 10.67 | 11.67 | 17.67 | 16.00 |
| MC Dropout | 0.6293 | 0.4740 | 2.0526 | 0.0476 | 0.01140 | 0.00676 | -1.100 | 0.06161 | 9.33 | 11.17 | 16.00 | 14.33 |
| SWAG | 0.6317 | 0.4750 | 2.3980 | 0.0485 | 0.01156 | 0.00678 | -0.477 | 0.06252 | 12.33 | 13.33 | 20.33 | 17.83 |
| BBP | 0.6481 | 0.5058 | 0.0789 | 0.0196 | 0.01179 | 0.00700 | -1.885 | 0.05909 | 14.17 | 15.67 | 7.67 | 4.00 |
| SGLD | 0.6360 | 0.4984 | **0.0544** | **0.0215** | 0.01359 | 0.00878 | **-3.785** | **0.02911** | 14.83 | 15.17 | **4.67** | **2.67** |
| Ensembles | **0.6250** | **0.4693** | 1.6046 | 0.0460 | **0.01143** | **0.00667** | -1.028 | 0.06199 | **8.17** | **8.67** | 13.50 | 14.00 |
| Uni-Mol | | | | | | | | | | | | |
| Deterministic | 0.6079 | 0.4509 | 0.8975 | 0.0425 | 0.00962 | 0.00538 | 0.014 | 0.06637 | 5.83 | 4.83 | 16.67 | 21.17 |
| MC Dropout | 0.5983 | 0.4438 | 1.3663 | 0.0440 | 0.00961 | 0.00535 | -0.251 | 0.06615 | 4.00 | 3.17 | 16.67 | 21.33 |
| SWAG | 0.6026 | 0.4476 | 1.0101 | 0.0453 | 0.00969 | 0.00541 | -0.462 | 0.06597 | 6.33 | 6.17 | 19.67 | 22.67 |
| BBP | 0.6044 | 0.4469 | **0.0679** | **0.0306** | 0.00952 | 0.00544 | -2.959 | 0.06179 | 3.17 | 3.00 | 4.33 | 10.33 |
| SGLD | 0.6040 | 0.4554 | 0.1565 | 0.0329 | 0.00950 | 0.00546 | **-4.209** | **0.04593** | **2.50** | 4.00 | **2.50** | 9.17 |
| Ensembles | **0.5809** | **0.4266** | 0.6450 | 0.0438 | **0.00948** | **0.00526** | -0.319 | 0.06629 | **2.50** | **1.83** | 11.00 | 20.67 |
| TorchMD-NET[a] | 1.0313 | 0.8196 | 0.8619 | 0.0195 | 0.00860 | 0.00464 | 2.262 | 0.06868 | - | - | - | - |
| GIN[a] | 0.8071 | 0.6515 | 0.3241 | 0.0020 | 0.01295 | 0.00814 | -3.521 | 0.04997 | - | - | - | - |

[a] We report the results from the Deep Ensembles UQ method for TorchMD-NET and GIN.

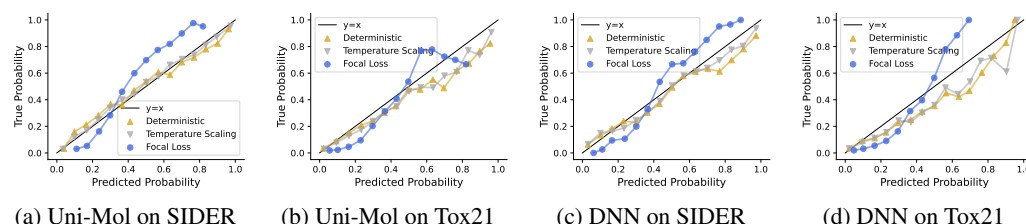

(a) Uni-Mol on SIDER  (b) Uni-Mol on Tox21  (c) DNN on SIDER  (d) DNN on Tox21

Figure 3: Calibration plot of example uncertainty estimation results. For both datasets, we draw the calibration based on the aggregated outputs from all tasks.

to fit the parameter distribution. The stochastic sampling employed by BBP and SGLD complicates network training and may impact the prediction performance as a result.

Looking from the perspective of primary backbones, Uni-Mol secures the best prediction performance for both classification and regression. The superior molecular representation capability of Uni-Mol is attributed to the large network size, the various pre-training data and tasks, and the integration of results from different conformations of the same molecule. When contrasting DNN, ChemBERTa, and GROVER, it becomes apparent that the expressiveness of the molecular descriptors varies for different molecules/tasks. Interestingly, pre-trained models do not invariably surpass heuristic features when integrated with UQ methods. Selecting a model attuned to the specific task is advised over an indiscriminate reliance on "deep and complex" networks.

**Uncertainty Quantification Performance**    Tables 1, 2 and Figure 2 depict the uncertainty estimation performances via ECE, NLL, Brier score, and CE columns. One discernible trend is the consistent performance enhancement from Deep Ensembles even when the number of ensembles is limited, such as the QM9 case (appendix D). MC Dropout also exhibits a similar trend, albeit less

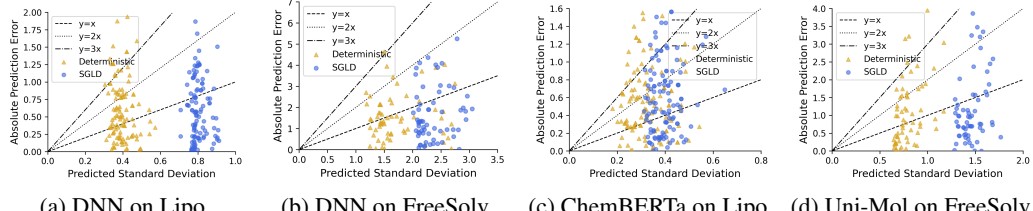

(a) DNN on Lipo     (b) DNN on FreeSolv     (c) ChemBERTa on Lipo     (d) Uni-Mol on FreeSolv

Figure 4: The absolute error between the model-predicted mean and true labels against the predicted standard deviation. We compare the performance of SGLD with the deterministic prediction on different backbones and datasets. The "$y = kx$" lines indicate whether the true labels lie within the $k$-std range of the predicted Gaussian. Also, a model is perfectly calibrated when its output points are arranged on an "$y = kx$" line for an arbitrary $k$.

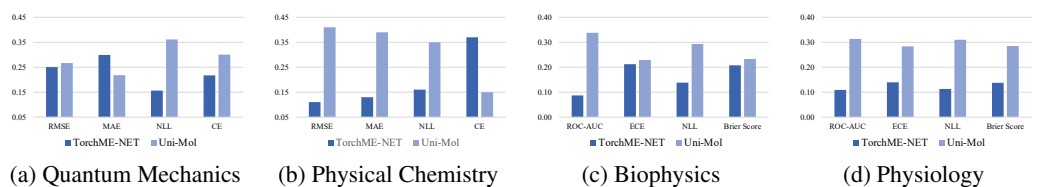

(a) Quantum Mechanics    (b) Physical Chemistry    (c) Biophysics    (d) Physiology

Figure 5: MRRs of TorchMDNet and Uni-Mol on datasets grouped by dataset property categories. MRR calculations are confined to results from these two backbones. Only relative values matter.

pronounced. Despite a marginal compromise in prediction accuracy, Temperature Scaling emerges as another method that almost invariably improves calibration, as evidenced in the average ranking columns—a finding that aligns with [28]. Figure 3 shows that deterministic prediction tends to be over-confident, and Temperature Scaling mitigates this issue. However, it is susceptible to calibration-test alignment. If the correlation is weak, Temperature Scaling may worsen the calibration, as presented in the ToxCast dataset. In contrast, Focal Loss does not work as impressively for binary classification, which is also observed by [55]. It is hypothesized that while the Sigmoid function sufficiently captures model confidence in binary cases, Focal Loss could over-regularize parameters, diminishing the prediction sharpness to an unreasonable value, a conjecture verified by the 'S'-shaped calibration curves in Figure 3.

Although limited in classification efficacy, both BBP and SGLD deliver commendable performance in predicting regression uncertainty, capturing 7 out of 8 top ranks for 4 backbones on 2 metrics, narrowly missing out to Deep Ensembles for the remaining one (Table 2). Yet, their inconsistent improvement of RMSE and MAE implies a greater influence on variance prediction than mean. Figure 4 reveals SGLD's tendency to "play safe" by predicting larger variances, while the deterministic method is prone to over-confident by ascribing small variances even to its inaccurate predictions. In addition, we do not observe a better correlation between SGLD's error and variance. We assume that the noisy training trajectory prevents SGLD and BBP to sufficiently minimize the gap between the predicted mean and true labels, thus encouraging them to maintain larger variances to compensate for the error. Please refer to appendix D.2 and appendix F.1 for more analysis on UQ performance.

Figure 2 indicates that Uni-Mol exhibits subpar calibration, particularly for regression. Comparing Uni-Mol, ChemBERTa, and DNN in Figure 4, we notice that larger models such as Uni-Mol are more confident to their results, as illustrated by their smaller variances and larger portion of (std, error) points exceed above the $y = kx$. This could be attributed to shared structural features in 3D conformations in the training and test molecules that remain unobserved in simpler descriptors. While this similarity benefits property prediction, it also potentially misleads the model into considering data points as in-distribution, thereby erroneously assigning high confidence.

**TorchMD-NET and GIN** Our analysis also encompasses TorchMD-NET and GIN, two additional backbone models excluded from the primary benchmark due to their limited capabilities. As presented in the tables and Figure 5, TorchMD-NET's performance is on par with Uni-Mol when predicting

Table 3: Performance with frozen backbone weights and on random split datasets compared with the original scores in Table 1 and 2. The result is calculated as $(\text{new} - \text{ori})/\text{ori}$ and is macro-averaged over all datasets, backbones and UQ methods. appendix F shows the complete results.

| | Classification | | | | Regression | | | |
|---|---|---|---|---|---|---|---|---|
| | ROC-AUC (%)↑ | ECE (%)↓ | NLL (%)↓ | Brier Score (%)↓ | RMSE (%)↓ | MAE (%)↓ | NLL (%)↓ | CE (%)↓ |
| Frozen Backbone | -24.07 | 145.13 | 53.43 | 88.98 | 78.60 | 92.40 | 58.18 | -46.06 |
| Random Split | 13.25 | -35.19 | -33.87 | -37.35 | -26.34 | -31.36 | -53.87 | 19.06 |

quantum mechanical properties but falls short in others. This outcome aligns with expectations, given that TorchMD-NET's architecture is tailored specifically for predicting quantum mechanical properties [78]. Moreover, it is pre-trained on the relatively niche dataset PCQM4Mv2 [32] with only the denoising objective, which might be suitable for molecular dynamics but limited for other properties. In contrast, Uni-Mol stands out as a versatile model, benefiting from diverse pre-training objectives that ensure superiority across various tasks. On the other hand, GIN's performance is consistently inferior to other models including DNN, with examples presented in Tables 1 and 2, likely due to the limited expressiveness of 2D graphs and the GNN architecture when pre-training is absent. Please refer to appendix F for complete results.

**Frozen Backbone and Randomly Split Datasets**   Table 3 demonstrates a notable drop in prediction performance when backbone weights are fixed; and random splits outperform scaffold splits. This is consistent with intuition: if backbone models serve solely as feature extractors instead of a part of the trainable predictors, they are less expressive for downstream tasks. Additionally, in-distribution features tend to be more predictable. An interesting exception emerges in regression calibration error, where frozen backbones perform better and random splits score lower. Upon examining the predicted values, we note that predictions for random splits exhibit a sharper distribution, *i.e.*, smaller $\hat{\sigma}$. This suggests that the models are more confident in regressing in-distribution data, aligning with our previous observation for Uni-Mol. Conversely, frozen backbones are less prone to overfitting due to their constrained expressiveness. This behavior underscores the original models' capability to distinguish between in-distribution and OOD features and assign confidence scores with precision.

## 5   Conclusion, Limitation, and Future Works

We introduce MUBEN, a benchmark that assesses an array of UQ methods across diverse categories, leveraging backbone models that incorporate different descriptors for multiple molecular property prediction tasks. Our results reveal that Deep Ensembles assures performance enhancement under all circumstances but with significant computation cost. Temperature scaling and MC Dropout are simple yet effective for classification, while BBP and SGLD are better suited for regression UQ. Among backbones, Uni-Mol, leveraging the expressiveness of 3D molecular conformations, is the most powerful but prone to overconfidence. Backbones with other descriptors are advantageous in different conditions and should be selected according to specific usage cases.

Given the rapid advancements in molecular representation learning and UQ methods, MUBEN cannot encompass all possible combinations, forcing us to focus on a curated selection of representative methods. Furthermore, we use coarse-grained hyperparameter grids to maintain experimental feasibility, which makes MUBEN, while indicative of trends, might not present the best possible results. We remain committed to refining MUBEN and welcome contributions from the broader community to enhance its inclusivity and utility for research in this field and related domains.

## Acknowledgments and Disclosure of Funding

This work was supported in part by ONR MURI N00014-17-1-2656, NSF IIS-2008334, IIS-2106961, and CAREER IIS-2144338.

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

# A    Related Works

**Pre-Training Molecular Representation Models**    Existing pre-trained models for molecular representation learning fall into three categories based on the type of molecular descriptors. The first category includes models like ChemBERTa [10], SMILES-BERT [82], and Molformer [66], which leverage string-based input descriptors such as SMILES [85] and pre-trained with masked language modeling (MLM) objectives [13]. Given that molecules can be represented as 2D topological graphs, the second category of pre-training strategies employs techniques such as context prediction [33, 65], contrastive learning [77, 93, 83], and replaced component detection [45, 37] to learn molecular representation models with the awareness of the 2D graph context. Recently, the research community has also started to explore pre-training with 3D geometric graphs as the third category [16, 48, 96, 75]. For example, GEM [16] incorporates bond angles and lengths as additional edge attributes to enrich the 3D information; Uni-Mol [96] employs a 3D position denoising loss and a masked atom prediction loss to effectively learn 3D spatial representations of molecules.

**Uncertainty Quantification**    Uncertainty estimation plays a crucial role in stimulating deep neural networks to acknowledge what they don't know, which has been applied to various domains such as object detection [9, 29] and decision making [3, 41]. UQ on neural networks generally takes two directions: Bayesian and non-Bayesian. Bayesian neural networks (BNNs) quantify predictive uncertainty by imposing probability distributions over model parameters instead of using point estimates. While BNNs provide a principled way of UQ, exact inference of parameter posteriors is often intractable. To circumvent this difficulty, existing works adopt Monte Carlo dropout [19], variational inference [6, 38] and Markov chain Monte Carlo (MCMC) [86, 51] for posterior approximation. Along the other direction, non-Bayesian methods estimate uncertainty by assembling multiple networks [44, 18], adopting specialized output distribution [70, 2, 71] or training scheme [47, 55], or calibrating model outputs through additional network layers during validation and test [62, 28, 72, 4, 5]. Recently, there have been several UQ benchmarks from the general domain [57, 69], biomedical engineering [53, 26] and materials science [34, 68, 79, 89]. MUBEN also targets the last domain, but covers a more diverse suite of tasks, stronger pre-trained molecular representation models, and a comparatively more comprehensive set of UQ methods. We hope MUBEN can serve as a standard benchmark to faithfully evaluate the ability of UQ approaches.

# B    Dataset Details

In this section, we provide details about the datasets used in our study. We follow the approach of previous works [16, 96] and select a subset of widely used and publicly available datasets from the MoleculeNet benchmark [90]. The datasets cover both classification and regression tasks from 4 property categories, including Physiology, Biophysics, Physical Chemistry and Quantum Mechanics. The classification datasets include:

- **BACE** provides binary binding properties for a set of inhibitors of human $\beta$-secretase 1 (BACE-1) from experimental values from the published papers;
- **BBBP (Blood-Brain Barrier Penetration)** studies the classification of molecules by their permeability of the blood-brain barrier;
- **ClinTox** consists of two classification tasks for drugs: whether they are absent of clinical toxicity and whether they are approved by the FDA;
- **Tox21 (Toxicology in the 21st Century)** consists of qualitative toxicity measurements of $8,014$ compounds on 12 different targets;
- **ToxCast** provides toxicology data of $8,576$ compounds on 617 different targets;
- **SIDER (Side Effect Resource)** contains marketed drugs and adverse drug reactions (ADR) extracted from package inserts;
- **HIV**, introduced by the Drug Therapeutics Program (DTP) AIDS Antiviral Screen, contains compounds that are either active or inactive against HIV replication;
- **MUV (Maximum Unbiased Validation)**, a challenging benchmark dataset selected from PubChem BioAssay for validation of virtual screening techniques, contains $93,087$ compounds tested for activity against 17 different targets.

The regression datasets include:

- **ESOL** provides experimental values for water solubility data for $1,128$ compounds;

Table 4: Dataset statistics in MUBEN.

| Property Category | Dataset | # Compounds | # Tasks | Average LIR[a] | Max LIR[a] |
|---|---|---|---|---|---|
| | | Classification | | | |
| Physiology | BBBP | 2,039 | 1 | 0.7651 | 0.7651 |
| | ClinTox | 1,478 | 2 | 0.9303 | 0.9364 |
| | Tox21 | 7,831 | 12 | 0.9225 | 0.9712 |
| | ToxCast | 8,575 | 617 | 0.8336 | 0.9972 |
| | SIDER | 1,427 | 27 | 0.7485 | 0.9846 |
| Biophysics | BACE | 1,513 | 1 | 0.5433 | 0.5433 |
| | HIV | 41,127 | 1 | 0.9649 | 0.9649 |
| | MUV | 93,087 | 17 | 0.9980 | 0.9984 |
| | | Regression | | | |
| Physical Chemistry | ESOL | 1,128 | 1 | - | - |
| | FreeSolv | 642 | 1 | - | - |
| | Lipophilicity | 4,200 | 1 | - | - |
| Quantum Mechanics | QM7 | 7,160 | 1 | - | - |
| | QM8 | 21,786 | 12 | - | - |
| | QM9 | 133,885 | $3^{(b)}$ | - | - |

[a] LIR stands for "label imbalance ratio", which is calculated as $\text{LIR}_k \in [0.5, 1] = \max\{p_{\text{pos}}, p_{\text{neg}}\}$ for a task indexed by $k \in \{1, \ldots, K\}$, where $p_{\text{pos}}$ and $p_{\text{neg}}$ are the proportions of positive and negative samples in the dataset, respectively. $\text{LIR} = 0.5$ indicates a balanced dataset and $\text{LIR} = 1$ indicates a completely imbalanced one, with either $y = 0$ or $y = 1$ being absent. Average LIR $= \sum_{k=1}^{K} \text{LIR}_k$, and Max LIR $= \max_k\{\text{LIR}_k\}$. LIR is calculated on the entire dataset, and the number could slightly vary for each partition.
[b] QM9 dataset contains 12 tasks, but we follow [16, 96] and use only 3 most popular ones (homo, lumo, and gap).

- **FreeSolv (Free Solvation Database)** contains experimental and calculated hydration free energies for 643 small molecules;
- **Lipophilicity** contains experimental results of octanol/water distribution coefficient for 4,200 compounds;
- **QM7/8/9** are subsets of GDB-13 and GDB-17 databases containing quantum mechanical calculations of energies, enthalpies, and dipole moments for organic molecules with up to 7/8/9 "heavy" atoms, respectively. For QM9, we follow previous works and select 3 tasks that predict properties (homo, lumo, and gap) with a similar quantitative range out of the total 12 [16, 96].

A summary of the dataset statistics is provided in Table 4. It is worth noting that some datasets, such as HIV and MUV, exhibit a high degree of class imbalance. This characteristic adds further challenges to the tasks of molecular property prediction and uncertainty quantification.

**Scaffold Splitting** MUBEN primarily focuses on dataset split by molecule scaffolds, which separates training, validation and test features as much as possible. This creates out-of-distribution (OOD) test sets and signifies the model's capability of processing patterns absent in the fine-tuning process. For all datasets, we use the conventional 8:1:1 training, validation, and test ratio. The raw datasets are available on the MoleculeNet website.[1] We take advantage of the pre-processed version shared in [96] and use the same splits as in their work.[2]

**Random Splitting** To explore the impact of dataset splitting techniques, we conduct experiments on four classification datasets (BACE, BBBP, ClinTox, Tox21) and four regression datasets (ESOL, FreeSolv, Lipophilicity, QM7) using random splitting. Despite their limited sizes, these datasets effectively represent the overall performance of the models. Each dataset is divided into three separate 8:1:1 training-validation-test sets using random seeds of 0, 1, and 2. Each backbone-UQ combination is trained once per split. We then average the results over 3 runs from the splits to determine the final

---

[1]https://moleculenet.org/
[2]Repository at https://github.com/dptech-corp/Uni-Mol/tree/main/unimol and download at https://bioos-hermite-beijing.tos-cn-beijing.volces.com/unimol_data/finetune/molecular_property_prediction.tar.gz.

Table 5: Model statistics.

| Model | # Parameters (M) | Average Time per Training Step (ms)[a] |
|---|---|---|
| DNN | 0.158 | 5.39 |
| ChemBERTa | 3.43 | 30.18 |
| GROVER | 48.71 | 334.47 |
| Uni-Mol | 47.59 | 392.55 |
| TorchMD-NET | 7.23 | 217.29 |
| GIN | 0.26 | 7.21 |

[a] All models are evaluated on the BBBP dataset with a batch size of 128. We only measure the time for forward passing, backward passing, and parameter updating. We train the model for 6 epochs and take the average of the last 5 to reduce the impact of GPU initialization.

metrics for each dataset. Deep Ensembles was the only exception, being trained 3 times for each dataset split using different seeds, totaling 9 training cycles per dataset. All other training settings were consistent with those in the scaffold-splitting experiments.

## C  Backbone Models and Implementation Details

We use Sigmoid output activation and binary cross-entropy (BCE) loss function on all classification models unless the UQ method specifies a different objective. For regression, we map the training labels to a standard Gaussian distribution on which the model is trained. During inference, the predicted values are converted back to their original distribution according to the label mean and variance from the training set before computing the metrics. For datasets with multiple prediction tasks such as QM9, a classification model features one output head for each task, while regression models have two—one for predicting mean $\hat{y}$ and the other for variance $\hat{\sigma}$, which is guaranteed to be positive using a SoftPlus activation [44]. Regression models are trained with Gaussian NLL objective.

Our experimental code is designed on the PyTorch framework [61]. The experiments are conducted on a single NVIDIA A100 Tensor Core GPU with a memory capacity of 80GB. The backbone models are fine-tuned using the AdamW optimizer [50] with a weight decay rate of 0.01 using full-precision floating-point numbers (FP32) for maximum compatibility. We apply different learning rates, numbers of training epochs and batch sizes for the backbone models, as specified in the following paragraphs. Unless the UQ method has other requirements, we adopt early stopping to select the best-performed checkpoint on the validation set, and all models have achieved their peak validation performance before the training ends. ROC-AUC is selected to assess classification validation performance. For regression, we follow [90] and use RMSE for Physical Chemistry properties and MAE for Quantum Mechanics. Notice that these metrics only concern prediction, and do not take the uncertainty into account during validation steps (appendix E.1). Table 5 presents the number of parameters and average time per training step for each backbone model. The training time is calculated for each model update step instead of each epoch based on our implementation, which might be slower than the models' original realizations. Below, we offer a detailed description of each backbone model's architecture and its implementation.

**ChemBERTa**  ChemBERTa [10, 1] leverages the RoBERTa model architecture and pre-training strategy [49] but with fewer layers, attention heads, and smaller hidden dimensionalities. Unlike language models which process sentences in natural language, ChemBERTa uses Simplified Molecular-Input Line-Entry System (SMILES) [85] strings as input. This representation is a compact and linear textual depiction of a molecule's structure that's frequently employed in cheminformatics. ChemBERTa pre-training adopts a corpus of 77M SMILES strings from PubChem [84], along with the masked language modeling (MLM) objective [13].

ChemBERTa is built with the HuggingFace Transformers library [88], and its pre-trained parameters are shared through the Huggingface's model hub. We retain the default model architecture and use the `DeepChem/ChemBERTa-77M-MLM` checkpoint for ChemBERTa's weight initialization.[3] We employ

---

[3]Accessible at https://huggingface.co/DeepChem/ChemBERTa-77M-MLM.

the last-layer hidden state corresponding to the token `[CLS]` to represent the input SMILES sequence and attach the output heads specified by the tasks/UQ methods on top of it for property prediction. On all datasets, ChemBERTa is fine-tuned with a learning rate of $10^{-5}$, a batch size of $128$, and for $200$ epochs. A tolerance of $40$ epochs for early stopping is adopted.

**GROVER** GROVER [65] is pre-trained on 11M unlabeled molecules as 2D molecular graphs from ZINC15 [76] and ChEMBL [21]. To enhance pre-training on molecular graphs, GROVER integrates the graph neural network (GNN)-based Message Passing Neural Networks (MPNN) [24] into the Transformer encoder architectures. It also devises self-supervised objectives from different molecular structural granularities to better capture the rich structural and semantic information of molecules. Specifically, GROVER replaces the linear layer used for mapping the input query, key, and value vectors of each multi-head attention module into the head-dependent latent feature spaces in the Transformer encoder with a dynamic message passing network. This network aggregates the latent features from the $k$-hop neighboring nodes, where $k$ is a random integer drawn from a pre-defined Uniform of Gaussian. In other words, GROVER puts GNN layers before each head's self-attention computation to better encode the graph structure of molecules. To facilitate learning, GROVER introduces training objectives including contextual property prediction, which predicts the statistical properties of masked subgraphs of arbitrary sizes, and graph-level motif prediction, which predicts the classes of the masked functional groups. GROVER uses a readout function to aggregate node features, which are then passed into linear layers for property prediction.

For implementation, we use the GROVER-base checkpoint for our model initialization.[4] We incorporate the "node-view" branch as discussed above and disregard the "edge-view" architecture detailed in the appendix of [65], in accordance with the default settings in their GitHub repository. Under this configuration, the model generates $2$ sets of node embeddings (but no edge embeddings), one from the node hidden states and another from the edge hidden states [65]. Each set of embeddings is passed into $2$ linear layers with GELU [30] activation and a dropout ratio of $0.1$ post-readout layer, which simply averages the embeddings in the default configuration. Each set of embeddings corresponds to an individual output branch, predicting the properties independently. In line with the original implementation, we compute the loss for each branch individually during fine-tuning and apply a squared Euclidean distance regularization with a coefficient of $0.1$ between the two. During inference, we average the logits from the $2$ branches to generate the final output logits.

In our experiments, we configure the fine-tuning batch size at $256$, and the number of epochs at $100$ with a tolerance of $40$ epochs for early stopping. The learning rate is set at $10^{-4}$, and the entire model has a dropout ratio of $0.1$. We substitute the original Noam learning rate scheduler [80] with a linear learning rate scheduler with a $0.1$ warm-up ratio for easier implementation. No substantial differences in model performance were observed between the two learning rate schedulers.

**Uni-Mol** Uni-Mol [96] is a universal molecular representation framework that enhances representational capacity and broadens applicability by incorporating 3D molecular structures as model input. For the property prediction task, Uni-Mol undergoes pre-training on 209M 3D conformations of organic molecules gathered from ZINC15 [76], ChEMBL [21], and a database comprising 12M purchasable molecules [96]. It portrays atoms as tokens and utilizes pair-type aware Gaussian kernels to encode the positional information in the 3D space, thereby ensuring rotational and translational invariance. Furthermore, Uni-Mol introduces a pair-level representation by orchestrating an "atom-to-pair" communication—updating positional encodings with query-key products—and a "pair-to-atom" communication—adding pair representation as bias terms in the self-attention atom update procedure. For pre-training, Uni-Mol employs masked atom prediction, akin to BERT's MLM, corrupting 3D positional encodings with random noise at a $15\%$ ratio. Additionally, the model is tasked with restoring the corrupted Euclidean distances between atoms and the coordinates for atoms.

Our codebase is developed atop the publicly accessible Uni-Mol repository,[5] and their pre-trained checkpoint for molecular prediction serves as our model initialization.[6] During fine-tuning, Uni-Mol generates $10$ sets of 3D conformations for each molecule, supplemented with an additional 2D molecular graph. Thereafter Uni-Mol samples one from these $11$ molecular representations for each

---

[4]Repository available at https://github.com/tencent-ailab/grover, and model can be downloaded at https://ai.tencent.com/ailab/ml/ml-data/grover-models/pretrain/grover_base.tar.gz.

[5]https://github.com/dptech-corp/Uni-Mol/tree/main.

[6]https://github.com/dptech-corp/Uni-Mol/releases/download/v0.1/mol_pre_no_h_220816.pt.

molecule at the beginning of every training epoch as the input feature. For inference, we average the logits from all 11 representations to generate the final output. We utilize the conformations prepared by the Uni-Mol repository in our implementation.

We configure the fine-tuning batch size at 128, the number of epochs at 100, and employ early stopping with a tolerance of 40 epochs. We use a linear learning rate scheduler with a 0.1 warm-up ratio and a peak learning rate of $5 \times 10^{-5}$. The model is trained with a dropout ratio of 0.1. Although the Uni-Mol repository provides a set of recommended hyperparameters, we observe no discernible improvement in model performance with these settings.

**DNN**  The Deep Neural Network (DNN) serves as a simple, randomly initialized baseline model designed to explore how heuristic descriptors like Morgan fingerprints [54] or RDKit features perform for molecular property prediction. DNN enables us to compare the pre-trained models, which learn the molecular representation automatically through self-learning, with heuristic molecular features, which are constructed manually, and investigate whether or under what circumstances the heuristic features can achieve comparable results. For the descriptor, we adopt the approach of previous work [90, 92, 65] and extract 200-dimensional molecule-level features using RDKit for each molecule, which are then used as DNN input.[7] The DNN consists of 8 fully connected 128-dimensional hidden layers with GELU activation and an intervening dropout ratio of 0.1. We find no performance gain from deeper or wider DNNs and thus assume that our model is fully capable of harnessing the expressivity of RDKit features. The model is trained with a batch size of 256 and a constant learning rate of $2 \times 10^{-4}$ for 400 epochs with an early stopping tolerance of 50 epochs.

**TorchMD-NET**  The architectural design of TorchMD-NET is detailed in [78], while its pre-training methodology is discussed in a separate study [94]. TorchMD-NET is an equivariant Transformer tailored for the prediction of quantum mechanical properties of 3D conformers. Unique elements of its architecture include a specialized embedding layer—encoding not just atomic numbers but also the interatomic distances influenced by neighboring atoms, a modified multi-head attention mechanism that integrates edge data, and an equivariant update layer computing atomic interactions. The model undergoes pre-training on the 3.4M PCQM4Mv2 dataset [59, 32], leveraging a denoising auto-encoder objective. This entails predicting Gaussian noise disturbances on atomic positions, mirroring techniques seen in prevalent diffusion models in computer vision [73].

To implement TorchMD-NET, we sourced the code and model checkpoint from [94].[8] We made minor architectural adjustments, replacing their single-head output block with our adaptive multi-head output layers. Consequently, we omitted the denoising objective during the fine-tuning process due to compatibility concerns. Our fine-tuning regimen for TorchMD-NET entails a batch size of 128 over 100 epochs, adopting an early stopping mechanism with the patience of 40 epochs. The learning rate peaks at $2 \times 10^{-4}$, coupled with a linear scheduler with a 0.1 warm-up ratio, and the model trains with a dropout ratio of 0.1.

**GIN**  Graph Isomorphism Network GIN [91] is a randomly initialized model with 2D graph structures as input. Compared with graph convolutional networks (GCN) [40], GIN mainly differs in that within the neighboring nodes message aggregation process, GIN adds a weight to each node's self-looping, which is either trainable or pre-defined. In addition, GIN substitutes the one-layer feed-forward network within each GCN layer with a multi-layer perceptron (MLP). It has been proved in theory that these minor changes make GIN among the most powerful graph neural networks [91].

We use the Pytorch Geometric [17] to realize GIN. Our implementation contains 5 GIN layers with 128 hidden units and 0.1 dropout ratio. The model is trained with a batch size of 128 for 200 epochs with an early stopping tolerance of 50 epochs, at a constant learning rate of $10^{-4}$.

---

[7]We use the "rdkit2dnormalized" descriptor in `DescriptaStorus`, available at https://github.com/bp-kelley/descriptastorus.

[8]The repository can be accessed at https://github.com/shehzaidi/pre-training-via-denoising.

# D Uncertainty Quantification

## D.1 Method and Implementation Details

**Focal Loss** First proposed in [47], Focal Loss is designed to address the class imbalance issue for dense object detection in computer vision, where the number of negative samples (background) far exceeds the number of positive ones (objects). It is adopted for uncertainty estimation and model calibration later in [55]. The idea is to add a modulating factor to the standard cross-entropy loss to down-weight the contribution from easy examples and thus focus more on hard examples. In the binary classification case, it adds a modulating factor $|y_n - \hat{p}_n|^\gamma$ to the standard cross-entropy loss, where $y_n \in \{0, 1\}$ is the ground truth label, $\hat{p}_n \in (0, 1)$ is the predicted Sigmoid probability for the $n$-th example, and $\gamma \geq 0$ is a focusing parameter:

$$\mathcal{L}_{\text{focal}} = -\frac{1}{N} \sum_{n=1}^{N} \left[ y_n (1 - \hat{p}_n)^\gamma \log \hat{p}_n + (1 - y_n) \hat{p}_n^\gamma \log(1 - \hat{p}_n) \right]. \tag{1}$$

The focusing parameter $\gamma$ smoothly adjusts the rate at which easy examples are down-weighted. When $\gamma = 0$, Focal Loss is equivalent to the cross-entropy loss. As $\gamma$ increases, the effect of the modulating factor increases likewise. In our implementation, we take advantage of the realization in the "torchvision" library and use their default hyper-parameters for all experiments.[9]

**Bayes by Backprop** Bayes by Backprop (BBP) [6, 38] is an algorithm for training BNNs, where weights are not point estimates but distributions. The idea is to replace the deterministic network weights with Gaussian a porteriori learned from the data, which allows quantifying the uncertainty in the predictions by assembling the predictions from random networks sampled from the posterior distribution of the weights:

$$\boldsymbol{w}^{\text{MAP}} = \arg\max_{\boldsymbol{w}} \log p(\boldsymbol{w}|\mathcal{D}) = \arg\max_{\boldsymbol{w}} \left( \log p(\mathcal{D}|\boldsymbol{w}) + \log p(\boldsymbol{w}) \right), \tag{2}$$

where $p(\boldsymbol{w})$ is the prior distribution of the weights, which are also Gaussian in our realization.

However, the true posterior is generally intractable for neural networks and can only be approximated with variational inference $q(\boldsymbol{w}|\boldsymbol{\theta})$ [39], where $\boldsymbol{\theta}$ are variational parameters. In our case, the variational parameters consist of the mean and standard deviation of multivariate Gaussian distribution $\boldsymbol{\theta} = \{\boldsymbol{\mu}, \boldsymbol{\rho}\}$. The learning then involves finding the $\boldsymbol{\theta}$ that minimizes the Kullback-Leibler (KL) divergence between the true posterior and the variational distribution. The loss can be written as

$$\mathcal{L} = \text{KL} \left[ q(\boldsymbol{w}|\boldsymbol{\theta}) || p(\boldsymbol{w}) \right] - \mathbb{E}_{q(\boldsymbol{w}|\boldsymbol{\theta})}[\log p(\mathcal{D}|\boldsymbol{w})]. \tag{3}$$

For each training step $i$, we first draw a sample from a porteriori $\boldsymbol{w}_i \sim q(\boldsymbol{w}|\boldsymbol{\theta})$ and then compute the Monte Carlo estimation of the loss:

$$\mathcal{L}_i \approx \log q(\boldsymbol{w}_i|\boldsymbol{\theta}) - \log p(\boldsymbol{w}_i) - \log p(\mathcal{D}|\boldsymbol{w}_i). \tag{4}$$

During backpropagation, the gradient can be pushed through the sampling process with the reparameterization trick [39]. Specifically, we adopt the local reparameterization trick [38], which samples the pre-activation $\boldsymbol{a}_i$ directly from the distribution $q(\boldsymbol{a}_i|\boldsymbol{x}_i)$ parameterized by the input feature $\boldsymbol{x}_i$, instead of computing the is as $\boldsymbol{a}_i = \boldsymbol{w}_i \boldsymbol{x}_i$ using the sampled network weights. This has parameters that are deterministic functions of $\boldsymbol{x}_i$ and $\boldsymbol{\theta}$, which reduces the variance of the gradient estimates and can improve the efficiency of the learning process. Additionally, we only apply BBP to the last layer of the model to reduce computational costs and implementation difficulty for large pre-trained backbone models. During inference, we sample 30 networks from the posterior distribution, generating 30 sets of prediction results, and computing the mean of the predictions as the final output.

**SGLD** Stochastic gradient Langevin dynamics (SGLD) [86] combines the efficiency of stochastic gradient descent (SGD) with Langevin diffusion which introduces the ability to estimate parameter a posteriori. The update rule for SGLD is given by:

$$\Delta\boldsymbol{\theta} = -\frac{\eta_t}{2} \nabla \mathcal{L}(\boldsymbol{\theta}) + \sqrt{\eta_t} \boldsymbol{\epsilon}, \tag{5}$$

---

[9]https://pytorch.org/vision/main/generated/torchvision.ops.sigmoid_focal_loss.html.

where $\boldsymbol{\theta}$ is the network parameters, $\eta_t$ is the learning rate at time $t$, and $\boldsymbol{\epsilon}_t$ is the standard Gaussian noise. With learning rate $\boldsymbol{\theta}$ or weight gradient $\nabla\mathcal{L}(\boldsymbol{\theta})$ decreasing to small values, the update rule can transit from network optimization to posterior estimation. After sufficient optimization, subsequent samples of parameters can be seen as drawing from the posterior distribution of the model parameters given the data.

In our implementation, we follow the previous implementation and use a constant learning rate.[10] Similar to BBP, we only apply SGLD to the last layer of the model. We first train the model until its performance has stopped improving on the validation set, and then continue training it for another 30 epochs, resulting in 30 networks sampled from the Langevin dynamics. This generates 30 sets of prediction results during the test, and we compute the mean of the predictions as the final output.

**MC Dropout**   Compared to other Bayesian networks, Monte-Carlo Dropout (MC Dropout) [19] is a simple and efficient for modeling the network uncertainty. Dropout is proposed to prevent overfitting by randomly setting some neurons' outputs to zero during training [74]. At test time, dropout is deactivated and the weights are scaled down by the dropout rate to simulate the presence of all neurons. In contrast, MC Dropout proposes to keep dropout active during testing and make predictions with dropout turned on. By running several (*e.g.*, 30 in our experiments) forward passes with random dropout masks, we effectively obtain a Monte Carlo estimation of the predictive distribution.

**SWAG**   Stochastic Weight Averaging-Gaussian (SWAG) [51] is an extension of the Stochastic Weight Averaging (SWA) [35] method, a technique used for finding wider optima in the loss landscape and leads to improved generalization. SWAG fits a Gaussian distribution with a low rank plus diagonal covariance derived from the SGD iterates to approximate the posterior distribution over neural network weights. In SWAG, the model keeps tracking the weights encountered during the last $T$ steps of the stochastic gradient descent updates and computes the Gaussian mean and covariance:

$$\boldsymbol{\mu}_{\boldsymbol{\theta}} = \frac{1}{T}\sum_{t=1}^{T}\boldsymbol{\theta}_t;$$

$$\boldsymbol{\Sigma}_{\boldsymbol{\theta}} = \frac{1}{T}\sum_{t=1}^{T}(\boldsymbol{\theta}_t - \boldsymbol{\mu}_{\boldsymbol{\theta}})(\boldsymbol{\theta}_t - \boldsymbol{\mu}_{\boldsymbol{\theta}})^{\mathsf{T}}.$$

(6)

However, such computation requires storing all the model weights in the last $T$ steps, which is expensive for large models. To address this issue, [51] propose to approximate the covariance matrix with a low-rank plus diagonal matrix, and compute the mean and covariance iteratively. Specifically, at update step $t \in \{1, \dots, T\}$,

$$\bar{\boldsymbol{\theta}}_t = \frac{t\bar{\boldsymbol{\theta}}_{t-1} + \boldsymbol{\theta}_t}{t+1}; \quad \overline{\boldsymbol{\theta}^2}_t = \frac{t\overline{\boldsymbol{\theta}^2}_{t-1} + \boldsymbol{\theta}_t^2}{t+1}; \quad \widehat{D}_{:,t} = \boldsymbol{\theta}_t - \bar{\boldsymbol{\theta}}_t,$$

(7)

where $\bar{\boldsymbol{\theta}}_0$ is the best parameter weights found during training prior to the SWA session, $\overline{\boldsymbol{\theta}^2}_T - \bar{\boldsymbol{\theta}}_T^2$ is the covariance diagonal, and $\frac{1}{T-1}\widehat{D}\widehat{D}^{\mathsf{T}} \in \mathbb{R}^{d,d}$ is the low-rank approximation of the covariance matrix with $d$ being the parameter dimensionality. We can write the Gaussian weight posterior as

$$\boldsymbol{\theta}_{\text{SWAG}} \sim \mathcal{N}_{\text{SWAG}}\left(\bar{\boldsymbol{\theta}}_T, \frac{1}{2}\left(\overline{\boldsymbol{\theta}^2}_T - \bar{\boldsymbol{\theta}}_T^2 + \frac{1}{T-1}\widehat{D}\widehat{D}^{\mathsf{T}}\right)\right).$$

(8)

Notice that $\hat{D}$ has a different rank $K <= T$ in [51], but we set $K = T < d$ for simplicity. Uncertainty in SWAG is estimated by drawing weight samples from $\mathcal{N}_{\text{SWAG}}$ and running these through the network. We set $T = 20$, and draw 30 samples during the test in our experiments.

**Temperature Scaling**   Temperature Scaling [62, 28] is a simple and effective post-hoc method for calibrating the confidence of a neural network. Post-hoc methods calibrate the output probabilities of a pre-trained model without updating the fine-tuned network parameters. The core idea behind Temperature Scaling is to add a learnable parameter $h$ (the temperature) to adjust the output probability of the model. For a trained binary classification model, Temperature Scaling scales the logits $z$ with

$$z' = \frac{z}{h}$$

(9)

---

[10]https://github.com/JavierAntoran/Bayesian-Neural-Networks/.

Table 6: Computation resources required by different uncertainty estimation methods. We assume that we have already trained a deterministic backbone model for property prediction, and would like to build up a UQ method on top of it.

| UQ Method | Training Starting Checkpoint | Additional Cost[a] |
|---|---|---|
| Deterministic | - | 0 |
| Temperature | from fine-tuned backbone | $(T_{\text{infer}} + T_{\text{train-FFN}}) \times M_{\text{train-extra}}$ |
| Focal Loss | from scratch | $T_{\text{train}} \times M_{\text{train}}$ |
| MC Dropout | no training | $T_{\text{infer}} \times M_{\text{infer}}$ |
| SWAG | from fine-tuned backbone | $T_{\text{train}} \times M_{\text{train-extra}} + T_{\text{infer}} \times M_{\text{infer}}$ |
| BBP | from scratch | $T_{\text{train}} \times M_{\text{train}} + T_{\text{infer}} \times M_{\text{infer}}$ |
| SGLD | from scratch | $T_{\text{train}} \times (M_{\text{train}} + M_{\text{train-extra}}) + T_{\text{infer}} \times M_{\text{infer}}$ |
| Ensembles | from scratch | $T_{\text{train}} \times M_{\text{train}} \times (N_{\text{ensembles}} - 1)$ |

[a] $T_{\text{train}}$ and $T_{\text{infer}}$ are the time for one epoch of training and inference of the backbone model, respectively. In general, $T_{\text{train}} \gg T_{\text{infer}}$. $M_{\text{train}}$, $M_{\text{train-extra}}$, and $M_{\text{infer}}$ are the number of training epochs, additional training epochs, and inference epochs, respectively (appendix D.1). In general, $M_{\text{train}} \gg M_{\text{train-extra}}$. Different backbones and UQ methods have different $T$s and $M$s, but we use the same symbols nonetheless for simplicity. The result is a rough estimation without considering the additional inference time or the early stopping if a model is retrained.

before feeding $z'$ into the Sigmoid output activation function. The temperature $h$ is learned by minimizing the negative log-likelihood of the training data with other network parameters frozen. For multi-task classification such as Tox21, we assign an individual temperature to each task.

In precise terms, "Temperature Scaling" is introduced for multi-class classification utilizing SoftMax output activation [28]. For binary classification in our study, we implement Platt Scaling, excluding the bias term [62]. Nonetheless, we continue using "Temperature Scaling" for its widespread recognition.

**Deep Ensembles** Deep Ensembles [44] is a technique where multiple deep learning models are independently trained from different initializations, and their predictions are combined to make a final prediction. This approach exploits the idea that different models will make different types of errors, which can be reduced by averaging model predictions, leading to better overall performance and more robust uncertainty estimates [18]. Formally, given $M$ models in the ensemble, each with parameters $\boldsymbol{\theta}_m, m \in \{1, \ldots, M\}$, the ensemble prediction for an input data point $\boldsymbol{x}$ is given by:

$$\hat{y} = \frac{1}{M} \sum_{m=1}^{M} \hat{y}_m = \frac{1}{M} \sum_{m=1}^{M} f(\boldsymbol{x}; \boldsymbol{\theta}_m) \tag{10}$$

where $f$ represents the model architecture, and $\hat{y}_m$ is the post-activation result of the $m$-th model. We set $M = 3$ for QM8, QM9 and MUV to reduce computational consumption, and $M = 10$ for other scaffold split datasets. For random split, we uniformly use $M = 3$, as mentioned in appendix B.

For regression tasks, [44] aggregate the variances of different network predictions through parameterizing a Gaussian mixture model. In contrast, we take a simpler approach by computing the mean of the variances as the final output variance.

### D.2 Resource Analysis

Table 6 summarizes the additional training cost to apply each UQ method to a backbone model already fine-tuned for property prediction. From the table, we can see that post-hoc calibration and MC Dropout are the most efficient methods, while Deep Ensembles is undoubtedly the most expensive one, even though it performs the best most of the time. Several works aim to reduce the computational cost [87, 46], but we do not consider them in MUBEN and leave them to future works.

## E  Metrics

In this section, we elaborate on the metrics utilized in our research for both property prediction and uncertainty estimation. Consistent with recommendations from [90] and other previous works

[16, 96], we report **ROC-AUC** (area under the receiver operating characteristic curve) as the metric for classification prediction and **RMSE** (root-mean-square error) and **MAE** (mean absolute error) for regression. In quantifying classification uncertainty, we use **ECE**, **NLL**, and **Brier Score** as introduced in § 2. For regression, we compute the **Gaussian NLL** and regression **CE**. We compute the metrics for each task individually before calculating their macro average. Each reported metric is the average of 3 individual training-test runs with random seeds 0, 1, and 2. This differs from Deep Ensembles, which aggregate model predictions prior to metric calculation.

## E.1 Property Prediction Metrics

**ROC-AUC**    The receiver operating characteristic area under the curve (ROC-AUC) is widely used for binary classification tasks. The ROC curve plots the true positive rate (TPR), or *recall*, against the false positive rate (FPR) at various decision thresholds $t \in (0, 1)$. Given a set of true positives (TP), true negatives (TN), false positives (FP), and false negatives (FN), the TPR and FPR are computed as $\text{TPR} = \frac{\text{TP}}{\text{TP+FN}}$ and $\text{FPR} = \frac{\text{FP}}{\text{FP+TN}}$ respectively. The AUC signifies the likelihood of a randomly selected positive instance being ranked above a randomly chosen negative instance. This integral under the ROC curve is calculated as

$$\text{ROC-AUC} = \int_0^1 \text{TPR}(t) \frac{d}{dt} \text{FPR}(t) dt \tag{11}$$

and can be approximated using numerical methods.[11]

**RMSE**    For regression tasks, the root mean square error (RMSE) quantifies the average discrepancy between predicted values $\hat{y}_n \in \mathbb{R}$ and actual values $y_n \in \mathbb{R}$ for $N$ data points, given by:

$$\text{RMSE} = \sqrt{\frac{1}{N} \sum_{n=1}^{N} (\hat{y}_n - y_n)^2}. \tag{12}$$

**MAE**    The mean absolute error (MAE) is another regression metric, measuring the average absolute deviation between predicted and actual values. It is calculated as:

$$\text{MAE} = \frac{1}{N} \sum_{i=1}^{N} |\hat{y}_n - y_n|. \tag{13}$$

## E.2 Uncertainty Quantification Metrics

**NLL**    Negative log-likelihood (NLL) quantifies the mean deviation between predicted and actual values in logarithmic space and is commonly used to evaluate UQ performance [19, 44, 22].

For binary classification with Sigmoid output activation, NLL is given by:

$$\text{NLL} = -\frac{1}{N} \sum_{n=1}^{N} \left[ y_n \log \hat{p}_n + (1 - y_n) \log(1 - \hat{p}_n) \right], \tag{14}$$

where $y_n \in \{0, 1\}$ is the true label and $\hat{p}_n \in (0, 1) = p_{\boldsymbol{\theta}}(\hat{y}_n \mid \boldsymbol{x}_n)$ is the predicted probability.

For regression, the Gaussian NLL is calculated as:

$$\text{NLL} = -\frac{1}{N} \sum_{n=1}^{N} \log \mathcal{N}(y_n; \hat{y}_n, \hat{\sigma}_n) = \frac{1}{N} \sum_{n=1}^{N} \frac{1}{2} \left[ \log(2\pi\hat{\sigma}_n) + \frac{(y_n - \hat{y}_n)^2}{\hat{\sigma}_n} \right], \tag{15}$$

where $\hat{y}_n \in \mathbb{R}$ is the predicted mean and $\hat{\sigma}_n \in \mathbb{R}_+$ is the predicted variance regularized by the SoftPlus activation.

---

[11]We use scikit-learn's roc_auc_score function in our implementation.

**Brier Score** In classification, the Brier score (BS) is a proper scoring rule for measuring the accuracy of predicted probabilities. Similar to mean square error (MSE) in regression, it measures the mean squared difference between predicted probabilities and actual binary outcomes. For binary classification, the Brier score is calculated as

$$\text{Brier Score} = \frac{1}{N} \sum_{i=1}^{N} (\hat{p}_n - y_n)^2, \tag{16}$$

where $y_n \in \{0, 1\}$ and $\hat{p}_n \in (0, 1)$.

The Brier score provides an insightful decomposition as $\text{BS} = \text{Uncertainty} - \text{Resolution} + \text{Reliability}$ [56]. "Uncertainty" corresponds to the inherent variability over labels; "Resolution" quantifies the deviation of individual predictions from the average; and "Reliability" gauges the extent to which predicted probabilities align with the actual probabilities.

**ECE** Expected calibration error (ECE) measures the average difference between predicted probabilities and their corresponding empirical probabilities, whose calculation is presented in § 2 and rewritten here specifically for the binary classification case.

Binary ECE puts the predicted probabilities $\{\hat{p}_n\}_{n=1}^{N}$ into $S$ (which is set to $S = 15$ in our experiments) equal-width bins $B_s = \{n \in \{1, \ldots, N\} \mid \hat{p}_n \in (\rho_s, \rho_{s+1}]\}$ with $s \in \{1, \ldots, S\}$. For binary classification, ECE is calculated as

$$\text{ECE} = \sum_{s=1}^{S} \frac{|B_s|}{N} \left| \text{acc}(B_s) - \text{conf}(B_s) \right|;$$

$$\text{acc}(B_s) = \frac{1}{|B_s|} \sum_{n \in B_s} y_n; \qquad \text{conf}(B_s) = \frac{1}{|B_s|} \sum_{n \in B_s} \hat{p}_n. \tag{17}$$

Here, $\text{acc}(\cdot)$ is the predicion accuracy and $\text{conf}(\cdot)$ is the confidence level within each bin. Notice that $|\cdot|$ represents the size when used with a set, and the absolute value when its input is a real number.

**Regression CE** Regression calibration is less studied than classification calibration, and regression calibration error (CE), proposed in [43], is adopted in only a few works [72, 36]. Similar to ECE, CE measures the average difference between the observed confidence level and the expected confidence level within each bin. Specifically, a confidence level of $\alpha$ is measured as the fraction of data points whose true value falls within $\frac{1-\alpha}{2}$th quantile and $(1 - \frac{\alpha}{2})$th quantile of the predicted distribution. For example, 90% confidence level contains data points with true values falling within the 5%th quantile and 95%th quantile of the predicted distribution. With this established, we can calculate the confidence levels with the predicted quantile function, in our case, the cumulative distribution function (CDF) of the predicted Gaussian. Specifically,

$$\text{CE} = \frac{1}{S} \sum_{s=1}^{S} \left( \rho_s - \frac{1}{N} \left| \{n \in \{1, \ldots, N\} \mid \Phi(y_n; \hat{y}_n, \hat{\sigma}_n) \leq \rho_s\} \right| \right)^2 \tag{18}$$

where $\rho_s = \frac{s}{S}$ is the expected quantile, $\Phi$ is the Gaussian CDF:

$$\Phi(y_n; \hat{y}_n, \hat{\sigma}_n) = \frac{1}{2} \left( 1 + \text{erf} \left( \frac{y_n - \hat{y}_n}{\sqrt{2}\hat{\sigma}_n} \right) \right);$$

$$\text{erf}(z) = \frac{2}{\sqrt{\pi}} \int_0^z e^{-t^2} dt. \tag{19}$$

$\hat{p}_s = \frac{1}{N} \left| \{n \in \{1, \ldots, N\} \mid \Phi(y_n; \hat{y}_n, \hat{\sigma}_n) \leq \rho_s\} \right|$ is the empirical frequency of the predicted values falling into the $\rho_s$th quantile. This is slightly different from the confidence level-based calculation, where

$$\text{CE} = \frac{2}{S} \sum_{s=1}^{\frac{S}{2}} \left( (\rho_{S-s} - \rho_s) - (\hat{p}_{S-s} - \hat{p}_s) \right)^2, \tag{20}$$

but it reveals the same trend. We use $S = 20$ in our experiments.

# F    Additional Results

In this section, we supplement the discussions from § 4 with further results and analyses. Given the extensive volume of outcomes from our experiments, we have consolidated them into tables at the end of this article. For enhanced accessibility and offline analysis, we have also made these results available as CSV files in our GitHub repository.[12]

## F.1    More Visualization of Uncertainty Estimation Performance

Table 7 displays the average rankings of UQ methods, complemented by Figure 6 which offers a visual representation. As elaborated in § 4, both Deep Ensembles and MC Dropout generally enhance prediction and uncertainty estimation across tasks. However, the performance boost is more noticeable with Deep Ensembles compared to MC Dropout, while DC Dropout is much cheaper to apply. For classification tasks, Temperature Scaling offers moderate efficacy. Given the minimal computational cost of implementing Temperature Scaling, it emerges as a favorable option for calibration enhancement when necessary. In regression tasks, both BBP and SGLD stand out as reliable choices irrespective of the backbone model, particularly for uncertainty estimation. Nonetheless, they mandate a full re-training. Conversely, SWAG and Focal Loss are underwhelming and warrant reconsideration in selection processes while dealing with molecular property prediction.

## F.2    Complete Fine-Tuning Results

Tables 9–16 illustrate scores associated with each backbone-UQ pairing across the scaffold split classification datasets, while Tables 17–22 detail the scores regarding regression metrics. Scores are represented in a "mean (standard deviation)" structure, where both mean and standard deviation values are derived from 3 experiment runs conducted with random seeds 0, 1, and 2. As Deep Ensembles aggregates network predictions prior to metric calculation, standard deviation values are not generated and metrics are calculated only once.

Challenges are encountered during training on the notably imbalanced HIV and MUV dataset, depicted in Table 4. SGLD at times proves unsuccessful due to weak directional cues from the true labels, overwhelmed by the random noise embedded in the Langevin dynamics. With TorchMD-NET backbone, SGLD even fails training and constantly produces NaN values regardless of the hyperparameters we choose. Conversely, the application of Focal Loss demonstrates effective performance on the MUV dataset, aligning with assertions in [47]. ECE also may fail to compute on these datasets as predicted probabilities $\hat{p}$ are excessively minimal, leading to unpopulated bins and subsequent "division by zero" errors.

In terms of the majority of datasets, the performance trends closely adhere to our previous discussions. However, an anomaly is observed with the BBBP and ClinTox datasets, where ChemBERTa produces higher performance than the more powerful models, deviating from prior findings [1, 96]. Conceptually, the ChemBERTa checkpoint we use is solely pre-trained on the MLM objective, which is unrelated to downstream tasks and should not include label information. Consequently, we hypothesize that either the SMILES strings are occasionally more expressive than other descriptors with respect to the physiological properties, or the checkpoint may have undergone fine-tuning prior to upload, resulting in potential contamination.

## F.3    Results on Random Split Datasets

Tables 23 to 30 illustrate the efficacy of primary backbone models across datasets with random divisions. Mirroring the discussion in appendix F.1, Table 8 and Figure 7 depict the average (reciprocal) rankings of UQ methods. For the most part, the comparative performance among the backbone models and UQ methods is consistent with those observed in scaffold split datasets. Nonetheless, certain exceptions emerge, especially concerning BBP, SGLD, and SWAG. On randomly split classification datasets, both BBP and SGLD continue to underperform, while a marked decline is noted in their regression NLL. This phenomenon might be attributed to the balanced nature of random split datasets, intensifying the predictive indications from actual labels. Given this environment, the noise from Langevin dynamics or Gaussian samples, which usually aids generalization, might

---

[12]Access them here: https://github.com/paper-submit-account/MUBen/tree/main/output.

Table 7: Averaged ranks of UQ methods across all datasets and backbone models. Lower value indicates better resuls. Temperature Scaling and Focal Loss are not applicable to regression tasks.

| | Classification | | | | Regression | | | |
|---|---|---|---|---|---|---|---|---|
| | ROC-AUC | ECE | NLL | BS | RMSE | MAE | NLL | CE |
| Deterministic | 16.12 | 15.41 | 17.03 | 16.53 | 12.75 | 12.21 | 16.12 | 15.58 |
| Temperature | 16.31 | 12.06 | 13.97 | 14.38 | - | - | - | - |
| Focal Loss | 17.12 | 22.84 | 20.41 | 21.97 | - | - | - | - |
| MC Dropout | 15.78 | 13.53 | 14.28 | 13.62 | 10.71 | 10.83 | 14.75 | 15.0 |
| SWAG | 15.47 | 17.94 | 19.62 | 17.69 | 14.0 | 13.92 | 18.58 | 17.21 |
| BBP | 18.97 | 20.28 | 17.0 | 18.88 | 14.04 | 14.42 | 8.29 | 7.67 |
| SGLD | 20.72 | 20.66 | 20.59 | 19.22 | 14.25 | 14.75 | 5.87 | 5.54 |
| Ensembles | 11.5 | 9.28 | 9.09 | 9.72 | 9.25 | 8.88 | 11.38 | 14.0 |

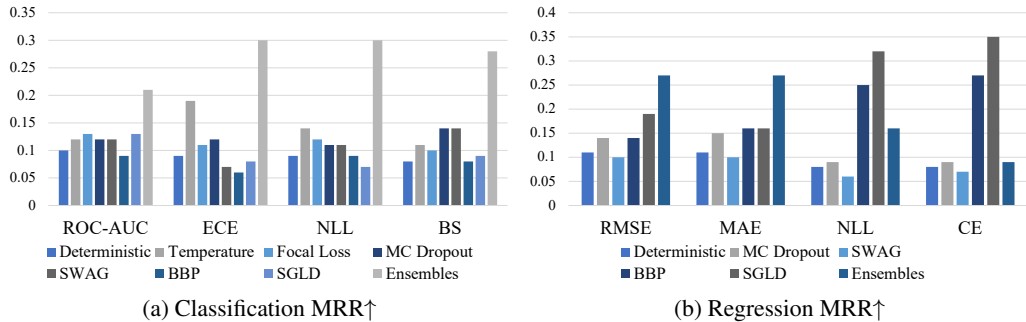

(a) Classification MRR↑  (b) Regression MRR↑

Figure 6: MRR of the UQ methods for each metric, each is macro-averaged from the reciprocal ranks of the results of all corresponding backbone models on all datasets.

become detrimental. This revelation highlights the capability of traditional Bayesian methods in aptly navigating OOD data (*i.e.*, addressing epistemic uncertainty), mainly through curbing overfitting. However, they may struggle when dealing with in-distribution attributes (*i.e.*, addressing aleatoric uncertainty) having relatively stable labels in the context of molecular property prediction. On the other hand, SWAG, another approximation of the Bayesian network that is almost useless on OOD data, delivers a much more impressive performance on random split datasets. Deep Ensembles, MC Dropout, and Temperature Scaling still emerge as the most reliable UQ methods, although Deep Ensembles no longer dominates the benchmark in the in-distribution case.

Upon comparing Figure 8 with Figure 2, it's evident that the relative positions of the backbone models remain largely unchanged for regression datasets across both splitting methods; however, this consistency falters for classification datasets. Notably, Uni-Mol displays a marked performance decline in the randomly split datasets, positioning it below the more straightforward and cost-efficient backbones, ChemBERTa and GROVER. While definitive explanations for this observed behavior remain elusive, we hypothesize that the molecular descriptor or the pre-training data inherent to ChemBERTa and GROVER could render them better equipped to adapt to Physiological and Biophysical features. Such potential advantages might be obscured by their model architectures' suboptimal generalization capabilities when scaffold splitting is employed.

### F.4 Results with Frozen Backbone Models

Analysis of Tables 31 through 38 reveals that when the network parameters of the backbone models are fixed, all models underperform significantly. Among the three primary backbone models, Uni-Mol experiences the most drastic decline, emerging as the least effective in the majority of evaluations. These findings highlight that these backbone models are not inherently optimized for mere feature extraction. The fine-tuning process is pivotal for ensuring optimal model performance, irrespective of the use of UQ methods.

Table 8: Averaged ranks of UQ methods across all randomly split datasets.

| | Classification | | | | Regression | | | |
|---|---|---|---|---|---|---|---|---|
| | ROC-AUC | ECE | NLL | BS | RMSE | MAE | NLL | CE |
| Deterministic | 16.44 | 16.31 | 17.06 | 16.5 | 13.25 | 12.69 | 12.62 | 15.88 |
| Temperature | 16.88 | 14.0 | 13.06 | 14.5 | - | - | - | - |
| Focal Loss | 17.31 | 28.75 | 23.62 | 26.56 | - | - | - | - |
| MC Dropout | 14.88 | 9.56 | 12.88 | 13.75 | 10.81 | 10.94 | 10.75 | 15.56 |
| SWAG | 13.81 | 15.0 | 15.06 | 14.0 | 10.44 | 10.31 | 11.19 | 16.44 |
| BBP | 19.94 | 16.12 | 18.12 | 17.31 | 16.56 | 17.5 | 15.94 | 5.44 |
| SGLD | 18.44 | 18.81 | 19.0 | 17.56 | 14.12 | 14.12 | 14.19 | 6.69 |
| Ensembles | 14.31 | 13.44 | 13.19 | 11.81 | 9.81 | 9.44 | 10.31 | 15.0 |

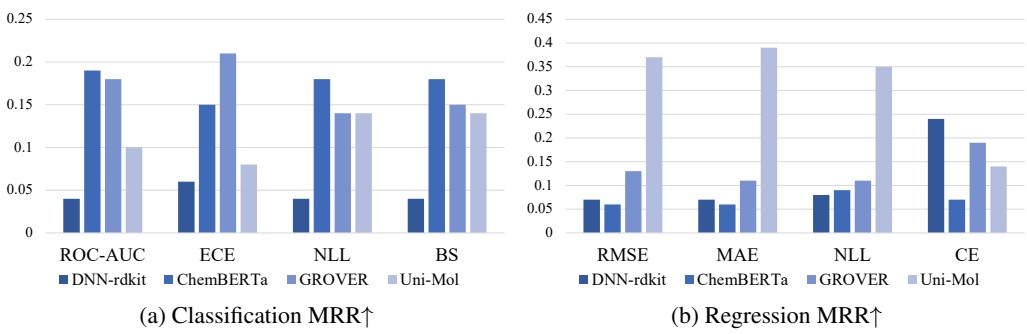

(a) Classification MRR↑  (b) Regression MRR↑

Figure 7: MRR of the UQ methods for each metric, each is macro-averaged from the reciprocal ranks of the results of all corresponding backbone models on all randomly split datasets.

(a) Classification MRR↑  (b) Regression MRR↑

Figure 8: MRR of the UQ methods for each backbone model, each is macro-averaged from the reciprocal ranks of the results of all corresponding UQ methods on all randomly split datasets.

Table 9: Test results on BACE in the format of "metric mean ± standard deviation".

|  |  | ROC-AUC | ECE | NLL | BS |
|---|---|---|---|---|---|
| DNN-rdkit | Deterministic | 0.8185 ± 0.0164 | 0.2827 ± 0.0155 | 0.8946 ± 0.0797 | 0.2810 ± 0.0125 |
|  | Temperature | 0.8185 ± 0.0164 | 0.2537 ± 0.0144 | 0.7412 ± 0.0472 | 0.2553 ± 0.0111 |
|  | Focal Loss | 0.8190 ± 0.0112 | 0.2543 ± 0.0011 | 0.6908 ± 0.0099 | 0.2488 ± 0.0027 |
|  | MC Dropout | 0.8168 ± 0.0121 | 0.2648 ± 0.0323 | 0.7667 ± 0.0446 | 0.2589 ± 0.0157 |
|  | SWAG | 0.8173 ± 0.0119 | 0.2728 ± 0.0068 | 0.8730 ± 0.0643 | 0.2730 ± 0.0085 |
|  | BBP | 0.8288 ± 0.0039 | 0.2711 ± 0.0193 | 0.7943 ± 0.0235 | 0.2535 ± 0.0075 |
|  | SGLD | 0.8181 ± 0.0160 | 0.2713 ± 0.0024 | 0.9023 ± 0.0416 | 0.2734 ± 0.0073 |
|  | Ensembles | 0.8207 ± - | 0.2550 ± - | 0.7985 ± - | 0.2557 ± - |
| ChemBERTa | Deterministic | 0.7223 ± 0.0089 | 0.2407 ± 0.0135 | 0.7567 ± 0.0203 | 0.2720 ± 0.0081 |
|  | Temperature | 0.7223 ± 0.0089 | 0.1758 ± 0.0059 | 0.6937 ± 0.0118 | 0.2498 ± 0.0052 |
|  | Focal Loss | 0.6985 ± 0.0062 | 0.2140 ± 0.0070 | 0.7322 ± 0.0051 | 0.2679 ± 0.0023 |
|  | MC Dropout | 0.7613 ± 0.0216 | 0.2391 ± 0.0262 | 0.7464 ± 0.0257 | 0.2673 ± 0.0070 |
|  | SWAG | 0.7580 ± 0.0139 | 0.2539 ± 0.0091 | 0.7707 ± 0.0374 | 0.2654 ± 0.0139 |
|  | BBP | 0.7372 ± 0.0233 | 0.2767 ± 0.0032 | 0.8325 ± 0.0190 | 0.2878 ± 0.0046 |
|  | SGLD | 0.7775 ± 0.0205 | 0.2492 ± 0.0217 | 0.7922 ± 0.0283 | 0.2588 ± 0.0069 |
|  | Ensembles | 0.7350 ± - | 0.2352 ± - | 0.7457 ± - | 0.2683 ± - |
| GROVER | Deterministic | 0.8308 ± 0.0012 | 0.2304 ± 0.0658 | 0.6921 ± 0.1114 | 0.2342 ± 0.0386 |
|  | Temperature | 0.8305 ± 0.0017 | 0.2083 ± 0.0521 | 0.6251 ± 0.0671 | 0.2184 ± 0.0283 |
|  | Focal Loss | 0.8509 ± 0.0072 | 0.2334 ± 0.0302 | 0.6397 ± 0.0300 | 0.2281 ± 0.0141 |
|  | MC Dropout | 0.8278 ± 0.0011 | 0.2366 ± 0.0629 | 0.6899 ± 0.1094 | 0.2368 ± 0.0394 |
|  | SWAG | 0.8604 ± 0.0048 | 0.2239 ± 0.0016 | 0.6656 ± 0.0169 | 0.2148 ± 0.0030 |
|  | BBP | 0.8322 ± 0.0099 | 0.2570 ± 0.0331 | 0.7001 ± 0.0583 | 0.2450 ± 0.0228 |
|  | SGLD | 0.8494 ± 0.0028 | 0.2281 ± 0.0066 | 0.6769 ± 0.0144 | 0.2204 ± 0.0038 |
|  | Ensembles | 0.8388 ± - | 0.2288 ± - | 0.6571 ± - | 0.2252 ± - |
| Uni-Mol | Deterministic | 0.8080 ± 0.0266 | 0.2419 ± 0.0409 | 0.7928 ± 0.1639 | 0.2517 ± 0.0335 |
|  | Temperature | 0.8087 ± 0.0277 | 0.2125 ± 0.0259 | 0.6782 ± 0.0662 | 0.2352 ± 0.0212 |
|  | Focal Loss | 0.8208 ± 0.0259 | 0.2445 ± 0.0287 | 0.6721 ± 0.0482 | 0.2406 ± 0.0198 |
|  | MC Dropout | 0.8268 ± 0.0269 | 0.2483 ± 0.0333 | 0.8065 ± 0.1383 | 0.2512 ± 0.0331 |
|  | SWAG | 0.8193 ± 0.0106 | 0.2677 ± 0.0075 | 0.9415 ± 0.0623 | 0.2625 ± 0.0058 |
|  | BBP | 0.8386 ± 0.0135 | 0.2673 ± 0.0473 | 0.8158 ± 0.1584 | 0.2588 ± 0.0375 |
|  | SGLD | 0.8699 ± 0.0118 | 0.2437 ± 0.0131 | 0.7354 ± 0.0502 | 0.2244 ± 0.0150 |
|  | Ensembles | 0.8505 ± - | 0.2176 ± - | 0.6301 ± - | 0.2184 ± - |
| TorchMD-NET | Deterministic | 0.7945 ± 0.0232 | 0.2193 ± 0.0374 | 1.1663 ± 0.6182 | 0.2272 ± 0.0201 |
|  | Temperature | 0.7945 ± 0.0231 | 0.1808 ± 0.0425 | 0.7151 ± 0.1720 | 0.2085 ± 0.0187 |
|  | Focal Loss | 0.7602 ± 0.0089 | 0.2141 ± 0.0249 | 0.7328 ± 0.0376 | 0.2420 ± 0.0200 |
|  | MC Dropout | 0.8018 ± 0.0260 | 0.2017 ± 0.0297 | 0.8454 ± 0.2767 | 0.2178 ± 0.0199 |
|  | SWAG | 0.7457 ± 0.0208 | 0.3287 ± 0.0164 | 1.9159 ± 0.3026 | 0.3172 ± 0.0122 |
|  | BBP | 0.7638 ± 0.0374 | 0.3130 ± 0.0457 | 2.8823 ± 1.2929 | 0.3035 ± 0.0425 |
|  | SGLD | 0.7787 ± 0.0173 | 0.2793 ± 0.0199 | 1.1648 ± 0.2135 | 0.2731 ± 0.0226 |
|  | Ensembles | 0.8169 ± - | 0.2220 ± - | 0.6703 ± - | 0.2197 ± - |
| GIN | Deterministic | 0.7312 ± 0.0152 | 0.4024 ± 0.0362 | 1.5262 ± 0.4443 | 0.3946 ± 0.0340 |
|  | Temperature | 0.7312 ± 0.0152 | 0.3494 ± 0.0392 | 1.0850 ± 0.1950 | 0.3482 ± 0.0285 |
|  | Focal Loss | 0.7272 ± 0.0374 | 0.3034 ± 0.1213 | 1.2375 ± 0.6434 | 0.3264 ± 0.0917 |
|  | MC Dropout | 0.7312 ± 0.0152 | 0.4024 ± 0.0362 | 1.5262 ± 0.4443 | 0.3946 ± 0.0340 |
|  | SWAG | 0.5554 ± 0.0075 | 0.1929 ± 0.0326 | 0.7817 ± 0.0442 | 0.2839 ± 0.0156 |
|  | BBP | 0.6775 ± 0.0533 | 0.2567 ± 0.0644 | 1.3036 ± 0.7463 | 0.2842 ± 0.0546 |
|  | SGLD | 0.6682 ± 0.0302 | 0.2271 ± 0.1028 | 0.8664 ± 0.2399 | 0.2887 ± 0.0569 |
|  | Ensembles | 0.7767 ± - | 0.2051 ± - | 0.7157 ± - | 0.2469 ± - |

Table 10: Test results on BBBP in the format of "metric mean ± standard deviation".

| | | ROC-AUC | ECE | NLL | BS |
|---|---|---|---|---|---|
| DNN-rdkit | Deterministic | 0.6721 ± 0.0129 | 0.3292 ± 0.0517 | 3.8147 ± 4.0782 | 0.3423 ± 0.0378 |
| | Temperature | 0.6719 ± 0.0129 | 0.2890 ± 0.0742 | 2.1338 ± 1.8963 | 0.3186 ± 0.0489 |
| | Focal Loss | 0.6693 ± 0.0047 | 0.1689 ± 0.1056 | 0.8925 ± 0.3424 | 0.2623 ± 0.0470 |
| | MC Dropout | 0.6669 ± 0.0151 | 0.3287 ± 0.0402 | 2.1699 ± 1.7478 | 0.3391 ± 0.0296 |
| | SWAG | 0.6777 ± 0.0136 | 0.3020 ± 0.0694 | 4.8415 ± 5.5407 | 0.3268 ± 0.0465 |
| | BBP | 0.6856 ± 0.0075 | 0.3392 ± 0.0506 | 1.8126 ± 0.6030 | 0.3463 ± 0.0357 |
| | SGLD | 0.6922 ± 0.0125 | 0.3134 ± 0.0487 | 1.7569 ± 1.0592 | 0.3247 ± 0.0396 |
| | Ensembles | 0.7018 ± - | 0.2615 ± - | 0.9043 ± - | 0.2938 ± - |
| ChemBERTa | Deterministic | 0.7407 ± 0.0101 | 0.2259 ± 0.0206 | 1.0306 ± 0.1002 | 0.2559 ± 0.0090 |
| | Temperature | 0.7407 ± 0.0101 | 0.2107 ± 0.0196 | 0.8539 ± 0.0266 | 0.2453 ± 0.0064 |
| | Focal Loss | 0.7410 ± 0.0122 | 0.1639 ± 0.0118 | 0.6995 ± 0.0246 | 0.2313 ± 0.0087 |
| | MC Dropout | 0.7451 ± 0.0067 | 0.2082 ± 0.0035 | 0.8814 ± 0.0292 | 0.2418 ± 0.0032 |
| | SWAG | 0.7408 ± 0.0112 | 0.2373 ± 0.0202 | 1.1435 ± 0.0810 | 0.2629 ± 0.0109 |
| | BBP | 0.7245 ± 0.0112 | 0.2415 ± 0.0293 | 0.9653 ± 0.1062 | 0.2652 ± 0.0120 |
| | SGLD | 0.7359 ± 0.0104 | 0.2398 ± 0.0256 | 1.0692 ± 0.1940 | 0.2580 ± 0.0056 |
| | Ensembles | 0.7399 ± - | 0.2290 ± - | 0.9672 ± - | 0.2506 ± - |
| GROVER | Deterministic | 0.7075 ± 0.0061 | 0.2902 ± 0.0111 | 0.9457 ± 0.0234 | 0.2998 ± 0.0060 |
| | Temperature | 0.7072 ± 0.0059 | 0.2685 ± 0.0147 | 0.8607 ± 0.0258 | 0.2865 ± 0.0069 |
| | Focal Loss | 0.7146 ± 0.0059 | 0.1340 ± 0.0177 | 0.6505 ± 0.0070 | 0.2284 ± 0.0040 |
| | MC Dropout | 0.7066 ± 0.0071 | 0.2901 ± 0.0147 | 0.9267 ± 0.0285 | 0.2996 ± 0.0071 |
| | SWAG | 0.7160 ± 0.0032 | 0.3314 ± 0.0032 | 1.1105 ± 0.0142 | 0.3251 ± 0.0025 |
| | BBP | 0.6968 ± 0.0152 | 0.2735 ± 0.0209 | 0.9146 ± 0.0326 | 0.2986 ± 0.0055 |
| | SGLD | 0.7266 ± 0.0037 | 0.3127 ± 0.0065 | 1.0187 ± 0.0202 | 0.3149 ± 0.0019 |
| | Ensembles | 0.7083 ± - | 0.2890 ± - | 0.9338 ± - | 0.3029 ± - |
| Uni-Mol | Deterministic | 0.7103 ± 0.0075 | 0.3311 ± 0.0506 | 1.4209 ± 0.3000 | 0.3354 ± 0.0397 |
| | Temperature | 0.7093 ± 0.0083 | 0.2922 ± 0.0418 | 1.0023 ± 0.1064 | 0.3084 ± 0.0315 |
| | Focal Loss | 0.7102 ± 0.0045 | 0.2031 ± 0.0816 | 0.7646 ± 0.0923 | 0.2632 ± 0.0275 |
| | MC Dropout | 0.7029 ± 0.0142 | 0.3328 ± 0.0483 | 1.4086 ± 0.2828 | 0.3382 ± 0.0372 |
| | SWAG | 0.7247 ± 0.0062 | 0.3547 ± 0.0136 | 1.8530 ± 0.0824 | 0.3456 ± 0.0078 |
| | BBP | 0.6902 ± 0.0286 | 0.3420 ± 0.0318 | 1.3548 ± 0.0713 | 0.3463 ± 0.0275 |
| | SGLD | 0.7147 ± 0.0118 | 0.3336 ± 0.0127 | 1.4174 ± 0.0475 | 0.3359 ± 0.0059 |
| | Ensembles | 0.7253 ± - | 0.3035 ± - | 1.0889 ± - | 0.3035 ± - |
| TorchMD-NET | Deterministic | 0.6753 ± 0.0060 | 0.3207 ± 0.0249 | 1.7352 ± 0.2714 | 0.3299 ± 0.0218 |
| | Temperature | 0.6753 ± 0.0060 | 0.2739 ± 0.0251 | 1.1382 ± 0.1017 | 0.2995 ± 0.0188 |
| | Focal Loss | 0.7060 ± 0.0206 | 0.1910 ± 0.0200 | 0.8688 ± 0.0498 | 0.2487 ± 0.0119 |
| | MC Dropout | 0.6767 ± 0.0057 | 0.3098 ± 0.0236 | 1.4864 ± 0.1962 | 0.3190 ± 0.0177 |
| | SWAG | 0.6720 ± 0.0131 | 0.3601 ± 0.0121 | 3.1976 ± 0.5027 | 0.3632 ± 0.0090 |
| | BBP | 0.6895 ± 0.0166 | 0.3019 ± 0.0598 | 3.1956 ± 1.7451 | 0.3159 ± 0.0343 |
| | SGLD | 0.6708 ± 0.0058 | 0.3629 ± 0.0083 | 2.3208 ± 0.5515 | 0.3617 ± 0.0068 |
| | Ensembles | 0.6844 ± - | 0.2700 ± - | 1.1018 ± - | 0.2934 ± - |
| GIN | Deterministic | 0.6282 ± 0.0025 | 0.2277 ± 0.0302 | 1.0566 ± 0.0130 | 0.2920 ± 0.0078 |
| | Temperature | 0.6281 ± 0.0025 | 0.1987 ± 0.0164 | 0.9092 ± 0.0166 | 0.2778 ± 0.0051 |
| | Focal Loss | 0.6234 ± 0.0081 | 0.1550 ± 0.0116 | 0.7581 ± 0.0143 | 0.2611 ± 0.0043 |
| | MC Dropout | 0.6282 ± 0.0025 | 0.2277 ± 0.0302 | 1.0566 ± 0.0130 | 0.2920 ± 0.0078 |
| | SWAG | 0.6067 ± 0.0131 | 0.3162 ± 0.0021 | 1.2223 ± 0.0235 | 0.3353 ± 0.0056 |
| | BBP | 0.6100 ± 0.0095 | 0.2738 ± 0.0056 | 1.1617 ± 0.0529 | 0.3119 ± 0.0058 |
| | SGLD | 0.5577 ± 0.0497 | 0.3493 ± 0.0217 | 1.4064 ± 0.1006 | 0.3681 ± 0.0218 |
| | Ensembles | 0.6298 ± - | 0.2112 ± - | 0.9696 ± - | 0.2802 ± - |

Table 11: Test results on ClinTox in the format of "metric mean ± standard deviation".

| | | ROC-AUC | ECE | NLL | BS |
|---|---|---|---|---|---|
| DNN-rdkit | Deterministic | 0.8301 ± 0.0226 | 0.0637 ± 0.0126 | 0.2719 ± 0.0481 | 0.0597 ± 0.0052 |
| | Temperature | 0.8300 ± 0.0224 | 0.0547 ± 0.0109 | 0.2371 ± 0.0059 | 0.0572 ± 0.0039 |
| | Focal Loss | 0.8507 ± 0.0165 | 0.0812 ± 0.0368 | 0.2284 ± 0.0305 | 0.0639 ± 0.0114 |
| | MC Dropout | 0.8194 ± 0.0217 | 0.0445 ± 0.0055 | 0.2149 ± 0.0307 | 0.0534 ± 0.0005 |
| | SWAG | 0.8290 ± 0.0163 | 0.0590 ± 0.0117 | 0.2964 ± 0.0617 | 0.0591 ± 0.0054 |
| | BBP | 0.7541 ± 0.0102 | 0.0574 ± 0.0121 | 0.2439 ± 0.0276 | 0.0609 ± 0.0046 |
| | SGLD | 0.8335 ± 0.0109 | 0.0567 ± 0.0125 | 0.2769 ± 0.0452 | 0.0553 ± 0.0063 |
| | Ensembles | 0.8154 ± - | 0.0283 ± - | 0.1860 ± - | 0.0484 ± - |
| ChemBERTa | Deterministic | 0.9856 ± 0.0028 | 0.0215 ± 0.0035 | 0.1206 ± 0.0318 | 0.0225 ± 0.0016 |
| | Temperature | 0.9856 ± 0.0028 | 0.0199 ± 0.0026 | 0.0964 ± 0.0147 | 0.0222 ± 0.0013 |
| | Focal Loss | 0.9832 ± 0.0027 | 0.0897 ± 0.0378 | 0.1367 ± 0.0350 | 0.0273 ± 0.0045 |
| | MC Dropout | 0.9848 ± 0.0021 | 0.0212 ± 0.0019 | 0.1083 ± 0.0247 | 0.0213 ± 0.0021 |
| | SWAG | 0.9850 ± 0.0031 | 0.0222 ± 0.0025 | 0.1327 ± 0.0317 | 0.0232 ± 0.0010 |
| | BBP | 0.9816 ± 0.0079 | 0.0325 ± 0.0212 | 0.0972 ± 0.0156 | 0.0226 ± 0.0024 |
| | SGLD | 0.9869 ± 0.0012 | 0.0183 ± 0.0006 | 0.0893 ± 0.0050 | 0.0214 ± 0.0005 |
| | Ensembles | 0.9812 ± - | 0.0157 ± - | 0.0810 ± - | 0.0208 ± - |
| GROVER | Deterministic | 0.9398 ± 0.0152 | 0.0635 ± 0.0114 | 0.1471 ± 0.0106 | 0.0367 ± 0.0031 |
| | Temperature | 0.9409 ± 0.0130 | 0.0449 ± 0.0055 | 0.1260 ± 0.0045 | 0.0333 ± 0.0034 |
| | Focal Loss | 0.9377 ± 0.0042 | 0.2158 ± 0.0034 | 0.2963 ± 0.0018 | 0.0731 ± 0.0011 |
| | MC Dropout | 0.9347 ± 0.0152 | 0.0706 ± 0.0122 | 0.1475 ± 0.0121 | 0.0366 ± 0.0036 |
| | SWAG | 0.9623 ± 0.0060 | 0.0460 ± 0.0117 | 0.1119 ± 0.0046 | 0.0280 ± 0.0022 |
| | BBP | 0.9085 ± 0.0109 | 0.0940 ± 0.0104 | 0.1976 ± 0.0136 | 0.0488 ± 0.0033 |
| | SGLD | 0.9507 ± 0.0058 | 0.0595 ± 0.0065 | 0.1287 ± 0.0063 | 0.0306 ± 0.0011 |
| | Ensembles | 0.9429 ± - | 0.0743 ± - | 0.1453 ± - | 0.0344 ± - |
| Uni-Mol | Deterministic | 0.8814 ± 0.0379 | 0.0522 ± 0.0120 | 0.1822 ± 0.0381 | 0.0484 ± 0.0124 |
| | Temperature | 0.8830 ± 0.0375 | 0.0417 ± 0.0158 | 0.1754 ± 0.0358 | 0.0477 ± 0.0121 |
| | Focal Loss | 0.8836 ± 0.0202 | 0.1651 ± 0.0556 | 0.2972 ± 0.0587 | 0.0827 ± 0.0207 |
| | MC Dropout | 0.8745 ± 0.0438 | 0.0560 ± 0.0131 | 0.1889 ± 0.0406 | 0.0521 ± 0.0138 |
| | SWAG | 0.9055 ± 0.0027 | 0.0311 ± 0.0034 | 0.1450 ± 0.0035 | 0.0337 ± 0.0009 |
| | BBP | 0.9257 ± 0.0287 | 0.0379 ± 0.0026 | 0.1468 ± 0.0047 | 0.0398 ± 0.0032 |
| | SGLD | 0.8598 ± 0.0209 | 0.0327 ± 0.0029 | 0.1619 ± 0.0059 | 0.0389 ± 0.0014 |
| | Ensembles | 0.9088 ± - | 0.0389 ± - | 0.1476 ± - | 0.0386 ± - |
| TorchMD-NET | Deterministic | 0.8938 ± 0.0178 | 0.0567 ± 0.0058 | 0.1905 ± 0.0083 | 0.0470 ± 0.0014 |
| | Temperature | 0.8938 ± 0.0178 | 0.0608 ± 0.0100 | 0.1869 ± 0.0076 | 0.0487 ± 0.0016 |
| | Focal Loss | 0.8952 ± 0.0195 | 0.1260 ± 0.0242 | 0.2582 ± 0.0218 | 0.0683 ± 0.0050 |
| | MC Dropout | 0.8898 ± 0.0192 | 0.0505 ± 0.0029 | 0.1711 ± 0.0054 | 0.0460 ± 0.0013 |
| | SWAG | 0.8995 ± 0.0185 | 0.0517 ± 0.0082 | 0.1969 ± 0.0065 | 0.0486 ± 0.0027 |
| | BBP | 0.8801 ± 0.0278 | 0.0667 ± 0.0139 | 0.2483 ± 0.0463 | 0.0610 ± 0.0077 |
| | SGLD | 0.8855 ± 0.0177 | 0.0475 ± 0.0045 | 0.1776 ± 0.0123 | 0.0475 ± 0.0045 |
| | Ensembles | 0.8944 ± - | 0.0569 ± - | 0.1886 ± - | 0.0485 ± - |
| GIN | Deterministic | 0.6965 ± 0.0184 | 0.1042 ± 0.0000 | 0.3398 ± 0.0260 | 0.0798 ± 0.0044 |
| | Temperature | 0.6965 ± 0.0184 | 0.0876 ± 0.0000 | 0.3595 ± 0.0346 | 0.0789 ± 0.0044 |
| | Focal Loss | 0.6777 ± 0.0168 | 0.2690 ± 0.0232 | 0.4610 ± 0.0314 | 0.1448 ± 0.0148 |
| | MC Dropout | 0.6965 ± 0.0184 | 0.1042 ± 0.0000 | 0.3398 ± 0.0260 | 0.0798 ± 0.0044 |
| | SWAG | 0.6552 ± 0.0197 | 0.1000 ± 0.0000 | 0.4573 ± 0.1485 | 0.0799 ± 0.0031 |
| | BBP | 0.6122 ± 0.0078 | - ± - | 0.5270 ± 0.1577 | 0.0888 ± 0.0131 |
| | SGLD | 0.6539 ± 0.0312 | 0.1256 ± 0.0000 | 0.4745 ± 0.1417 | 0.0827 ± 0.0046 |
| | Ensembles | 0.6872 ± - | 0.1170 ± - | 0.3222 ± - | 0.0868 ± - |

Table 12: Test results on Tox21 in the format of "metric mean ± standard deviation".

|  |  | ROC-AUC | ECE | NLL | BS |
|---|---|---|---|---|---|
| DNN-rdkit | Deterministic | 0.7386 ± 0.0061 | 0.0417 ± 0.0028 | 0.2771 ± 0.0024 | 0.0779 ± 0.0013 |
|  | Temperature | 0.7386 ± 0.0061 | 0.0342 ± 0.0027 | 0.2723 ± 0.0028 | 0.0773 ± 0.0013 |
|  | Focal Loss | 0.7374 ± 0.0038 | 0.1058 ± 0.0060 | 0.3161 ± 0.0078 | 0.0871 ± 0.0024 |
|  | MC Dropout | 0.7376 ± 0.0039 | 0.0356 ± 0.0016 | 0.2727 ± 0.0016 | 0.0763 ± 0.0013 |
|  | SWAG | 0.7364 ± 0.0045 | 0.0438 ± 0.0010 | 0.2793 ± 0.0037 | 0.0790 ± 0.0017 |
|  | BBP | 0.7243 ± 0.0036 | 0.0422 ± 0.0019 | 0.2847 ± 0.0010 | 0.0814 ± 0.0002 |
|  | SGLD | 0.7257 ± 0.0018 | 0.1192 ± 0.0332 | 0.3455 ± 0.0315 | 0.0978 ± 0.0100 |
|  | Ensembles | 0.7540 ± - | 0.0344 ± - | 0.2648 ± - | 0.0746 ± - |
| ChemBERTa | Deterministic | 0.7542 ± 0.0009 | 0.0571 ± 0.0020 | 0.2962 ± 0.0067 | 0.0812 ± 0.0009 |
|  | Temperature | 0.7542 ± 0.0009 | 0.0424 ± 0.0040 | 0.2744 ± 0.0027 | 0.0792 ± 0.0011 |
|  | Focal Loss | 0.7523 ± 0.0022 | 0.0969 ± 0.0012 | 0.3052 ± 0.0018 | 0.0845 ± 0.0004 |
|  | MC Dropout | 0.7641 ± 0.0032 | 0.0423 ± 0.0023 | 0.2697 ± 0.0032 | 0.0744 ± 0.0005 |
|  | SWAG | 0.7538 ± 0.0021 | 0.0592 ± 0.0022 | 0.3008 ± 0.0079 | 0.0818 ± 0.0011 |
|  | BBP | 0.7433 ± 0.0055 | 0.0459 ± 0.0028 | 0.2765 ± 0.0035 | 0.0780 ± 0.0014 |
|  | SGLD | 0.7475 ± 0.0036 | 0.0504 ± 0.0038 | 0.2784 ± 0.0030 | 0.0795 ± 0.0014 |
|  | Ensembles | 0.7681 ± - | 0.0440 ± - | 0.2679 ± - | 0.0750 ± - |
| GROVER | Deterministic | 0.7808 ± 0.0017 | 0.0358 ± 0.0038 | 0.2473 ± 0.0022 | 0.0694 ± 0.0005 |
|  | Temperature | 0.7810 ± 0.0016 | 0.0291 ± 0.0016 | 0.2439 ± 0.0008 | 0.0686 ± 0.0003 |
|  | Focal Loss | 0.7779 ± 0.0028 | 0.1148 ± 0.0102 | 0.3052 ± 0.0094 | 0.0811 ± 0.0028 |
|  | MC Dropout | 0.7817 ± 0.0018 | 0.0346 ± 0.0015 | 0.2455 ± 0.0016 | 0.0689 ± 0.0005 |
|  | SWAG | 0.7837 ± 0.0017 | 0.0359 ± 0.0009 | 0.2482 ± 0.0009 | 0.0689 ± 0.0001 |
|  | BBP | 0.7697 ± 0.0031 | 0.0438 ± 0.0017 | 0.2552 ± 0.0016 | 0.0711 ± 0.0003 |
|  | SGLD | 0.7635 ± 0.0009 | 0.0402 ± 0.0024 | 0.2558 ± 0.0010 | 0.0716 ± 0.0002 |
|  | Ensembles | 0.7876 ± - | 0.0316 ± - | 0.2411 ± - | 0.0675 ± - |
| Uni-Mol | Deterministic | 0.7895 ± 0.0017 | 0.0454 ± 0.0056 | 0.2601 ± 0.0072 | 0.0716 ± 0.0022 |
|  | Temperature | 0.7896 ± 0.0016 | 0.0346 ± 0.0027 | 0.2483 ± 0.0035 | 0.0704 ± 0.0014 |
|  | Focal Loss | 0.7904 ± 0.0040 | 0.0972 ± 0.0045 | 0.2899 ± 0.0009 | 0.0785 ± 0.0001 |
|  | MC Dropout | 0.7891 ± 0.0021 | 0.0480 ± 0.0066 | 0.2628 ± 0.0065 | 0.0726 ± 0.0021 |
|  | SWAG | 0.7842 ± 0.0048 | 0.0593 ± 0.0025 | 0.2994 ± 0.0054 | 0.0728 ± 0.0012 |
|  | BBP | 0.7932 ± 0.0043 | 0.0396 ± 0.0048 | 0.2520 ± 0.0039 | 0.0703 ± 0.0010 |
|  | SGLD | 0.7887 ± 0.0059 | 0.0433 ± 0.0023 | 0.2569 ± 0.0055 | 0.0684 ± 0.0014 |
|  | Ensembles | 0.8052 ± - | 0.0332 ± - | 0.2389 ± - | 0.0662 ± - |
| TorchMD-NET | Deterministic | 0.7342 ± 0.0143 | 0.0738 ± 0.0146 | 0.4523 ± 0.1083 | 0.0857 ± 0.0044 |
|  | Temperature | 0.7342 ± 0.0143 | 0.0515 ± 0.0047 | 0.2944 ± 0.0115 | 0.0801 ± 0.0014 |
|  | Focal Loss | 0.7403 ± 0.0032 | 0.0639 ± 0.0044 | 0.3102 ± 0.0140 | 0.0808 ± 0.0027 |
|  | MC Dropout | 0.7369 ± 0.0144 | 0.0685 ± 0.0131 | 0.3913 ± 0.0759 | 0.0837 ± 0.0033 |
|  | SWAG | 0.7347 ± 0.0093 | 0.0740 ± 0.0040 | 0.4567 ± 0.0576 | 0.0832 ± 0.0018 |
|  | BBP | 0.7459 ± 0.0067 | 0.0808 ± 0.0075 | 0.5355 ± 0.1830 | 0.0888 ± 0.0016 |
|  | SGLD | 0.7387 ± 0.0051 | 0.0427 ± 0.0019 | 0.2761 ± 0.0029 | 0.0757 ± 0.0004 |
|  | Ensembles | 0.7793 ± - | 0.0409 ± - | 0.2614 ± - | 0.0708 ± - |
| GIN | Deterministic | 0.6706 ± 0.0032 | 0.0789 ± 0.0172 | 0.3573 ± 0.0052 | 0.0916 ± 0.0070 |
|  | Temperature | 0.6706 ± 0.0032 | 0.0914 ± 0.0158 | 0.3393 ± 0.0150 | 0.0941 ± 0.0068 |
|  | Focal Loss | 0.6571 ± 0.0061 | 0.1040 ± 0.0073 | 0.3361 ± 0.0036 | 0.0930 ± 0.0014 |
|  | MC Dropout | 0.6706 ± 0.0032 | 0.0789 ± 0.0172 | 0.3573 ± 0.0052 | 0.0916 ± 0.0070 |
|  | SWAG | 0.6619 ± 0.0103 | 0.0671 ± 0.0032 | 0.3536 ± 0.0029 | 0.0867 ± 0.0026 |
|  | BBP | 0.6425 ± 0.0034 | 0.0851 ± 0.0084 | 0.3632 ± 0.0039 | 0.0949 ± 0.0033 |
|  | SGLD | 0.6190 ± 0.0048 | 0.0967 ± 0.0135 | 0.3817 ± 0.0275 | 0.0995 ± 0.0056 |
|  | Ensembles | 0.6829 ± - | 0.0634 ± - | 0.3268 ± - | 0.0840 ± - |

Table 13: Test results on ToxCast in the format of "metric mean $\pm$ standard deviation".

| | | ROC-AUC | ECE | NLL | BS |
|---|---|---|---|---|---|
| DNN-rdkit | Deterministic | 0.6222 ± 0.0042 | 0.1168 ± 0.0096 | 0.4436 ± 0.0146 | 0.1397 ± 0.0021 |
| | Temperature | 0.6220 ± 0.0046 | 0.1114 ± 0.0050 | 0.4882 ± 0.0170 | 0.1398 ± 0.0015 |
| | Focal Loss | 0.6289 ± 0.0077 | 0.1264 ± 0.0034 | 0.4389 ± 0.0084 | 0.1396 ± 0.0024 |
| | MC Dropout | 0.6248 ± 0.0030 | 0.1093 ± 0.0075 | 0.4319 ± 0.0096 | 0.1358 ± 0.0015 |
| | SWAG | 0.6207 ± 0.0073 | 0.1175 ± 0.0098 | 0.4440 ± 0.0150 | 0.1400 ± 0.0023 |
| | BBP | 0.6020 ± 0.0028 | 0.1443 ± 0.0060 | 0.4673 ± 0.0059 | 0.1510 ± 0.0020 |
| | SGLD | 0.5319 ± 0.0040 | 0.3054 ± 0.0034 | 0.6685 ± 0.0035 | 0.2378 ± 0.0017 |
| | Ensembles | 0.6486 ± - | 0.0900 ± - | 0.4008 ± - | 0.1292 ± - |
| ChemBERTa | Deterministic | 0.6554 ± 0.0037 | 0.1209 ± 0.0026 | 0.4313 ± 0.0054 | 0.1330 ± 0.0008 |
| | Temperature | 0.6540 ± 0.0042 | 0.1067 ± 0.0019 | 0.4817 ± 0.0261 | 0.1313 ± 0.0004 |
| | Focal Loss | 0.6442 ± 0.0170 | 0.1197 ± 0.0037 | 0.4243 ± 0.0153 | 0.1346 ± 0.0051 |
| | MC Dropout | 0.6624 ± 0.0024 | 0.1069 ± 0.0036 | 0.4070 ± 0.0052 | 0.1276 ± 0.0015 |
| | SWAG | 0.6556 ± 0.0033 | 0.1202 ± 0.0036 | 0.4305 ± 0.0064 | 0.1327 ± 0.0010 |
| | BBP | 0.5814 ± 0.0075 | 0.1276 ± 0.0015 | 0.4545 ± 0.0011 | 0.1469 ± 0.0003 |
| | SGLD | 0.5436 ± 0.0056 | 0.2238 ± 0.0019 | 0.5602 ± 0.0022 | 0.1881 ± 0.0011 |
| | Ensembles | 0.6733 ± - | 0.1037 ± - | 0.3986 ± - | 0.1258 ± - |
| GROVER | Deterministic | 0.6587 ± 0.0018 | 0.1043 ± 0.0006 | 0.4091 ± 0.0023 | 0.1298 ± 0.0004 |
| | Temperature | 0.6496 ± 0.0023 | 0.1424 ± 0.0026 | 0.4612 ± 0.0045 | 0.1424 ± 0.0013 |
| | Focal Loss | 0.6359 ± 0.0026 | 0.1221 ± 0.0005 | 0.4365 ± 0.0012 | 0.1383 ± 0.0004 |
| | MC Dropout | 0.6615 ± 0.0014 | 0.1009 ± 0.0010 | 0.4042 ± 0.0022 | 0.1288 ± 0.0004 |
| | SWAG | 0.6603 ± 0.0020 | 0.1060 ± 0.0007 | 0.4114 ± 0.0015 | 0.1301 ± 0.0001 |
| | BBP | 0.5995 ± 0.0026 | 0.1731 ± 0.0041 | 0.5090 ± 0.0040 | 0.1660 ± 0.0015 |
| | SGLD | 0.5542 ± 0.0069 | 0.2712 ± 0.0008 | 0.6194 ± 0.0006 | 0.2139 ± 0.0003 |
| | Ensembles | 0.6646 ± - | 0.1034 ± - | 0.4061 ± - | 0.1290 ± - |
| Uni-Mol | Deterministic | 0.6734 ± 0.0101 | 0.1020 ± 0.0023 | 0.3983 ± 0.0075 | 0.1274 ± 0.0025 |
| | Temperature | 0.7028 ± 0.0101 | 0.1456 ± 0.0045 | 0.4566 ± 0.0104 | 0.1355 ± 0.0029 |
| | Focal Loss | 0.6934 ± 0.0035 | 0.1227 ± 0.0011 | 0.4079 ± 0.0016 | 0.1284 ± 0.0003 |
| | MC Dropout | 0.6833 ± 0.0117 | 0.1074 ± 0.0090 | 0.4015 ± 0.0112 | 0.1274 ± 0.0025 |
| | SWAG | 0.6870 ± 0.0075 | 0.1085 ± 0.0034 | 0.4005 ± 0.0055 | 0.1271 ± 0.0019 |
| | BBP | 0.6273 ± 0.0066 | 0.1296 ± 0.0065 | 0.4522 ± 0.0068 | 0.1456 ± 0.0021 |
| | SGLD | 0.5700 ± 0.0005 | 0.1953 ± 0.0033 | 0.5207 ± 0.0029 | 0.1717 ± 0.0012 |
| | Ensembles | 0.6841 ± - | 0.0953 ± - | 0.3877 ± - | 0.1247 ± - |
| TorchMD-NET | Deterministic | 0.6361 ± 0.0001 | 0.1734 ± 0.0053 | 0.4870 ± 0.0164 | 0.1486 ± 0.0027 |
| | Temperature | 0.6384 ± 0.0020 | - ± - | 0.6053 ± 0.0984 | 0.1466 ± 0.0009 |
| | Focal Loss | 0.6443 ± 0.0073 | 0.1543 ± 0.0032 | 0.4337 ± 0.0011 | 0.1395 ± 0.0006 |
| | MC Dropout | 0.6339 ± 0.0042 | 0.1707 ± 0.0048 | 0.4778 ± 0.0141 | 0.1477 ± 0.0027 |
| | SWAG | 0.6457 ± 0.0064 | 0.1648 ± 0.0012 | 0.4617 ± 0.0025 | 0.1446 ± 0.0004 |
| | BBP | 0.6103 ± 0.0182 | 0.1471 ± 0.0049 | 0.4539 ± 0.0171 | 0.1434 ± 0.0026 |
| | SGLD | 0.5224 ± 0.0048 | 0.2976 ± 0.0020 | 0.6678 ± 0.0020 | 0.2374 ± 0.0010 |
| | Ensembles | 0.6540 ± - | 0.1546 ± - | 0.4424 ± - | 0.1396 ± - |
| GIN | Deterministic | 0.5667 ± 0.0032 | 0.1443 ± 0.0018 | 0.5142 ± 0.0048 | 0.1506 ± 0.0015 |
| | Temperature | 0.5669 ± 0.0028 | 0.1402 ± 0.0013 | 0.5333 ± 0.0192 | 0.1476 ± 0.0014 |
| | Focal Loss | 0.5483 ± 0.0028 | 0.1446 ± 0.0027 | 0.4615 ± 0.0009 | 0.1468 ± 0.0006 |
| | MC Dropout | 0.5667 ± 0.0032 | 0.1443 ± 0.0018 | 0.5142 ± 0.0048 | 0.1506 ± 0.0015 |
| | SWAG | 0.5633 ± 0.0059 | 0.1418 ± 0.0018 | 0.5125 ± 0.0052 | 0.1495 ± 0.0014 |
| | BBP | 0.5096 ± 0.0059 | 0.2243 ± 0.0007 | 0.6405 ± 0.0128 | 0.1935 ± 0.0014 |
| | SGLD | 0.4872 ± 0.0050 | 0.3166 ± 0.0019 | 0.8083 ± 0.0245 | 0.2668 ± 0.0047 |
| | Ensembles | 0.5752 ± - | 0.1381 ± - | 0.4835 ± - | 0.1477 ± - |

Table 14: Test results on SIDER in the format of "metric mean ± standard deviation".

| | | ROC-AUC | ECE | NLL | BS |
|---|---|---|---|---|---|
| DNN-rdkit | Deterministic | 0.5981 ± 0.0152 | 0.1188 ± 0.0344 | 0.5462 ± 0.0469 | 0.1760 ± 0.0124 |
| | Temperature | 0.5981 ± 0.0152 | 0.1033 ± 0.0271 | 0.5186 ± 0.0270 | 0.1709 ± 0.0094 |
| | Focal Loss | 0.5864 ± 0.0071 | 0.1663 ± 0.0217 | 0.5605 ± 0.0193 | 0.1893 ± 0.0075 |
| | MC Dropout | 0.5977 ± 0.0164 | 0.1071 ± 0.0271 | 0.5245 ± 0.0304 | 0.1705 ± 0.0084 |
| | SWAG | 0.5988 ± 0.0210 | 0.1206 ± 0.0333 | 0.5477 ± 0.0491 | 0.1764 ± 0.0125 |
| | BBP | 0.5738 ± 0.0272 | 0.1215 ± 0.0046 | 0.5241 ± 0.0077 | 0.1733 ± 0.0027 |
| | SGLD | 0.5895 ± 0.0087 | 0.1559 ± 0.0119 | 0.5653 ± 0.0119 | 0.1900 ± 0.0051 |
| | Ensembles | 0.6158 ± - | 0.0954 ± - | 0.4950 ± - | 0.1631 ± - |
| ChemBERTa | Deterministic | 0.6146 ± 0.0078 | 0.1483 ± 0.0102 | 0.5740 ± 0.0217 | 0.1804 ± 0.0055 |
| | Temperature | 0.6146 ± 0.0078 | 0.1275 ± 0.0104 | 0.5256 ± 0.0138 | 0.1721 ± 0.0043 |
| | Focal Loss | 0.6119 ± 0.0105 | 0.1498 ± 0.0136 | 0.5346 ± 0.0129 | 0.1777 ± 0.0046 |
| | MC Dropout | 0.6166 ± 0.0022 | 0.1136 ± 0.0143 | 0.5191 ± 0.0131 | 0.1672 ± 0.0037 |
| | SWAG | 0.6149 ± 0.0095 | 0.1527 ± 0.0090 | 0.5836 ± 0.0230 | 0.1825 ± 0.0056 |
| | BBP | 0.6067 ± 0.0108 | 0.1184 ± 0.0018 | 0.5127 ± 0.0043 | 0.1686 ± 0.0013 |
| | SGLD | 0.6127 ± 0.0110 | 0.1299 ± 0.0118 | 0.5255 ± 0.0157 | 0.1725 ± 0.0053 |
| | Ensembles | 0.6193 ± - | 0.1305 ± - | 0.5396 ± - | 0.1718 ± - |
| GROVER | Deterministic | 0.6213 ± 0.0027 | 0.1034 ± 0.0061 | 0.5023 ± 0.0071 | 0.1637 ± 0.0021 |
| | Temperature | 0.6209 ± 0.0030 | 0.0939 ± 0.0040 | 0.4935 ± 0.0046 | 0.1618 ± 0.0018 |
| | Focal Loss | 0.6059 ± 0.0165 | 0.1817 ± 0.0019 | 0.5666 ± 0.0027 | 0.1903 ± 0.0011 |
| | MC Dropout | 0.6248 ± 0.0060 | 0.0969 ± 0.0035 | 0.4942 ± 0.0057 | 0.1616 ± 0.0018 |
| | SWAG | 0.6133 ± 0.0024 | 0.1183 ± 0.0016 | 0.5280 ± 0.0066 | 0.1702 ± 0.0019 |
| | BBP | 0.5926 ± 0.0052 | 0.1347 ± 0.0014 | 0.5367 ± 0.0023 | 0.1762 ± 0.0009 |
| | SGLD | 0.5989 ± 0.0129 | 0.1012 ± 0.0040 | 0.5017 ± 0.0051 | 0.1637 ± 0.0019 |
| | Ensembles | 0.6237 ± - | 0.0878 ± - | 0.4917 ± - | 0.1604 ± - |
| Uni-Mol | Deterministic | 0.6214 ± 0.0073 | 0.1054 ± 0.0130 | 0.5047 ± 0.0184 | 0.1661 ± 0.0065 |
| | Temperature | 0.6210 ± 0.0075 | 0.0950 ± 0.0074 | 0.4937 ± 0.0101 | 0.1625 ± 0.0040 |
| | Focal Loss | 0.6200 ± 0.0118 | 0.1660 ± 0.0096 | 0.5471 ± 0.0090 | 0.1834 ± 0.0038 |
| | MC Dropout | 0.6125 ± 0.0060 | 0.1269 ± 0.0178 | 0.5324 ± 0.0270 | 0.1739 ± 0.0071 |
| | SWAG | 0.6246 ± 0.0090 | 0.1327 ± 0.0084 | 0.5397 ± 0.0125 | 0.1753 ± 0.0033 |
| | BBP | 0.6301 ± 0.0260 | 0.0991 ± 0.0130 | 0.4928 ± 0.0119 | 0.1612 ± 0.0049 |
| | SGLD | 0.6323 ± 0.0108 | 0.1286 ± 0.0093 | 0.5408 ± 0.0137 | 0.1725 ± 0.0030 |
| | Ensembles | 0.6378 ± - | 0.0872 ± - | 0.4836 ± - | 0.1585 ± - |
| TorchMD-NET | Deterministic | 0.5881 ± 0.0060 | 0.1305 ± 0.0239 | 0.5988 ± 0.0541 | 0.1789 ± 0.0072 |
| | Temperature | 0.5881 ± 0.0060 | 0.1142 ± 0.0109 | 0.5380 ± 0.0147 | 0.1721 ± 0.0038 |
| | Focal Loss | 0.5909 ± 0.0048 | 0.1515 ± 0.0034 | 0.5470 ± 0.0058 | 0.1848 ± 0.0031 |
| | MC Dropout | 0.5862 ± 0.0059 | 0.1230 ± 0.0205 | 0.5551 ± 0.0327 | 0.1756 ± 0.0061 |
| | SWAG | 0.6081 ± 0.0085 | 0.1281 ± 0.0152 | 0.5596 ± 0.0326 | 0.1751 ± 0.0059 |
| | BBP | 0.6072 ± 0.0150 | 0.1293 ± 0.0005 | 0.5619 ± 0.0059 | 0.1753 ± 0.0008 |
| | SGLD | 0.6026 ± 0.0089 | 0.1143 ± 0.0076 | 0.5162 ± 0.0054 | 0.1693 ± 0.0017 |
| | Ensembles | 0.6329 ± - | 0.1093 ± - | 0.5175 ± - | 0.1660 ± - |
| GIN | Deterministic | 0.6159 ± 0.0016 | 0.0900 ± 0.0020 | 0.5385 ± 0.0084 | 0.1627 ± 0.0009 |
| | Temperature | 0.6159 ± 0.0017 | 0.0916 ± 0.0031 | 0.5219 ± 0.0125 | 0.1625 ± 0.0009 |
| | Focal Loss | 0.5892 ± 0.0046 | 0.1863 ± 0.0011 | 0.5764 ± 0.0034 | 0.1953 ± 0.0013 |
| | MC Dropout | 0.6159 ± 0.0016 | 0.0900 ± 0.0020 | 0.5385 ± 0.0084 | 0.1627 ± 0.0009 |
| | SWAG | 0.6054 ± 0.0034 | 0.0959 ± 0.0042 | 0.5520 ± 0.0021 | 0.1642 ± 0.0011 |
| | BBP | 0.5499 ± 0.0132 | 0.1477 ± 0.0071 | 0.5666 ± 0.0079 | 0.1876 ± 0.0037 |
| | SGLD | 0.5792 ± 0.0074 | 0.1121 ± 0.0028 | 0.6447 ± 0.0421 | 0.1723 ± 0.0018 |
| | Ensembles | 0.6232 ± - | 0.0874 ± - | 0.5238 ± - | 0.1584 ± - |

Table 15: Test results on HIV in the format of "metric mean $\pm$ standard deviation".

| | | ROC-AUC | ECE | NLL | BS |
|---|---|---|---|---|---|
| DNN-rdkit | Deterministic | 0.7345 $\pm$ 0.0065 | 0.0077 $\pm$ 0.0010 | 0.1236 $\pm$ 0.0016 | 0.0276 $\pm$ 0.0006 |
| | Temperature | 0.7345 $\pm$ 0.0065 | 0.0136 $\pm$ 0.0005 | 0.1277 $\pm$ 0.0021 | 0.0277 $\pm$ 0.0007 |
| | Focal Loss | 0.7338 $\pm$ 0.0078 | 0.1205 $\pm$ 0.0179 | 0.2059 $\pm$ 0.0179 | 0.0429 $\pm$ 0.0042 |
| | MC Dropout | 0.7365 $\pm$ 0.0088 | 0.0066 $\pm$ 0.0010 | 0.1219 $\pm$ 0.0016 | 0.0270 $\pm$ 0.0004 |
| | SWAG | 0.7439 $\pm$ 0.0014 | 0.0105 $\pm$ 0.0025 | 0.1240 $\pm$ 0.0013 | 0.0279 $\pm$ 0.0005 |
| | BBP | 0.7541 $\pm$ 0.0120 | 0.0181 $\pm$ 0.0065 | 0.1235 $\pm$ 0.0036 | 0.0277 $\pm$ 0.0011 |
| | SGLD | 0.7414 $\pm$ 0.0019 | 0.0160 $\pm$ 0.0048 | 0.1313 $\pm$ 0.0061 | 0.0281 $\pm$ 0.0007 |
| | Ensembles | 0.7613 $\pm$ - | 0.0074 $\pm$ - | 0.1160 $\pm$ - | 0.0260 $\pm$ - |
| ChemBERTa | Deterministic | 0.7681 $\pm$ 0.0061 | 0.0214 $\pm$ 0.0033 | 0.1247 $\pm$ 0.0058 | 0.0292 $\pm$ 0.0019 |
| | Temperature | 0.7681 $\pm$ 0.0061 | 0.0221 $\pm$ 0.0039 | 0.1259 $\pm$ 0.0067 | 0.0289 $\pm$ 0.0018 |
| | Focal Loss | 0.7509 $\pm$ 0.0106 | 0.0651 $\pm$ 0.0351 | 0.1599 $\pm$ 0.0205 | 0.0343 $\pm$ 0.0025 |
| | MC Dropout | 0.7717 $\pm$ 0.0042 | 0.0079 $\pm$ 0.0014 | 0.1105 $\pm$ 0.0011 | 0.0247 $\pm$ 0.0005 |
| | SWAG | 0.7474 $\pm$ 0.0123 | 0.0263 $\pm$ 0.0045 | 0.1384 $\pm$ 0.0063 | 0.0319 $\pm$ 0.0017 |
| | BBP | 0.7672 $\pm$ 0.0031 | 0.0217 $\pm$ 0.0043 | 0.1241 $\pm$ 0.0071 | 0.0289 $\pm$ 0.0027 |
| | SGLD | 0.7224 $\pm$ 0.0095 | 0.0374 $\pm$ 0.0011 | 0.1602 $\pm$ 0.0106 | 0.0376 $\pm$ 0.0002 |
| | Ensembles | 0.7885 $\pm$ - | 0.0205 $\pm$ - | 0.1161 $\pm$ - | 0.0267 $\pm$ - |
| GROVER | Deterministic | 0.7979 $\pm$ 0.0045 | 0.0078 $\pm$ 0.0032 | 0.1133 $\pm$ 0.0019 | 0.0263 $\pm$ 0.0005 |
| | Temperature | 0.7977 $\pm$ 0.0046 | 0.0120 $\pm$ 0.0011 | 0.1154 $\pm$ 0.0022 | 0.0265 $\pm$ 0.0004 |
| | Focal Loss | 0.7746 $\pm$ 0.0113 | 0.1286 $\pm$ 0.0223 | 0.2156 $\pm$ 0.0259 | 0.0444 $\pm$ 0.0066 |
| | MC Dropout | 0.8002 $\pm$ 0.0018 | 0.0077 $\pm$ 0.0028 | 0.1126 $\pm$ 0.0006 | 0.0261 $\pm$ 0.0001 |
| | SWAG | 0.8021 $\pm$ 0.0033 | 0.0090 $\pm$ 0.0019 | 0.1118 $\pm$ 0.0018 | 0.0258 $\pm$ 0.0005 |
| | BBP | 0.7928 $\pm$ 0.0116 | 0.0162 $\pm$ 0.0062 | 0.1151 $\pm$ 0.0034 | 0.0258 $\pm$ 0.0002 |
| | SGLD | 0.7420 $\pm$ 0.0131 | 0.0200 $\pm$ 0.0012 | 0.1231 $\pm$ 0.0001 | 0.0272 $\pm$ 0.0000 |
| | Ensembles | 0.8103 $\pm$ - | 0.0035 $\pm$ - | 0.1081 $\pm$ - | 0.0251 $\pm$ - |
| Uni-Mol | Deterministic | 0.7913 $\pm$ 0.0042 | 0.0189 $\pm$ 0.0063 | 0.1335 $\pm$ 0.0225 | 0.0273 $\pm$ 0.0014 |
| | Temperature | 0.7915 $\pm$ 0.0041 | 0.0151 $\pm$ 0.0017 | 0.1190 $\pm$ 0.0040 | 0.0269 $\pm$ 0.0009 |
| | Focal Loss | 0.7785 $\pm$ 0.0073 | 0.0713 $\pm$ 0.0251 | 0.1561 $\pm$ 0.0205 | 0.0316 $\pm$ 0.0032 |
| | MC Dropout | 0.7858 $\pm$ 0.0044 | 0.0229 $\pm$ 0.0091 | 0.1557 $\pm$ 0.0339 | 0.0294 $\pm$ 0.0033 |
| | SWAG | 0.7786 $\pm$ 0.0061 | 0.0264 $\pm$ 0.0015 | 0.1676 $\pm$ 0.0160 | 0.0286 $\pm$ 0.0011 |
| | BBP | 0.7781 $\pm$ 0.0128 | 0.0191 $\pm$ 0.0071 | 0.1428 $\pm$ 0.0327 | 0.0284 $\pm$ 0.0016 |
| | SGLD | 0.7777 $\pm$ 0.0121 | 0.0205 $\pm$ 0.0006 | 0.1373 $\pm$ 0.0087 | 0.0263 $\pm$ 0.0002 |
| | Ensembles | 0.8020 $\pm$ - | 0.0181 $\pm$ - | 0.1181 $\pm$ - | 0.0265 $\pm$ - |
| TorchMD-NET | Deterministic | 0.7221 $\pm$ 0.0133 | 0.0270 $\pm$ 0.0031 | 0.2340 $\pm$ 0.0668 | 0.0315 $\pm$ 0.0007 |
| | Temperature | 0.7221 $\pm$ 0.0133 | 0.0266 $\pm$ 0.0025 | 0.1532 $\pm$ 0.0024 | 0.0317 $\pm$ 0.0008 |
| | Focal Loss | 0.7350 $\pm$ 0.0170 | 0.0673 $\pm$ 0.0312 | 0.1585 $\pm$ 0.0212 | 0.0332 $\pm$ 0.0036 |
| | MC Dropout | 0.7225 $\pm$ 0.0098 | 0.0265 $\pm$ 0.0036 | 0.2191 $\pm$ 0.0590 | 0.0313 $\pm$ 0.0009 |
| | SWAG | 0.7555 $\pm$ 0.0132 | 0.0253 $\pm$ 0.0007 | 0.2110 $\pm$ 0.0153 | 0.0288 $\pm$ 0.0006 |
| | BBP | 0.7521 $\pm$ 0.0078 | 0.0195 $\pm$ 0.0114 | 0.1384 $\pm$ 0.0251 | 0.0290 $\pm$ 0.0036 |
| | SGLD | - $\pm$ - | - $\pm$ - | - $\pm$ - | - $\pm$ - |
| | Ensembles | 0.7370 $\pm$ - | 0.0218 $\pm$ - | 0.1537 $\pm$ - | 0.0284 $\pm$ - |
| GIN | Deterministic | 0.6893 $\pm$ 0.0214 | - $\pm$ - | 0.1715 $\pm$ 0.0040 | 0.0396 $\pm$ 0.0021 |
| | Temperature | 0.6893 $\pm$ 0.0214 | - $\pm$ - | 0.1708 $\pm$ 0.0032 | 0.0378 $\pm$ 0.0024 |
| | Focal Loss | 0.6703 $\pm$ 0.0204 | - $\pm$ - | 0.2283 $\pm$ 0.0151 | 0.0515 $\pm$ 0.0046 |
| | MC Dropout | 0.6893 $\pm$ 0.0214 | - $\pm$ - | 0.1715 $\pm$ 0.0040 | 0.0396 $\pm$ 0.0021 |
| | SWAG | 0.6863 $\pm$ 0.0305 | - $\pm$ - | 0.1496 $\pm$ 0.0091 | 0.0318 $\pm$ 0.0007 |
| | BBP | 0.6947 $\pm$ 0.0106 | - $\pm$ - | 0.1625 $\pm$ 0.0055 | 0.0360 $\pm$ 0.0021 |
| | SGLD | 0.6845 $\pm$ 0.0063 | - $\pm$ - | 0.1954 $\pm$ 0.0322 | 0.0362 $\pm$ 0.0026 |
| | Ensembles | 0.7209 $\pm$ - | - $\pm$ - | 0.1492 $\pm$ - | 0.0336 $\pm$ - |

Table 16: Test results on MUV in the format of "metric mean $\pm$ standard deviation".

| | | ROC-AUC | ECE | NLL | BS |
|---|---|---|---|---|---|
| DNN-rdkit | Deterministic | 0.6029 ± 0.0576 | 0.0015 ± 0.0003 | 0.0326 ± 0.0196 | 0.0026 ± 0.0001 |
| | Temperature | 0.6059 ± 0.0538 | - ± - | 0.0321 ± 0.0145 | 0.0027 ± 0.0003 |
| | Focal Loss | 0.6261 ± 0.0322 | 0.0780 ± 0.0033 | 0.0720 ± 0.0262 | 0.0076 ± 0.0032 |
| | MC Dropout | 0.6282 ± 0.0530 | 0.0015 ± 0.0003 | 0.0256 ± 0.0114 | 0.0025 ± 0.0000 |
| | SWAG | 0.5830 ± 0.0392 | 0.0025 ± 0.0006 | 0.0393 ± 0.0254 | 0.0027 ± 0.0001 |
| | BBP | 0.5974 ± 0.0287 | - ± - | 0.0251 ± 0.0068 | 0.0027 ± 0.0001 |
| | SGLD | 0.4955 ± 0.0509 | 0.2380 ± 0.0156 | 0.2798 ± 0.0201 | 0.0610 ± 0.0071 |
| | Ensembles | 0.6736 ± - | 0.0013 ± - | 0.0175 ± - | 0.0025 ± - |
| ChemBERTa | Deterministic | 0.6695 ± 0.0343 | 0.0030 ± 0.0001 | 0.0241 ± 0.0011 | 0.0029 ± 0.0000 |
| | Temperature | 0.6695 ± 0.0343 | 0.0026 ± 0.0003 | 0.0196 ± 0.0010 | 0.0029 ± 0.0000 |
| | Focal Loss | 0.7150 ± 0.0038 | 0.0208 ± 0.0119 | 0.0333 ± 0.0100 | 0.0040 ± 0.0007 |
| | MC Dropout | 0.7168 ± 0.0100 | 0.0026 ± 0.0003 | 0.0230 ± 0.0027 | 0.0026 ± 0.0001 |
| | SWAG | 0.6672 ± 0.0342 | 0.0031 ± 0.0000 | 0.0261 ± 0.0015 | 0.0029 ± 0.0000 |
| | BBP | 0.5970 ± 0.0622 | 0.0026 ± 0.0010 | 0.0198 ± 0.0018 | 0.0030 ± 0.0004 |
| | SGLD | 0.5030 ± 0.0346 | 0.0074 ± 0.0003 | 0.0217 ± 0.0002 | 0.0026 ± 0.0000 |
| | Ensembles | 0.7224 ± - | 0.0025 ± - | 0.0197 ± - | 0.0027 ± - |
| GROVER | Deterministic | 0.7157 ± 0.0330 | 0.0016 ± 0.0004 | 0.0169 ± 0.0004 | 0.0025 ± 0.0000 |
| | Temperature | 0.7155 ± 0.0331 | 0.0014 ± 0.0001 | 0.0187 ± 0.0004 | 0.0025 ± 0.0000 |
| | Focal Loss | 0.6825 ± 0.0127 | 0.0869 ± 0.0033 | 0.0995 ± 0.0036 | 0.0101 ± 0.0006 |
| | MC Dropout | 0.7163 ± 0.0363 | 0.0016 ± 0.0004 | 0.0170 ± 0.0004 | 0.0025 ± 0.0000 |
| | SWAG | 0.7281 ± 0.0226 | 0.0014 ± 0.0003 | 0.0162 ± 0.0002 | 0.0025 ± 0.0000 |
| | BBP | 0.7490 ± 0.0068 | 0.0052 ± 0.0015 | 0.0192 ± 0.0011 | 0.0025 ± 0.0000 |
| | SGLD | 0.5532 ± 0.0255 | 0.0375 ± 0.0009 | 0.0488 ± 0.0008 | 0.0039 ± 0.0001 |
| | Ensembles | 0.7268 ± - | 0.0014 ± - | 0.0166 ± - | 0.0025 ± - |
| Uni-Mol | Deterministic | 0.7718 ± 0.0200 | 0.0024 ± 0.0002 | 0.0225 ± 0.0027 | 0.0025 ± 0.0000 |
| | Temperature | 0.7625 ± 0.0098 | 0.0017 ± 0.0000 | 0.0233 ± 0.0039 | 0.0025 ± 0.0001 |
| | Focal Loss | 0.8239 ± 0.0140 | 0.0250 ± 0.0126 | 0.0354 ± 0.0116 | 0.0036 ± 0.0006 |
| | MC Dropout | 0.7781 ± 0.0201 | 0.0026 ± 0.0001 | 0.0253 ± 0.0017 | 0.0025 ± 0.0000 |
| | SWAG | 0.7429 ± 0.0442 | 0.0028 ± 0.0001 | 0.0276 ± 0.0016 | 0.0027 ± 0.0001 |
| | BBP | 0.7394 ± 0.0140 | 0.0016 ± 0.0002 | 0.0169 ± 0.0002 | 0.0025 ± 0.0000 |
| | SGLD | 0.5901 ± 0.0200 | 0.0014 ± 0.0000 | 0.0178 ± 0.0000 | 0.0025 ± 0.0000 |
| | Ensembles | 0.7904 ± - | 0.0023 ± - | 0.0195 ± - | 0.0025 ± - |
| TorchMD-NET | Deterministic | 0.7043 ± 0.0027 | - ± - | 0.0553 ± 0.0092 | 0.0027 ± 0.0001 |
| | Temperature | 0.7073 ± 0.0016 | - ± - | 0.0318 ± 0.0059 | 0.0027 ± 0.0001 |
| | Focal Loss | 0.7227 ± 0.0117 | - ± - | 0.0225 ± 0.0034 | 0.0028 ± 0.0002 |
| | MC Dropout | 0.7003 ± 0.0057 | - ± - | 0.0455 ± 0.0074 | 0.0026 ± 0.0001 |
| | SWAG | 0.7285 ± 0.0094 | - ± - | 0.0457 ± 0.0011 | 0.0029 ± 0.0001 |
| | BBP | 0.7330 ± 0.0044 | - ± - | 0.0407 ± 0.0070 | 0.0029 ± 0.0002 |
| | SGLD | - ± - | - ± - | - ± - | - ± - |
| | Ensembles | 0.7045 ± - | - ± - | 0.0437 ± - | 0.0025 ± - |
| GIN | Deterministic | 0.6996 ± 0.0094 | 0.0024 ± 0.0006 | 0.0224 ± 0.0034 | 0.0028 ± 0.0001 |
| | Temperature | 0.6885 ± 0.0072 | - ± - | 0.0284 ± 0.0037 | 0.0028 ± 0.0002 |
| | Focal Loss | 0.6964 ± 0.0121 | 0.0146 ± 0.0053 | 0.0280 ± 0.0042 | 0.0037 ± 0.0004 |
| | MC Dropout | 0.6996 ± 0.0094 | 0.0024 ± 0.0006 | 0.0224 ± 0.0034 | 0.0028 ± 0.0001 |
| | SWAG | 0.6898 ± 0.0132 | 0.0021 ± 0.0003 | 0.0226 ± 0.0030 | 0.0026 ± 0.0000 |
| | BBP | 0.6425 ± 0.0114 | 0.0022 ± 0.0002 | 0.0229 ± 0.0009 | 0.0027 ± 0.0001 |
| | SGLD | 0.5553 ± 0.0110 | 0.0063 ± 0.0007 | 0.0400 ± 0.0058 | 0.0029 ± 0.0001 |
| | Ensembles | 0.7155 ± - | 0.0021 ± - | 0.0191 ± - | 0.0026 ± - |

Table 17: Test results on ESOL in the format of "metric mean ± standard deviation".

| | | RMSE | MAE | NLL | CE |
|---|---|---|---|---|---|
| DNN-rdkit | Deterministic | 0.9048 ± 0.0289 | 0.6747 ± 0.0059 | 1.0033 ± 0.3765 | 0.0465 ± 0.0052 |
| | MC Dropout | 0.9129 ± 0.0212 | 0.6853 ± 0.0049 | 1.0246 ± 0.3698 | 0.0495 ± 0.0051 |
| | SWAG | 0.9165 ± 0.0244 | 0.6827 ± 0.0074 | 0.9461 ± 0.3428 | 0.0459 ± 0.0048 |
| | BBP | 0.9557 ± 0.0298 | 0.7070 ± 0.0237 | 0.4306 ± 0.0325 | 0.0325 ± 0.0038 |
| | SGLD | 0.9630 ± 0.0425 | 0.7175 ± 0.0302 | 0.5101 ± 0.0380 | 0.0162 ± 0.0017 |
| | Ensembles | 0.8690 ± - | 0.6455 ± - | 0.5674 ± - | 0.0453 ± - |
| ChemBERTa | Deterministic | 1.0044 ± 0.0358 | 0.7989 ± 0.0216 | 2.2803 ± 0.1218 | 0.0500 ± 0.0007 |
| | MC Dropout | 0.9497 ± 0.0271 | 0.7620 ± 0.0176 | 1.4258 ± 0.1905 | 0.0481 ± 0.0023 |
| | SWAG | 0.9966 ± 0.0334 | 0.7940 ± 0.0206 | 2.5815 ± 0.2470 | 0.0513 ± 0.0013 |
| | BBP | 1.0125 ± 0.0113 | 0.8139 ± 0.0110 | 0.5990 ± 0.0499 | 0.0337 ± 0.0013 |
| | SGLD | 0.9978 ± 0.0340 | 0.8083 ± 0.0225 | 1.2727 ± 0.0711 | 0.0464 ± 0.0013 |
| | Ensembles | 0.9752 ± - | 0.7763 ± - | 1.4981 ± - | 0.0470 ± - |
| GROVER | Deterministic | 0.9355 ± 0.0252 | 0.7396 ± 0.0193 | 0.9888 ± 0.0395 | 0.0451 ± 0.0006 |
| | MC Dropout | 0.9437 ± 0.0246 | 0.7491 ± 0.0216 | 1.0065 ± 0.0351 | 0.0449 ± 0.0007 |
| | SWAG | 0.9443 ± 0.0219 | 0.7420 ± 0.0189 | 1.2996 ± 0.1883 | 0.0470 ± 0.0011 |
| | BBP | 0.9892 ± 0.0430 | 0.7821 ± 0.0327 | 0.5470 ± 0.0378 | 0.0153 ± 0.0028 |
| | SGLD | 0.9955 ± 0.0104 | 0.7942 ± 0.0153 | 0.4786 ± 0.0140 | 0.0243 ± 0.0008 |
| | Ensembles | 0.9110 ± - | 0.7209 ± - | 0.5854 ± - | 0.0394 ± - |
| Uni-Mol | Deterministic | 0.8530 ± 0.0224 | 0.6809 ± 0.0164 | 1.0739 ± 0.1136 | 0.0527 ± 0.0004 |
| | MC Dropout | 0.8367 ± 0.0237 | 0.6648 ± 0.0209 | 1.1098 ± 0.2625 | 0.0527 ± 0.0012 |
| | SWAG | 0.8644 ± 0.0097 | 0.6950 ± 0.0042 | 1.3868 ± 0.1522 | 0.0540 ± 0.0003 |
| | BBP | 0.8441 ± 0.0332 | 0.6719 ± 0.0279 | 0.3151 ± 0.0292 | 0.0329 ± 0.0015 |
| | SGLD | 0.8037 ± 0.0121 | 0.6444 ± 0.0112 | 0.2685 ± 0.0187 | 0.0343 ± 0.0018 |
| | Ensembles | 0.8181 ± - | 0.6516 ± - | 0.8372 ± - | 0.0512 ± - |
| TorchMD-NET | Deterministic | 0.8772 ± 0.0638 | 0.6688 ± 0.0517 | 0.6817 ± 0.1947 | 0.0409 ± 0.0059 |
| | MC Dropout | 0.8720 ± 0.0645 | 0.6660 ± 0.0510 | 0.6166 ± 0.1780 | 0.0402 ± 0.0063 |
| | SWAG | 0.8569 ± 0.0581 | 0.6642 ± 0.0401 | 0.4144 ± 0.1136 | 0.0372 ± 0.0053 |
| | BBP | 2.4684 ± 0.5232 | 1.9253 ± 0.3748 | 1.5283 ± 0.0753 | 0.0191 ± 0.0173 |
| | SGLD | 0.9691 ± 0.0579 | 0.7843 ± 0.0439 | 0.5284 ± 0.0243 | 0.0143 ± 0.0016 |
| | Ensembles | 1.0512 ± - | 0.8394 ± - | 0.9756 ± - | 0.0005 ± - |
| GIN | Deterministic | 1.5513 ± 0.0639 | 1.2268 ± 0.0618 | 1.0705 ± 0.1566 | 0.0122 ± 0.0072 |
| | MC Dropout | 1.5513 ± 0.0639 | 1.2268 ± 0.0618 | 1.0705 ± 0.1566 | 0.0122 ± 0.0072 |
| | SWAG | 1.8390 ± 0.2814 | 1.4650 ± 0.2128 | 1.4527 ± 0.2933 | 0.0166 ± 0.0089 |
| | BBP | 1.9791 ± 0.0920 | 1.5041 ± 0.0290 | 1.4189 ± 0.2794 | 0.0048 ± 0.0013 |
| | SGLD | 3.3157 ± 2.1193 | 2.7240 ± 1.8086 | 1.6595 ± 0.4615 | 0.0424 ± 0.0276 |
| | Ensembles | 1.3952 ± - | 1.0846 ± - | 0.9598 ± - | 0.0025 ± - |

Table 18: Test results on FreeSolv in the format of "metric mean ± standard deviation".

| | | RMSE | MAE | NLL | CE |
|---|---|---|---|---|---|
| DNN-rdkit | Deterministic | 2.2403 ± 0.1366 | 1.6768 ± 0.1007 | 1.3370 ± 0.1120 | 0.0272 ± 0.0026 |
| | MC Dropout | 2.2316 ± 0.1070 | 1.6469 ± 0.1122 | 1.3357 ± 0.1257 | 0.0247 ± 0.0015 |
| | SWAG | 2.4516 ± 0.1188 | 1.7853 ± 0.1478 | 1.8461 ± 0.2920 | 0.0358 ± 0.0043 |
| | BBP | 2.8456 ± 0.1416 | 2.1049 ± 0.1037 | 1.5285 ± 0.0469 | 0.0173 ± 0.0056 |
| | SGLD | 2.5503 ± 0.1010 | 1.8549 ± 0.1092 | 1.4260 ± 0.0095 | 0.0137 ± 0.0033 |
| | Ensembles | 1.9627 ± - | 1.4359 ± - | 1.0149 ± - | 0.0240 ± - |
| ChemBERTa | Deterministic | 2.5267 ± 0.0864 | 1.7760 ± 0.0318 | 1.7859 ± 0.1916 | 0.0364 ± 0.0034 |
| | MC Dropout | 2.5042 ± 0.0981 | 1.7804 ± 0.0456 | 1.7810 ± 0.1526 | 0.0359 ± 0.0033 |
| | SWAG | 2.6269 ± 0.1627 | 1.8562 ± 0.1055 | 2.2239 ± 0.2879 | 0.0426 ± 0.0003 |
| | BBP | 2.6340 ± 0.1368 | 1.9072 ± 0.1600 | 1.6143 ± 0.1930 | 0.0322 ± 0.0066 |
| | SGLD | 2.6564 ± 0.0357 | 1.8633 ± 0.0344 | 1.7627 ± 0.0809 | 0.0347 ± 0.0053 |
| | Ensembles | 2.3727 ± - | 1.6904 ± - | 1.5142 ± - | 0.0341 ± - |
| GROVER | Deterministic | 2.1352 ± 0.0193 | 1.4770 ± 0.0303 | 2.8244 ± 0.8307 | 0.0428 ± 0.0034 |
| | MC Dropout | 2.0981 ± 0.0069 | 1.4660 ± 0.0231 | 2.6575 ± 0.7926 | 0.0415 ± 0.0040 |
| | SWAG | 2.2321 ± 0.0902 | 1.5041 ± 0.0460 | 3.3021 ± 0.0530 | 0.0469 ± 0.0008 |
| | BBP | 2.6133 ± 0.0662 | 1.8288 ± 0.0843 | 1.4699 ± 0.0598 | 0.0126 ± 0.0014 |
| | SGLD | 2.1117 ± 0.0985 | 1.3784 ± 0.0459 | 1.2791 ± 0.1411 | 0.0238 ± 0.0020 |
| | Ensembles | 2.0238 ± - | 1.3539 ± - | 1.9474 ± - | 0.0393 ± - |
| Uni-Mol | Deterministic | 1.7871 ± 0.1265 | 1.2906 ± 0.0483 | 2.5450 ± 1.2435 | 0.0521 ± 0.0018 |
| | MC Dropout | 1.7585 ± 0.1322 | 1.2652 ± 0.0559 | 2.3425 ± 1.2327 | 0.0515 ± 0.0018 |
| | SWAG | 1.8524 ± 0.1449 | 1.3481 ± 0.1063 | 2.9865 ± 1.1583 | 0.0545 ± 0.0013 |
| | BBP | 1.7464 ± 0.0246 | 1.2424 ± 0.0049 | 0.9940 ± 0.0065 | 0.0289 ± 0.0027 |
| | SGLD | 1.7462 ± 0.0908 | 1.2963 ± 0.0610 | 1.0358 ± 0.0881 | 0.0296 ± 0.0048 |
| | Ensembles | 1.6716 ± - | 1.1780 ± - | 1.4306 ± - | 0.0504 ± - |
| TorchMD-NET | Deterministic | 2.6436 ± 0.0969 | 1.9379 ± 0.1236 | 1.9251 ± 0.2705 | 0.0219 ± 0.0060 |
| | MC Dropout | 2.6360 ± 0.0986 | 1.9475 ± 0.1212 | 1.8397 ± 0.2050 | 0.0212 ± 0.0052 |
| | SWAG | 2.6997 ± 0.1009 | 1.8466 ± 0.0863 | 2.5681 ± 0.4816 | 0.0399 ± 0.0026 |
| | BBP | 3.6709 ± 0.7927 | 2.7858 ± 0.6062 | 1.8484 ± 0.2682 | 0.0227 ± 0.0055 |
| | SGLD | 2.7769 ± 0.1054 | 1.9696 ± 0.1725 | 1.5583 ± 0.1031 | 0.0155 ± 0.0025 |
| | Ensembles | 2.9347 ± - | 2.0826 ± - | 1.6610 ± - | 0.0138 ± - |
| GIN | Deterministic | 2.2009 ± 0.2750 | 1.7292 ± 0.2873 | 1.3182 ± 0.0981 | 0.0142 ± 0.0039 |
| | MC Dropout | 2.2009 ± 0.2750 | 1.7292 ± 0.2873 | 1.3182 ± 0.0981 | 0.0142 ± 0.0039 |
| | SWAG | 2.3232 ± 0.0818 | 1.8154 ± 0.1109 | 1.5179 ± 0.1044 | 0.0198 ± 0.0106 |
| | BBP | 3.1290 ± 0.6024 | 2.4897 ± 0.5111 | 1.8594 ± 0.2348 | 0.0092 ± 0.0048 |
| | SGLD | 2.7563 ± 0.6927 | 2.2125 ± 0.6421 | 1.6046 ± 0.2001 | 0.0116 ± 0.0093 |
| | Ensembles | 2.0088 ± - | 1.5676 ± - | 1.5171 ± - | 0.0005 ± - |

Table 19: Test results on Lipophilicity in the format of "metric mean $\pm$ standard deviation".

| | | RMSE | MAE | NLL | CE |
|---|---|---|---|---|---|
| DNN-rdkit | Deterministic | 0.7575 $\pm$ 0.0116 | 0.5793 $\pm$ 0.0138 | 0.6154 $\pm$ 0.2016 | 0.0293 $\pm$ 0.0030 |
| | MC Dropout | 0.7559 $\pm$ 0.0110 | 0.5773 $\pm$ 0.0132 | 0.9071 $\pm$ 0.3721 | 0.0341 $\pm$ 0.0052 |
| | SWAG | 0.7572 $\pm$ 0.0101 | 0.5823 $\pm$ 0.0129 | 0.7191 $\pm$ 0.2267 | 0.0308 $\pm$ 0.0027 |
| | BBP | 0.7730 $\pm$ 0.0253 | 0.5938 $\pm$ 0.0245 | 0.7578 $\pm$ 0.1627 | 0.0305 $\pm$ 0.0046 |
| | SGLD | 0.7468 $\pm$ 0.0018 | 0.5743 $\pm$ 0.0016 | 0.2152 $\pm$ 0.0097 | 0.0090 $\pm$ 0.0009 |
| | Ensembles | 0.7172 $\pm$ - | 0.5490 $\pm$ - | 0.6165 $\pm$ - | 0.0322 $\pm$ - |
| ChemBERTa | Deterministic | 0.7553 $\pm$ 0.0074 | 0.5910 $\pm$ 0.0065 | 1.2368 $\pm$ 0.3994 | 0.0362 $\pm$ 0.0032 |
| | MC Dropout | 0.7142 $\pm$ 0.0063 | 0.5601 $\pm$ 0.0029 | 0.8178 $\pm$ 0.2701 | 0.0349 $\pm$ 0.0033 |
| | SWAG | 0.7672 $\pm$ 0.0101 | 0.5992 $\pm$ 0.0080 | 1.5809 $\pm$ 0.4168 | 0.0395 $\pm$ 0.0021 |
| | BBP | 0.7542 $\pm$ 0.0050 | 0.5869 $\pm$ 0.0087 | 0.4419 $\pm$ 0.0775 | 0.0279 $\pm$ 0.0023 |
| | SGLD | 0.7622 $\pm$ 0.0045 | 0.5982 $\pm$ 0.0070 | 0.8719 $\pm$ 0.1359 | 0.0355 $\pm$ 0.0020 |
| | Ensembles | 0.7367 $\pm$ - | 0.5763 $\pm$ - | 0.9756 $\pm$ - | 0.0360 $\pm$ - |
| GROVER | Deterministic | 0.6316 $\pm$ 0.0066 | 0.4747 $\pm$ 0.0040 | 2.1512 $\pm$ 0.2203 | 0.0478 $\pm$ 0.0009 |
| | MC Dropout | 0.6293 $\pm$ 0.0067 | 0.4740 $\pm$ 0.0028 | 2.0526 $\pm$ 0.2139 | 0.0476 $\pm$ 0.0009 |
| | SWAG | 0.6317 $\pm$ 0.0075 | 0.4750 $\pm$ 0.0041 | 2.3980 $\pm$ 0.3597 | 0.0485 $\pm$ 0.0007 |
| | BBP | 0.6481 $\pm$ 0.0063 | 0.5058 $\pm$ 0.0041 | 0.0789 $\pm$ 0.0206 | 0.0196 $\pm$ 0.0024 |
| | SGLD | 0.6360 $\pm$ 0.0041 | 0.4984 $\pm$ 0.0017 | 0.0544 $\pm$ 0.0199 | 0.0215 $\pm$ 0.0010 |
| | Ensembles | 0.6250 $\pm$ - | 0.4693 $\pm$ - | 1.6046 $\pm$ - | 0.0460 $\pm$ - |
| Uni-Mol | Deterministic | 0.6079 $\pm$ 0.0032 | 0.4509 $\pm$ 0.0044 | 0.8975 $\pm$ 0.5565 | 0.0425 $\pm$ 0.0061 |
| | MC Dropout | 0.5983 $\pm$ 0.0099 | 0.4438 $\pm$ 0.0117 | 1.3663 $\pm$ 1.2187 | 0.0440 $\pm$ 0.0084 |
| | SWAG | 0.6026 $\pm$ 0.0004 | 0.4476 $\pm$ 0.0022 | 1.0101 $\pm$ 0.1639 | 0.0453 $\pm$ 0.0012 |
| | BBP | 0.6044 $\pm$ 0.0016 | 0.4469 $\pm$ 0.0025 | 0.0679 $\pm$ 0.0218 | 0.0306 $\pm$ 0.0010 |
| | SGLD | 0.6040 $\pm$ 0.0031 | 0.4554 $\pm$ 0.0042 | 0.1565 $\pm$ 0.0573 | 0.0329 $\pm$ 0.0017 |
| | Ensembles | 0.5809 $\pm$ - | 0.4266 $\pm$ - | 0.6450 $\pm$ - | 0.0438 $\pm$ - |
| TorchMD-NET | Deterministic | 0.8235 $\pm$ 0.2218 | 0.6462 $\pm$ 0.1853 | 4.3336 $\pm$ 2.6567 | 0.0447 $\pm$ 0.0127 |
| | MC Dropout | 0.8887 $\pm$ 0.3142 | 0.7001 $\pm$ 0.2635 | 3.9796 $\pm$ 2.2502 | 0.0439 $\pm$ 0.0130 |
| | SWAG | 0.6819 $\pm$ 0.0271 | 0.5288 $\pm$ 0.0225 | 2.2934 $\pm$ 1.5338 | 0.0394 $\pm$ 0.0144 |
| | BBP | 2.8400 $\pm$ 2.2383 | 2.5087 $\pm$ 2.1497 | 1.6533 $\pm$ 0.9853 | 0.1008 $\pm$ 0.1219 |
| | SGLD | 0.8608 $\pm$ 0.0679 | 0.7002 $\pm$ 0.0502 | 0.9818 $\pm$ 0.4045 | 0.0183 $\pm$ 0.0128 |
| | Ensembles | 1.0313 $\pm$ - | 0.8196 $\pm$ - | 0.8619 $\pm$ - | 0.0195 $\pm$ - |
| GIN | Deterministic | 0.9286 $\pm$ 0.1113 | 0.7455 $\pm$ 0.1056 | 0.4542 $\pm$ 0.1096 | 0.0048 $\pm$ 0.0055 |
| | MC Dropout | 0.9286 $\pm$ 0.1113 | 0.7455 $\pm$ 0.1056 | 0.4542 $\pm$ 0.1096 | 0.0048 $\pm$ 0.0055 |
| | SWAG | 1.1105 $\pm$ 0.2448 | 0.9010 $\pm$ 0.2179 | 0.6805 $\pm$ 0.2447 | 0.0147 $\pm$ 0.0090 |
| | BBP | 1.0863 $\pm$ 0.0704 | 0.8606 $\pm$ 0.0492 | 0.6095 $\pm$ 0.0841 | 0.0018 $\pm$ 0.0010 |
| | SGLD | 1.3804 $\pm$ 0.4327 | 1.1187 $\pm$ 0.3763 | 0.9884 $\pm$ 0.4025 | 0.0229 $\pm$ 0.0137 |
| | Ensembles | 0.8071 $\pm$ - | 0.6515 $\pm$ - | 0.3241 $\pm$ - | 0.0020 $\pm$ - |

Table 20: Test results on QM7 in the format of "metric mean $\pm$ standard deviation".

| | | RMSE | MAE | NLL | CE |
|---|---|---|---|---|---|
| DNN-rdkit | Deterministic | 142.9279 $\pm$ 2.9477 | 95.7200 $\pm$ 3.2512 | 10.2976 $\pm$ 1.0848 | 0.0393 $\pm$ 0.0036 |
| | MC Dropout | 138.6817 $\pm$ 2.5873 | 90.7039 $\pm$ 3.1531 | 7.4627 $\pm$ 0.4138 | 0.0339 $\pm$ 0.0020 |
| | SWAG | 143.6922 $\pm$ 1.9420 | 97.2590 $\pm$ 1.9750 | 11.3649 $\pm$ 1.4063 | 0.0418 $\pm$ 0.0041 |
| | BBP | 141.5333 $\pm$ 4.8179 | 99.0430 $\pm$ 3.9287 | 6.8897 $\pm$ 0.8805 | 0.0330 $\pm$ 0.0064 |
| | SGLD | 139.8857 $\pm$ 5.4764 | 99.4873 $\pm$ 2.0159 | 5.4071 $\pm$ 0.0338 | 0.0098 $\pm$ 0.0019 |
| | Ensembles | 128.7182 $\pm$ - | 84.7269 $\pm$ - | 5.4568 $\pm$ - | 0.0281 $\pm$ - |
| ChemBERTa | Deterministic | 114.1954 $\pm$ 1.9473 | 66.6504 $\pm$ 3.6637 | 6.9619 $\pm$ 1.4000 | 0.0251 $\pm$ 0.0059 |
| | MC Dropout | 112.4931 $\pm$ 1.2013 | 64.7828 $\pm$ 3.4349 | 6.1942 $\pm$ 0.7998 | 0.0309 $\pm$ 0.0061 |
| | SWAG | 114.2023 $\pm$ 1.5302 | 68.0992 $\pm$ 3.5122 | 6.9684 $\pm$ 1.3236 | 0.0236 $\pm$ 0.0052 |
| | BBP | 116.2333 $\pm$ 1.8548 | 69.9783 $\pm$ 0.3732 | 5.5954 $\pm$ 0.2659 | 0.0262 $\pm$ 0.0034 |
| | SGLD | 116.9667 $\pm$ 0.3664 | 74.6714 $\pm$ 1.3652 | 5.0906 $\pm$ 0.0281 | 0.0147 $\pm$ 0.0019 |
| | Ensembles | 111.2764 $\pm$ - | 63.8752 $\pm$ - | 5.3896 $\pm$ - | 0.0224 $\pm$ - |
| GROVER | Deterministic | 139.5969 $\pm$ 3.1922 | 98.6424 $\pm$ 4.8571 | 5.4936 $\pm$ 0.0870 | 0.0121 $\pm$ 0.0012 |
| | MC Dropout | 136.1780 $\pm$ 2.9204 | 95.4940 $\pm$ 4.4117 | 5.4272 $\pm$ 0.0697 | 0.0107 $\pm$ 0.0014 |
| | SWAG | 142.6412 $\pm$ 1.3203 | 107.1310 $\pm$ 1.6574 | 5.7041 $\pm$ 0.1256 | 0.0153 $\pm$ 0.0018 |
| | BBP | 136.8576 $\pm$ 3.2036 | 99.8460 $\pm$ 2.2180 | 5.4555 $\pm$ 0.0135 | 0.0035 $\pm$ 0.0007 |
| | SGLD | 159.7489 $\pm$ 2.9839 | 126.6406 $\pm$ 2.8867 | 5.5587 $\pm$ 0.0308 | 0.0072 $\pm$ 0.0009 |
| | Ensembles | 131.9411 $\pm$ - | 89.4504 $\pm$ - | 5.3241 $\pm$ - | 0.0090 $\pm$ - |
| Uni-Mol | Deterministic | 103.9043 $\pm$ 1.3754 | 56.6287 $\pm$ 2.3360 | 4.6198 $\pm$ 0.2317 | 0.0338 $\pm$ 0.0065 |
| | MC Dropout | 103.5853 $\pm$ 1.3201 | 54.8976 $\pm$ 1.3061 | 4.5158 $\pm$ 0.2036 | 0.0386 $\pm$ 0.0019 |
| | SWAG | 107.8128 $\pm$ 0.1969 | 63.3542 $\pm$ 2.2114 | 5.5588 $\pm$ 0.6515 | 0.0412 $\pm$ 0.0026 |
| | BBP | 101.5479 $\pm$ 1.1382 | 51.9550 $\pm$ 0.7905 | 4.5153 $\pm$ 0.0535 | 0.0263 $\pm$ 0.0022 |
| | SGLD | 105.1573 $\pm$ 2.6264 | 57.0601 $\pm$ 4.5219 | 4.5642 $\pm$ 0.0453 | 0.0290 $\pm$ 0.0014 |
| | Ensembles | 101.1805 $\pm$ - | 52.0409 $\pm$ - | 4.3732 $\pm$ - | 0.0360 $\pm$ - |
| TorchMD-NET | Deterministic | 104.1467 $\pm$ 0.3144 | 49.6743 $\pm$ 0.7443 | 11.6606 $\pm$ 0.7198 | 0.0619 $\pm$ 0.0004 |
| | MC Dropout | 104.1265 $\pm$ 0.3415 | 49.8274 $\pm$ 0.5136 | 10.8641 $\pm$ 0.6233 | 0.0609 $\pm$ 0.0003 |
| | SWAG | 103.6753 $\pm$ 0.6387 | 50.1259 $\pm$ 1.4952 | 7.4456 $\pm$ 0.2718 | 0.0580 $\pm$ 0.0005 |
| | BBP | 104.4771 $\pm$ 0.4558 | 51.7268 $\pm$ 0.6870 | 9.5890 $\pm$ 0.4097 | 0.0568 $\pm$ 0.0002 |
| | SGLD | 112.7406 $\pm$ 3.9319 | 69.4583 $\pm$ 2.8874 | 5.1299 $\pm$ 0.0314 | 0.0212 $\pm$ 0.0022 |
| | Ensembles | 102.0535 $\pm$ - | 48.2234 $\pm$ - | 10.6292 $\pm$ - | 0.0618 $\pm$ - |
| GIN | Deterministic | 122.4303 $\pm$ 0.0690 | 76.2333 $\pm$ 1.0882 | 5.2490 $\pm$ 0.0500 | 0.0136 $\pm$ 0.0003 |
| | MC Dropout | 122.4303 $\pm$ 0.0690 | 76.2333 $\pm$ 1.0882 | 5.2490 $\pm$ 0.0500 | 0.0136 $\pm$ 0.0003 |
| | SWAG | 125.4810 $\pm$ 0.9210 | 83.3207 $\pm$ 1.3398 | 5.3423 $\pm$ 0.0619 | 0.0129 $\pm$ 0.0001 |
| | BBP | 125.2453 $\pm$ 0.6326 | 85.2772 $\pm$ 1.4557 | 5.3348 $\pm$ 0.0059 | 0.0077 $\pm$ 0.0014 |
| | SGLD | 132.1431 $\pm$ 6.9275 | 97.6270 $\pm$ 11.0966 | 5.3910 $\pm$ 0.0487 | 0.0126 $\pm$ 0.0011 |
| | Ensembles | 118.0550 $\pm$ - | 71.2196 $\pm$ - | 5.1922 $\pm$ - | 0.0105 $\pm$ - |

Table 21: Test results on QM8 in the format of "metric mean ± standard deviation".

| | | RMSE | MAE | NLL | CE |
|---|---|---|---|---|---|
| DNN-rdkit | Deterministic | 0.0356 ± 0.0004 | 0.0209 ± 0.0001 | -0.7811 ± 2.1740 | 0.0282 ± 0.0008 |
| | MC Dropout | 0.0352 ± 0.0002 | 0.0209 ± 0.0002 | -2.7716 ± 0.1922 | 0.0267 ± 0.0001 |
| | SWAG | 0.0360 ± 0.0002 | 0.0212 ± 0.0001 | -0.7143 ± 1.5808 | 0.0289 ± 0.0005 |
| | BBP | 0.0370 ± 0.0002 | 0.0214 ± 0.0001 | -0.9807 ± 0.5485 | 0.0269 ± 0.0004 |
| | SGLD | 0.0446 ± 0.0008 | 0.0294 ± 0.0005 | -2.7053 ± 0.0184 | 0.0121 ± 0.0001 |
| | Ensembles | 0.0345 ± - | 0.0202 ± - | -3.0118 ± - | 0.0279 ± - |
| ChemBERTa | Deterministic | 0.0377 ± 0.0001 | 0.0208 ± 0.0000 | 3.8214 ± 1.1599 | 0.0356 ± 0.0001 |
| | MC Dropout | 0.0370 ± 0.0002 | 0.0204 ± 0.0001 | -0.6884 ± 0.1974 | 0.0332 ± 0.0001 |
| | SWAG | 0.0379 ± 0.0002 | 0.0210 ± 0.0000 | 5.4548 ± 1.9400 | 0.0360 ± 0.0002 |
| | BBP | 0.0358 ± 0.0003 | 0.0206 ± 0.0001 | -1.5642 ± 0.0700 | 0.0316 ± 0.0005 |
| | SGLD | 0.0362 ± 0.0002 | 0.0212 ± 0.0004 | -3.1380 ± 0.0611 | 0.0169 ± 0.0007 |
| | Ensembles | 0.0373 ± - | 0.0205 ± - | 0.9180 ± - | 0.0349 ± - |
| GROVER | Deterministic | 0.0319 ± 0.0007 | 0.0174 ± 0.0003 | 0.0201 ± 1.4388 | 0.0386 ± 0.0020 |
| | MC Dropout | 0.0318 ± 0.0006 | 0.0174 ± 0.0003 | -0.8810 ± 1.1057 | 0.0381 ± 0.0021 |
| | SWAG | 0.0323 ± 0.0004 | 0.0176 ± 0.0003 | 0.6655 ± 0.9582 | 0.0409 ± 0.0013 |
| | BBP | 0.0324 ± 0.0004 | 0.0183 ± 0.0003 | -3.3037 ± 0.0260 | 0.0199 ± 0.0005 |
| | SGLD | 0.0340 ± 0.0007 | 0.0205 ± 0.0007 | -3.1826 ± 0.0378 | 0.0157 ± 0.0006 |
| | Ensembles | 0.0315 ± - | 0.0171 ± - | -1.1036 ± - | 0.0383 ± - |
| Uni-Mol | Deterministic | 0.0334 ± 0.0006 | 0.0172 ± 0.0000 | 2.7759 ± 2.9972 | 0.0432 ± 0.0020 |
| | MC Dropout | 0.0331 ± 0.0006 | 0.0171 ± 0.0001 | 0.8170 ± 2.8226 | 0.0412 ± 0.0023 |
| | SWAG | 0.0338 ± 0.0004 | 0.0173 ± 0.0001 | 3.5568 ± 2.5309 | 0.0440 ± 0.0013 |
| | BBP | 0.0309 ± 0.0007 | 0.0164 ± 0.0002 | -2.4581 ± 0.4546 | 0.0316 ± 0.0023 |
| | SGLD | 0.0300 ± 0.0003 | 0.0163 ± 0.0001 | -3.2399 ± 0.0908 | 0.0210 ± 0.0004 |
| | Ensembles | 0.0331 ± - | 0.0169 ± - | -0.0541 ± - | 0.0421 ± - |
| TorchMD-NET | Deterministic | 0.0324 ± 0.0006 | 0.0178 ± 0.0006 | 1.4661 ± 2.0676 | 0.0439 ± 0.0045 |
| | MC Dropout | 0.0324 ± 0.0007 | 0.0179 ± 0.0006 | 0.2637 ± 1.5600 | 0.0419 ± 0.0042 |
| | SWAG | 0.0319 ± 0.0005 | 0.0177 ± 0.0005 | -1.1581 ± 0.9675 | 0.0403 ± 0.0019 |
| | BBP | 0.0332 ± 0.0002 | 0.0176 ± 0.0001 | 4.4554 ± 0.7641 | 0.0469 ± 0.0007 |
| | SGLD | 0.0354 ± 0.0003 | 0.0217 ± 0.0001 | -2.9735 ± 0.0091 | 0.0091 ± 0.0003 |
| | Ensembles | 0.0315 ± - | 0.0173 ± - | -1.9286 ± - | 0.0419 ± - |
| GIN | Deterministic | 0.0344 ± 0.0004 | 0.0206 ± 0.0003 | -3.3711 ± 0.0959 | 0.0271 ± 0.0008 |
| | MC Dropout | 0.0344 ± 0.0004 | 0.0206 ± 0.0003 | -3.3711 ± 0.0959 | 0.0271 ± 0.0008 |
| | SWAG | 0.0352 ± 0.0005 | 0.0215 ± 0.0004 | -3.3091 ± 0.0569 | 0.0277 ± 0.0003 |
| | BBP | 0.0352 ± 0.0003 | 0.0220 ± 0.0001 | -3.2994 ± 0.0296 | 0.0238 ± 0.0007 |
| | SGLD | 0.0428 ± 0.0011 | 0.0290 ± 0.0009 | -2.8866 ± 0.0191 | 0.0189 ± 0.0016 |
| | Ensembles | 0.0341 ± - | 0.0205 ± - | -3.4906 ± - | 0.0271 ± - |

Table 22: Test results on QM9 in the format of "metric mean ± standard deviation".

| | | RMSE | MAE | NLL | CE |
|---|---|---|---|---|---|
| DNN-rdkit | Deterministic | 0.0151 ± 0.0001 | 0.0101 ± 0.0001 | -3.3798 ± 0.0592 | 0.0442 ± 0.0003 |
| | MC Dropout | 0.0148 ± 0.0001 | 0.0100 ± 0.0001 | -3.5269 ± 0.0258 | 0.0433 ± 0.0003 |
| | SWAG | 0.0152 ± 0.0002 | 0.0102 ± 0.0001 | -3.2845 ± 0.0814 | 0.0449 ± 0.0003 |
| | BBP | 0.0153 ± 0.0002 | 0.0103 ± 0.0001 | -3.3470 ± 0.0280 | 0.0445 ± 0.0004 |
| | SGLD | 0.0196 ± 0.0003 | 0.0144 ± 0.0002 | -3.3352 ± 0.0052 | 0.0070 ± 0.0003 |
| | Ensembles | 0.0143 ± - | 0.0096 ± - | -3.6023 ± - | 0.0436 ± - |
| ChemBERTa | Deterministic | 0.0146 ± 0.0004 | 0.0092 ± 0.0002 | -2.4104 ± 0.0698 | 0.0547 ± 0.0002 |
| | MC Dropout | 0.0141 ± 0.0002 | 0.0088 ± 0.0001 | -3.1501 ± 0.1094 | 0.0513 ± 0.0005 |
| | SWAG | 0.0148 ± 0.0003 | 0.0092 ± 0.0002 | -2.1703 ± 0.0205 | 0.0553 ± 0.0001 |
| | BBP | 0.0144 ± 0.0001 | 0.0093 ± 0.0001 | -2.5932 ± 0.0665 | 0.0540 ± 0.0002 |
| | SGLD | 0.0153 ± 0.0001 | 0.0101 ± 0.0002 | -3.7586 ± 0.0106 | 0.0338 ± 0.0007 |
| | Ensembles | 0.0140 ± - | 0.0087 ± - | -2.8766 ± - | 0.0543 ± - |
| GROVER | Deterministic | 0.0115 ± 0.0000 | 0.0068 ± 0.0000 | -0.7874 ± 0.5323 | 0.0621 ± 0.0008 |
| | MC Dropout | 0.0114 ± 0.0000 | 0.0068 ± 0.0000 | -1.1002 ± 0.5003 | 0.0616 ± 0.0008 |
| | SWAG | 0.0116 ± 0.0000 | 0.0068 ± 0.0000 | -0.4778 ± 0.1923 | 0.0625 ± 0.0002 |
| | BBP | 0.0118 ± 0.0000 | 0.0070 ± 0.0000 | -1.8855 ± 0.1189 | 0.0591 ± 0.0004 |
| | SGLD | 0.0136 ± 0.0006 | 0.0088 ± 0.0003 | -3.7855 ± 0.0411 | 0.0291 ± 0.0017 |
| | Ensembles | 0.0114 ± - | 0.0067 ± - | -1.0286 ± - | 0.0620 ± - |
| Uni-Mol | Deterministic | 0.0096 ± 0.0001 | 0.0054 ± 0.0001 | 0.0143 ± 0.4195 | 0.0664 ± 0.0003 |
| | MC Dropout | 0.0096 ± 0.0001 | 0.0054 ± 0.0001 | -0.2513 ± 0.3961 | 0.0661 ± 0.0003 |
| | SWAG | 0.0097 ± 0.0000 | 0.0054 ± 0.0000 | -0.4624 ± 0.0619 | 0.0660 ± 0.0001 |
| | BBP | 0.0095 ± 0.0002 | 0.0054 ± 0.0000 | -2.9595 ± 0.1250 | 0.0618 ± 0.0003 |
| | SGLD | 0.0095 ± 0.0001 | 0.0055 ± 0.0001 | -4.2093 ± 0.0049 | 0.0459 ± 0.0002 |
| | Ensembles | 0.0095 ± - | 0.0053 ± - | -0.3192 ± - | 0.0663 ± - |
| TorchMD-NET | Deterministic | 0.0087 ± 0.0001 | 0.0048 ± 0.0000 | 3.6334 ± 1.0190 | 0.0690 ± 0.0003 |
| | MC Dropout | 0.0087 ± 0.0001 | 0.0048 ± 0.0000 | 3.1525 ± 1.0291 | 0.0687 ± 0.0003 |
| | SWAG | 0.0087 ± 0.0000 | 0.0048 ± 0.0000 | -0.7161 ± 0.1242 | 0.0670 ± 0.0001 |
| | BBP | 0.0083 ± 0.0002 | 0.0046 ± 0.0001 | 1.7970 ± 1.8061 | 0.0685 ± 0.0010 |
| | SGLD | 0.0113 ± 0.0006 | 0.0075 ± 0.0004 | -3.6560 ± 0.0469 | 0.0132 ± 0.0013 |
| | Ensembles | 0.0086 ± - | 0.0046 ± - | 2.2619 ± - | 0.0687 ± - |
| GIN | Deterministic | 0.0132 ± 0.0001 | 0.0084 ± 0.0001 | -3.4286 ± 0.0389 | 0.0497 ± 0.0003 |
| | MC Dropout | 0.0132 ± 0.0001 | 0.0084 ± 0.0001 | -3.4286 ± 0.0389 | 0.0497 ± 0.0003 |
| | SWAG | 0.0136 ± 0.0000 | 0.0088 ± 0.0001 | -3.3355 ± 0.0589 | 0.0495 ± 0.0005 |
| | BBP | 0.0131 ± 0.0000 | 0.0084 ± 0.0000 | -3.3469 ± 0.0650 | 0.0499 ± 0.0003 |
| | SGLD | 0.0161 ± 0.0003 | 0.0115 ± 0.0004 | -3.5925 ± 0.0198 | 0.0211 ± 0.0031 |
| | Ensembles | 0.0130 ± - | 0.0081 ± - | -3.5207 ± - | 0.0500 ± - |

Table 23: Test results on the randomly split BACE dataset.

| | | ROC-AUC | ECE | NLL | BS |
|---|---|---|---|---|---|
| DNN-rdkit | Deterministic | 0.8850 ± 0.0220 | 0.1123 ± 0.0233 | 0.4820 ± 0.0723 | 0.1380 ± 0.0266 |
| | Temperature | 0.8850 ± 0.0220 | 0.1022 ± 0.0019 | 0.4475 ± 0.0624 | 0.1359 ± 0.0231 |
| | Focal Loss | 0.8884 ± 0.0167 | 0.1351 ± 0.0228 | 0.4623 ± 0.0239 | 0.1471 ± 0.0072 |
| | MC Dropout | 0.8910 ± 0.0245 | 0.0969 ± 0.0134 | 0.4328 ± 0.0594 | 0.1344 ± 0.0215 |
| | SWAG | 0.8898 ± 0.0220 | 0.1019 ± 0.0070 | 0.4346 ± 0.0525 | 0.1346 ± 0.0211 |
| | BBP | 0.8784 ± 0.0133 | 0.1101 ± 0.0256 | 0.4954 ± 0.0755 | 0.1414 ± 0.0156 |
| | SGLD | 0.8830 ± 0.0214 | 0.1119 ± 0.0294 | 0.4706 ± 0.0793 | 0.1360 ± 0.0244 |
| | Ensembles | 0.8872 ± 0.0223 | 0.0972 ± 0.0101 | 0.4572 ± 0.0712 | 0.1308 ± 0.0252 |
| ChemBERTa | Deterministic | 0.8835 ± 0.0044 | 0.1240 ± 0.0226 | 0.5182 ± 0.0812 | 0.1492 ± 0.0072 |
| | Temperature | 0.8835 ± 0.0044 | 0.1077 ± 0.0201 | 0.4566 ± 0.0274 | 0.1427 ± 0.0029 |
| | Focal Loss | 0.8884 ± 0.0096 | 0.1371 ± 0.0357 | 0.4490 ± 0.0328 | 0.1437 ± 0.0110 |
| | MC Dropout | 0.8825 ± 0.0124 | 0.0978 ± 0.0099 | 0.4544 ± 0.0082 | 0.1447 ± 0.0073 |
| | SWAG | 0.8868 ± 0.0059 | 0.1248 ± 0.0265 | 0.5286 ± 0.0964 | 0.1479 ± 0.0103 |
| | BBP | 0.8918 ± 0.0057 | 0.1230 ± 0.0362 | 0.5192 ± 0.0661 | 0.1468 ± 0.0117 |
| | SGLD | 0.8846 ± 0.0092 | 0.1362 ± 0.0046 | 0.5406 ± 0.0774 | 0.1438 ± 0.0073 |
| | Ensembles | 0.8959 ± 0.0058 | 0.1219 ± 0.0141 | 0.4715 ± 0.0441 | 0.1370 ± 0.0071 |
| GROVER | Deterministic | 0.9067 ± 0.0148 | 0.1003 ± 0.0139 | 0.4230 ± 0.0372 | 0.1254 ± 0.0114 |
| | Temperature | 0.9067 ± 0.0148 | 0.0884 ± 0.0130 | 0.4049 ± 0.0351 | 0.1237 ± 0.0113 |
| | Focal Loss | 0.9084 ± 0.0154 | 0.1600 ± 0.0069 | 0.4651 ± 0.0323 | 0.1492 ± 0.0137 |
| | MC Dropout | 0.9078 ± 0.0168 | 0.0802 ± 0.0069 | 0.4073 ± 0.0394 | 0.1238 ± 0.0119 |
| | SWAG | 0.9112 ± 0.0141 | 0.0887 ± 0.0260 | 0.3916 ± 0.0351 | 0.1203 ± 0.0112 |
| | BBP | 0.9089 ± 0.0065 | 0.0807 ± 0.0112 | 0.3946 ± 0.0178 | 0.1209 ± 0.0103 |
| | SGLD | 0.9066 ± 0.0145 | 0.0852 ± 0.0101 | 0.4029 ± 0.0313 | 0.1231 ± 0.0097 |
| | Ensembles | 0.9046 ± 0.0133 | 0.0915 ± 0.0054 | 0.4207 ± 0.0289 | 0.1248 ± 0.0104 |
| Uni-Mol | Deterministic | 0.9056 ± 0.0204 | 0.0894 ± 0.0129 | 0.4081 ± 0.0529 | 0.1257 ± 0.0191 |
| | Temperature | 0.9059 ± 0.0205 | 0.0854 ± 0.0082 | 0.4060 ± 0.0491 | 0.1261 ± 0.0180 |
| | Focal Loss | 0.8939 ± 0.0206 | 0.1634 ± 0.0148 | 0.5027 ± 0.0139 | 0.1655 ± 0.0065 |
| | MC Dropout | 0.9061 ± 0.0199 | 0.0864 ± 0.0086 | 0.4069 ± 0.0514 | 0.1256 ± 0.0188 |
| | SWAG | 0.9105 ± 0.0142 | 0.0988 ± 0.0122 | 0.3822 ± 0.0340 | 0.1179 ± 0.0131 |
| | BBP | 0.8990 ± 0.0154 | 0.0867 ± 0.0206 | 0.4303 ± 0.0513 | 0.1309 ± 0.0184 |
| | SGLD | 0.9080 ± 0.0104 | 0.0834 ± 0.0139 | 0.3949 ± 0.0317 | 0.1223 ± 0.0138 |
| | Ensembles | 0.9023 ± 0.0189 | 0.1020 ± 0.0140 | 0.4040 ± 0.0491 | 0.1272 ± 0.0220 |

Table 24: Test results on the randomly split BBBP dataset.

| | | ROC-AUC | ECE | NLL | BS |
|---|---|---|---|---|---|
| DNN-rdkit | Deterministic | 0.9215 ± 0.0254 | 0.0838 ± 0.0047 | 0.3722 ± 0.0275 | 0.0918 ± 0.0052 |
| | Temperature | 0.9215 ± 0.0254 | 0.0711 ± 0.0035 | 0.3131 ± 0.0141 | 0.0878 ± 0.0063 |
| | Focal Loss | 0.9180 ± 0.0142 | 0.1214 ± 0.0528 | 0.3506 ± 0.0599 | 0.1038 ± 0.0188 |
| | MC Dropout | 0.9226 ± 0.0244 | 0.0683 ± 0.0007 | 0.3156 ± 0.0151 | 0.0864 ± 0.0079 |
| | SWAG | 0.9205 ± 0.0234 | 0.0816 ± 0.0065 | 0.3358 ± 0.0247 | 0.0876 ± 0.0081 |
| | BBP | 0.9238 ± 0.0149 | 0.0846 ± 0.0188 | 0.3752 ± 0.0675 | 0.0932 ± 0.0113 |
| | SGLD | 0.9172 ± 0.0284 | 0.0865 ± 0.0125 | 0.3930 ± 0.0735 | 0.0905 ± 0.0072 |
| | Ensembles | 0.9226 ± 0.0088 | 0.0700 ± 0.0120 | 0.3324 ± 0.0220 | 0.0856 ± 0.0066 |
| ChemBERTa | Deterministic | 0.9605 ± 0.0034 | 0.0686 ± 0.0066 | 0.2494 ± 0.0089 | 0.0731 ± 0.0030 |
| | Temperature | 0.9605 ± 0.0034 | 0.0591 ± 0.0013 | 0.2275 ± 0.0111 | 0.0697 ± 0.0040 |
| | Focal Loss | 0.9587 ± 0.0027 | 0.0816 ± 0.0262 | 0.2645 ± 0.0437 | 0.0795 ± 0.0133 |
| | MC Dropout | 0.9607 ± 0.0027 | 0.0615 ± 0.0094 | 0.2438 ± 0.0193 | 0.0717 ± 0.0018 |
| | SWAG | 0.9588 ± 0.0018 | 0.0737 ± 0.0041 | 0.2856 ± 0.0097 | 0.0780 ± 0.0039 |
| | BBP | 0.9590 ± 0.0011 | 0.0702 ± 0.0125 | 0.2979 ± 0.0891 | 0.0766 ± 0.0052 |
| | SGLD | 0.9580 ± 0.0033 | 0.0854 ± 0.0057 | 0.3513 ± 0.0324 | 0.0826 ± 0.0055 |
| | Ensembles | 0.9604 ± 0.0018 | 0.0691 ± 0.0124 | 0.2480 ± 0.0045 | 0.0716 ± 0.0028 |
| GROVER | Deterministic | 0.9300 ± 0.0172 | 0.0551 ± 0.0138 | 0.2746 ± 0.0256 | 0.0742 ± 0.0075 |
| | Temperature | 0.9300 ± 0.0171 | 0.0575 ± 0.0185 | 0.2689 ± 0.0252 | 0.0740 ± 0.0074 |
| | Focal Loss | 0.9294 ± 0.0198 | 0.1944 ± 0.0271 | 0.3963 ± 0.0258 | 0.1153 ± 0.0102 |
| | MC Dropout | 0.9311 ± 0.0171 | 0.0549 ± 0.0115 | 0.2701 ± 0.0242 | 0.0742 ± 0.0072 |
| | SWAG | 0.9324 ± 0.0219 | 0.0601 ± 0.0112 | 0.2736 ± 0.0300 | 0.0741 ± 0.0097 |
| | BBP | 0.9196 ± 0.0210 | 0.0549 ± 0.0103 | 0.2712 ± 0.0225 | 0.0751 ± 0.0084 |
| | SGLD | 0.9306 ± 0.0182 | 0.0588 ± 0.0140 | 0.2750 ± 0.0304 | 0.0757 ± 0.0092 |
| | Ensembles | 0.9320 ± 0.0141 | 0.0633 ± 0.0124 | 0.2739 ± 0.0305 | 0.0738 ± 0.0092 |
| Uni-Mol | Deterministic | 0.9285 ± 0.0126 | 0.0785 ± 0.0103 | 0.3165 ± 0.0332 | 0.0810 ± 0.0108 |
| | Temperature | 0.9272 ± 0.0121 | 0.0687 ± 0.0094 | 0.2838 ± 0.0231 | 0.0790 ± 0.0099 |
| | Focal Loss | 0.9242 ± 0.0086 | 0.1001 ± 0.0261 | 0.3201 ± 0.0332 | 0.0936 ± 0.0109 |
| | MC Dropout | 0.9284 ± 0.0125 | 0.0750 ± 0.0087 | 0.3124 ± 0.0322 | 0.0807 ± 0.0107 |
| | SWAG | 0.9359 ± 0.0048 | 0.0797 ± 0.0086 | 0.3233 ± 0.0266 | 0.0792 ± 0.0091 |
| | BBP | 0.9175 ± 0.0121 | 0.0706 ± 0.0047 | 0.3256 ± 0.0262 | 0.0809 ± 0.0094 |
| | SGLD | 0.9234 ± 0.0069 | 0.0869 ± 0.0053 | 0.3671 ± 0.0262 | 0.0846 ± 0.0109 |
| | Ensembles | 0.9341 ± 0.0117 | 0.0652 ± 0.0140 | 0.2925 ± 0.0429 | 0.0766 ± 0.0107 |

Table 25: Test results on the randomly split ClinTox dataset.

| | | ROC-AUC | ECE | NLL | BS |
|---|---|---|---|---|---|
| DNN-rdkit | Deterministic | 0.8659 ± 0.0483 | 0.0754 ± 0.0129 | 0.5589 ± 0.4190 | 0.0672 ± 0.0104 |
| | Temperature | 0.8628 ± 0.0300 | 0.0719 ± 0.0087 | 0.4434 ± 0.1979 | 0.0645 ± 0.0068 |
| | Focal Loss | 0.8813 ± 0.0564 | 0.0652 ± 0.0054 | 0.1980 ± 0.0696 | 0.0523 ± 0.0093 |
| | MC Dropout | 0.8757 ± 0.0382 | 0.0616 ± 0.0093 | 0.2560 ± 0.0963 | 0.0566 ± 0.0067 |
| | SWAG | 0.8819 ± 0.0424 | 0.0619 ± 0.0070 | 0.2574 ± 0.1058 | 0.0572 ± 0.0066 |
| | BBP | 0.8358 ± 0.0778 | 0.0547 ± 0.0198 | 0.2679 ± 0.1254 | 0.0534 ± 0.0083 |
| | SGLD | 0.8745 ± 0.0506 | 0.0666 ± 0.0085 | 0.2488 ± 0.0714 | 0.0572 ± 0.0068 |
| | Ensembles | 0.8241 ± 0.0587 | 0.0646 ± 0.0088 | 0.4562 ± 0.3799 | 0.0561 ± 0.0077 |
| ChemBERTa | Deterministic | 0.9862 ± 0.0063 | 0.0193 ± 0.0026 | 0.0703 ± 0.0063 | 0.0168 ± 0.0015 |
| | Temperature | 0.9862 ± 0.0063 | 0.0181 ± 0.0045 | 0.0639 ± 0.0039 | 0.0163 ± 0.0013 |
| | Focal Loss | 0.9824 ± 0.0038 | 0.0493 ± 0.0183 | 0.0907 ± 0.0132 | 0.0193 ± 0.0025 |
| | MC Dropout | 0.9870 ± 0.0051 | 0.0162 ± 0.0029 | 0.0640 ± 0.0087 | 0.0148 ± 0.0025 |
| | SWAG | 0.9863 ± 0.0060 | 0.0216 ± 0.0036 | 0.0751 ± 0.0084 | 0.0179 ± 0.0019 |
| | BBP | 0.9855 ± 0.0077 | 0.0268 ± 0.0124 | 0.0687 ± 0.0095 | 0.0165 ± 0.0021 |
| | SGLD | 0.9851 ± 0.0053 | 0.0201 ± 0.0058 | 0.0713 ± 0.0020 | 0.0178 ± 0.0035 |
| | Ensembles | 0.9885 ± 0.0034 | 0.0242 ± 0.0108 | 0.0633 ± 0.0056 | 0.0154 ± 0.0015 |
| GROVER | Deterministic | 0.8823 ± 0.0347 | 0.0392 ± 0.0059 | 0.1550 ± 0.0289 | 0.0409 ± 0.0069 |
| | Temperature | 0.8818 ± 0.0341 | 0.0387 ± 0.0041 | 0.1575 ± 0.0294 | 0.0412 ± 0.0069 |
| | Focal Loss | 0.8554 ± 0.0542 | 0.1687 ± 0.0024 | 0.2767 ± 0.0073 | 0.0682 ± 0.0026 |
| | MC Dropout | 0.8840 ± 0.0332 | 0.0324 ± 0.0015 | 0.1505 ± 0.0242 | 0.0392 ± 0.0058 |
| | SWAG | 0.8897 ± 0.0337 | 0.0400 ± 0.0033 | 0.1490 ± 0.0267 | 0.0397 ± 0.0060 |
| | BBP | 0.8466 ± 0.0496 | 0.0684 ± 0.0059 | 0.1748 ± 0.0226 | 0.0419 ± 0.0064 |
| | SGLD | 0.8904 ± 0.0285 | 0.0424 ± 0.0021 | 0.1545 ± 0.0297 | 0.0414 ± 0.0086 |
| | Ensembles | 0.8854 ± 0.0371 | 0.0342 ± 0.0102 | 0.1470 ± 0.0295 | 0.0385 ± 0.0071 |
| Uni-Mol | Deterministic | 0.8391 ± 0.0620 | 0.0625 ± 0.0119 | 0.1949 ± 0.0291 | 0.0531 ± 0.0078 |
| | Temperature | 0.8433 ± 0.0611 | 0.0633 ± 0.0119 | 0.1880 ± 0.0236 | 0.0526 ± 0.0073 |
| | Focal Loss | 0.8416 ± 0.0705 | 0.0961 ± 0.0199 | 0.2153 ± 0.0136 | 0.0586 ± 0.0050 |
| | MC Dropout | 0.8394 ± 0.0616 | 0.0627 ± 0.0112 | 0.1931 ± 0.0281 | 0.0528 ± 0.0077 |
| | SWAG | 0.8211 ± 0.0871 | 0.0515 ± 0.0131 | 0.1974 ± 0.0588 | 0.0484 ± 0.0097 |
| | BBP | 0.8389 ± 0.0345 | 0.0593 ± 0.0234 | 0.1788 ± 0.0387 | 0.0472 ± 0.0132 |
| | SGLD | 0.8367 ± 0.0855 | 0.0462 ± 0.0092 | 0.1867 ± 0.0655 | 0.0475 ± 0.0110 |
| | Ensembles | 0.8387 ± 0.0463 | 0.0671 ± 0.0197 | 0.1803 ± 0.0254 | 0.0469 ± 0.0032 |

Table 26: Test results on the randomly split Tox21 dataset.

| | | ROC-AUC | ECE | NLL | BS |
|---|---|---|---|---|---|
| DNN-rdkit | Deterministic | 0.8451 ± 0.0162 | 0.0335 ± 0.0012 | 0.1999 ± 0.0041 | 0.0537 ± 0.0005 |
| | Temperature | 0.8451 ± 0.0162 | 0.0338 ± 0.0023 | 0.1945 ± 0.0016 | 0.0534 ± 0.0005 |
| | Focal Loss | 0.8422 ± 0.0120 | 0.0829 ± 0.0065 | 0.2265 ± 0.0055 | 0.0587 ± 0.0010 |
| | MC Dropout | 0.8494 ± 0.0137 | 0.0259 ± 0.0009 | 0.1885 ± 0.0012 | 0.0510 ± 0.0002 |
| | SWAG | 0.8496 ± 0.0129 | 0.0256 ± 0.0009 | 0.1882 ± 0.0008 | 0.0509 ± 0.0003 |
| | BBP | 0.8216 ± 0.0033 | 0.0293 ± 0.0018 | 0.2052 ± 0.0054 | 0.0565 ± 0.0011 |
| | SGLD | 0.8114 ± 0.0067 | 0.0864 ± 0.0038 | 0.2568 ± 0.0084 | 0.0685 ± 0.0028 |
| | Ensembles | 0.8544 ± 0.0101 | 0.0271 ± 0.0025 | 0.1849 ± 0.0018 | 0.0506 ± 0.0005 |
| ChemBERTa | Deterministic | 0.8373 ± 0.0140 | 0.0345 ± 0.0046 | 0.1993 ± 0.0110 | 0.0545 ± 0.0027 |
| | Temperature | 0.8373 ± 0.0140 | 0.0360 ± 0.0020 | 0.1973 ± 0.0066 | 0.0543 ± 0.0022 |
| | Focal Loss | 0.8339 ± 0.0128 | 0.0939 ± 0.0078 | 0.2361 ± 0.0024 | 0.0605 ± 0.0007 |
| | MC Dropout | 0.8430 ± 0.0101 | 0.0240 ± 0.0017 | 0.1859 ± 0.0069 | 0.0506 ± 0.0019 |
| | SWAG | 0.8369 ± 0.0149 | 0.0370 ± 0.0025 | 0.2027 ± 0.0110 | 0.0553 ± 0.0022 |
| | BBP | 0.8301 ± 0.0131 | 0.0403 ± 0.0038 | 0.2052 ± 0.0077 | 0.0575 ± 0.0025 |
| | SGLD | 0.8314 ± 0.0127 | 0.0376 ± 0.0018 | 0.2002 ± 0.0102 | 0.0552 ± 0.0031 |
| | Ensembles | 0.8509 ± 0.0082 | 0.0304 ± 0.0027 | 0.1890 ± 0.0100 | 0.0519 ± 0.0025 |
| GROVER | Deterministic | 0.8754 ± 0.0155 | 0.0259 ± 0.0018 | 0.1690 ± 0.0067 | 0.0457 ± 0.0015 |
| | Temperature | 0.8754 ± 0.0154 | 0.0281 ± 0.0030 | 0.1700 ± 0.0060 | 0.0458 ± 0.0015 |
| | Focal Loss | 0.8785 ± 0.0099 | 0.1337 ± 0.0012 | 0.2623 ± 0.0046 | 0.0636 ± 0.0017 |
| | MC Dropout | 0.8760 ± 0.0149 | 0.0279 ± 0.0015 | 0.1697 ± 0.0062 | 0.0457 ± 0.0015 |
| | SWAG | 0.8776 ± 0.0135 | 0.0247 ± 0.0014 | 0.1674 ± 0.0059 | 0.0452 ± 0.0014 |
| | BBP | 0.8667 ± 0.0135 | 0.0439 ± 0.0038 | 0.1850 ± 0.0049 | 0.0483 ± 0.0016 |
| | SGLD | 0.8724 ± 0.0123 | 0.0390 ± 0.0023 | 0.1811 ± 0.0061 | 0.0473 ± 0.0019 |
| | Ensembles | 0.8775 ± 0.0137 | 0.0244 ± 0.0010 | 0.1680 ± 0.0055 | 0.0453 ± 0.0013 |
| Uni-Mol | Deterministic | 0.8666 ± 0.0138 | 0.0302 ± 0.0016 | 0.1773 ± 0.0052 | 0.0466 ± 0.0012 |
| | Temperature | 0.8661 ± 0.0136 | 0.0295 ± 0.0008 | 0.1727 ± 0.0037 | 0.0464 ± 0.0010 |
| | Focal Loss | 0.8631 ± 0.0071 | 0.0824 ± 0.0093 | 0.2154 ± 0.0097 | 0.0543 ± 0.0028 |
| | MC Dropout | 0.8667 ± 0.0139 | 0.0292 ± 0.0021 | 0.1758 ± 0.0050 | 0.0465 ± 0.0012 |
| | SWAG | 0.8666 ± 0.0170 | 0.0308 ± 0.0018 | 0.1779 ± 0.0042 | 0.0452 ± 0.0010 |
| | BBP | 0.8731 ± 0.0107 | 0.0259 ± 0.0031 | 0.1731 ± 0.0046 | 0.0469 ± 0.0013 |
| | SGLD | 0.8663 ± 0.0149 | 0.0333 ± 0.0027 | 0.1807 ± 0.0058 | 0.0470 ± 0.0011 |
| | Ensembles | 0.8778 ± 0.0090 | 0.0262 ± 0.0024 | 0.1660 ± 0.0043 | 0.0440 ± 0.0008 |

Table 27: Test results on the randomly split ESOL dataset.

|  |  | RMSE | MAE | NLL | CE |
|---|---|---|---|---|---|
| DNN-rdkit | Deterministic | $0.6530 \pm 0.0615$ | $0.4788 \pm 0.0279$ | $0.1067 \pm 0.0716$ | $0.0419 \pm 0.0034$ |
|  | MC Dropout | $0.6597 \pm 0.0666$ | $0.4820 \pm 0.0351$ | $0.1212 \pm 0.0925$ | $0.0436 \pm 0.0037$ |
|  | SWAG | $0.6542 \pm 0.0766$ | $0.4752 \pm 0.0392$ | $0.0977 \pm 0.0884$ | $0.0435 \pm 0.0034$ |
|  | BBP | $0.7060 \pm 0.0604$ | $0.5294 \pm 0.0458$ | $0.2153 \pm 0.1182$ | $0.0263 \pm 0.0050$ |
|  | SGLD | $0.7033 \pm 0.0648$ | $0.5235 \pm 0.0273$ | $0.3825 \pm 0.0019$ | $0.0163 \pm 0.0019$ |
|  | Ensembles | $0.6455 \pm 0.0518$ | $0.4735 \pm 0.0310$ | $0.0791 \pm 0.0905$ | $0.0405 \pm 0.0031$ |
| ChemBERTa | Deterministic | $0.7760 \pm 0.0921$ | $0.5920 \pm 0.0465$ | $0.3768 \pm 0.1070$ | $0.0464 \pm 0.0042$ |
|  | MC Dropout | $0.6919 \pm 0.0947$ | $0.5192 \pm 0.0619$ | $0.1908 \pm 0.1299$ | $0.0448 \pm 0.0043$ |
|  | SWAG | $0.7704 \pm 0.0784$ | $0.5921 \pm 0.0350$ | $0.4094 \pm 0.1176$ | $0.0475 \pm 0.0046$ |
|  | BBP | $0.7648 \pm 0.0328$ | $0.5923 \pm 0.0176$ | $0.2208 \pm 0.0289$ | $0.0326 \pm 0.0022$ |
|  | SGLD | $0.7282 \pm 0.0764$ | $0.5539 \pm 0.0290$ | $0.2455 \pm 0.0835$ | $0.0456 \pm 0.0030$ |
|  | Ensembles | $0.7318 \pm 0.0778$ | $0.5641 \pm 0.0416$ | $0.3073 \pm 0.1107$ | $0.0470 \pm 0.0032$ |
| GROVER | Deterministic | $0.6137 \pm 0.0392$ | $0.4730 \pm 0.0173$ | $0.0299 \pm 0.0935$ | $0.0377 \pm 0.0063$ |
|  | MC Dropout | $0.5997 \pm 0.0256$ | $0.4621 \pm 0.0110$ | $0.0124 \pm 0.0890$ | $0.0376 \pm 0.0065$ |
|  | SWAG | $0.5895 \pm 0.0251$ | $0.4476 \pm 0.0102$ | $-0.0342 \pm 0.0624$ | $0.0419 \pm 0.0035$ |
|  | BBP | $0.6449 \pm 0.0517$ | $0.4892 \pm 0.0286$ | $0.3780 \pm 0.0474$ | $0.0152 \pm 0.0018$ |
|  | SGLD | $0.5888 \pm 0.0379$ | $0.4542 \pm 0.0252$ | $0.0748 \pm 0.0683$ | $0.0299 \pm 0.0010$ |
|  | Ensembles | $0.5847 \pm 0.0326$ | $0.4526 \pm 0.0210$ | $-0.0377 \pm 0.0725$ | $0.0384 \pm 0.0031$ |
| Uni-Mol | Deterministic | $0.6253 \pm 0.0500$ | $0.4691 \pm 0.0291$ | $0.0024 \pm 0.0693$ | $0.0388 \pm 0.0008$ |
|  | MC Dropout | $0.6224 \pm 0.0501$ | $0.4640 \pm 0.0299$ | $0.0058 \pm 0.0647$ | $0.0375 \pm 0.0007$ |
|  | SWAG | $0.5642 \pm 0.0564$ | $0.4030 \pm 0.0342$ | $-0.0603 \pm 0.1092$ | $0.0361 \pm 0.0028$ |
|  | BBP | $0.6762 \pm 0.0766$ | $0.4884 \pm 0.0544$ | $0.2844 \pm 0.0276$ | $0.0235 \pm 0.0024$ |
|  | SGLD | $0.6063 \pm 0.0281$ | $0.4388 \pm 0.0098$ | $0.1947 \pm 0.2311$ | $0.0270 \pm 0.0096$ |
|  | Ensembles | $0.5930 \pm 0.0374$ | $0.4311 \pm 0.0129$ | $-0.0273 \pm 0.0238$ | $0.0375 \pm 0.0015$ |

Table 28: Test results on the randomly split FreeSolv dataset.

|  |  | RMSE | MAE | NLL | CE |
|---|---|---|---|---|---|
| DNN-rdkit | Deterministic | $1.1867 \pm 0.0369$ | $0.8582 \pm 0.0521$ | $0.7262 \pm 0.0311$ | $0.0252 \pm 0.0038$ |
|  | MC Dropout | $1.1893 \pm 0.0716$ | $0.8576 \pm 0.0283$ | $0.7085 \pm 0.0277$ | $0.0263 \pm 0.0038$ |
|  | SWAG | $1.1183 \pm 0.0960$ | $0.7817 \pm 0.0602$ | $0.6026 \pm 0.0144$ | $0.0305 \pm 0.0047$ |
|  | BBP | $1.8405 \pm 0.2306$ | $1.2685 \pm 0.0533$ | $1.2661 \pm 0.0458$ | $0.0060 \pm 0.0013$ |
|  | SGLD | $1.5395 \pm 0.1830$ | $1.0855 \pm 0.0925$ | $1.2214 \pm 0.0395$ | $0.0045 \pm 0.0020$ |
|  | Ensembles | $1.1690 \pm 0.0630$ | $0.8500 \pm 0.0239$ | $0.7444 \pm 0.0502$ | $0.0240 \pm 0.0042$ |
| ChemBERTa | Deterministic | $1.4125 \pm 0.1890$ | $1.0323 \pm 0.1206$ | $0.8274 \pm 0.1999$ | $0.0387 \pm 0.0059$ |
|  | MC Dropout | $1.3675 \pm 0.2456$ | $1.0278 \pm 0.1732$ | $0.6970 \pm 0.1730$ | $0.0357 \pm 0.0067$ |
|  | SWAG | $1.3789 \pm 0.2128$ | $1.0024 \pm 0.1570$ | $0.8477 \pm 0.2851$ | $0.0408 \pm 0.0041$ |
|  | BBP | $1.4355 \pm 0.2128$ | $1.0643 \pm 0.1312$ | $0.8143 \pm 0.0637$ | $0.0270 \pm 0.0032$ |
|  | SGLD | $1.3864 \pm 0.2747$ | $0.9852 \pm 0.1613$ | $0.6654 \pm 0.1475$ | $0.0361 \pm 0.0023$ |
|  | Ensembles | $1.2276 \pm 0.2395$ | $0.8745 \pm 0.1727$ | $0.5995 \pm 0.1546$ | $0.0399 \pm 0.0013$ |
| GROVER | Deterministic | $1.3823 \pm 0.1108$ | $1.0223 \pm 0.0942$ | $0.9105 \pm 0.0684$ | $0.0188 \pm 0.0018$ |
|  | MC Dropout | $1.3570 \pm 0.1082$ | $1.0093 \pm 0.0940$ | $0.9247 \pm 0.0680$ | $0.0177 \pm 0.0017$ |
|  | SWAG | $1.3181 \pm 0.0898$ | $0.9858 \pm 0.0727$ | $0.8386 \pm 0.0352$ | $0.0218 \pm 0.0042$ |
|  | BBP | $1.4832 \pm 0.1322$ | $1.1451 \pm 0.0893$ | $1.1459 \pm 0.0166$ | $0.0060 \pm 0.0019$ |
|  | SGLD | $1.4965 \pm 0.2643$ | $1.1682 \pm 0.2289$ | $1.1154 \pm 0.1317$ | $0.0072 \pm 0.0016$ |
|  | Ensembles | $1.2826 \pm 0.1700$ | $0.9424 \pm 0.1316$ | $0.9476 \pm 0.0558$ | $0.0140 \pm 0.0020$ |
| Uni-Mol | Deterministic | $1.0429 \pm 0.0170$ | $0.7345 \pm 0.0308$ | $0.5741 \pm 0.0637$ | $0.0322 \pm 0.0054$ |
|  | MC Dropout | $1.0381 \pm 0.0249$ | $0.7280 \pm 0.0394$ | $0.5872 \pm 0.0636$ | $0.0310 \pm 0.0055$ |
|  | SWAG | $0.9959 \pm 0.0902$ | $0.6842 \pm 0.1248$ | $0.6050 \pm 0.0705$ | $0.0271 \pm 0.0051$ |
|  | BBP | $1.1541 \pm 0.1779$ | $0.8681 \pm 0.1550$ | $0.9795 \pm 0.0621$ | $0.0154 \pm 0.0027$ |
|  | SGLD | $1.0486 \pm 0.1898$ | $0.7543 \pm 0.1538$ | $1.0065 \pm 0.1283$ | $0.0126 \pm 0.0016$ |
|  | Ensembles | $1.0009 \pm 0.0416$ | $0.7201 \pm 0.0378$ | $0.6766 \pm 0.0365$ | $0.0270 \pm 0.0045$ |

Table 29: Test results on the randomly split Lipophilicity dataset.

| | | RMSE | MAE | NLL | CE |
|---|---|---|---|---|---|
| DNN-rdkit | Deterministic | $0.6790 \pm 0.0415$ | $0.5007 \pm 0.0253$ | $0.6158 \pm 0.1691$ | $0.0365 \pm 0.0049$ |
| | MC Dropout | $0.6732 \pm 0.0398$ | $0.4971 \pm 0.0286$ | $0.5558 \pm 0.1208$ | $0.0366 \pm 0.0048$ |
| | SWAG | $0.6699 \pm 0.0441$ | $0.4926 \pm 0.0277$ | $0.6208 \pm 0.1477$ | $0.0370 \pm 0.0052$ |
| | BBP | $0.6833 \pm 0.0444$ | $0.5092 \pm 0.0222$ | $0.2681 \pm 0.2026$ | $0.0281 \pm 0.0040$ |
| | SGLD | $0.6819 \pm 0.0427$ | $0.5090 \pm 0.0222$ | $0.1218 \pm 0.0554$ | $0.0134 \pm 0.0010$ |
| | Ensembles | $0.6399 \pm 0.0402$ | $0.4690 \pm 0.0199$ | $0.3242 \pm 0.1517$ | $0.0349 \pm 0.0004$ |
| ChemBERTa | Deterministic | $0.7033 \pm 0.0521$ | $0.5282 \pm 0.0234$ | $1.4475 \pm 1.1982$ | $0.0418 \pm 0.0064$ |
| | MC Dropout | $0.6583 \pm 0.0330$ | $0.4882 \pm 0.0095$ | $0.8234 \pm 0.7527$ | $0.0380 \pm 0.0056$ |
| | SWAG | $0.7114 \pm 0.0502$ | $0.5363 \pm 0.0196$ | $1.7413 \pm 1.2705$ | $0.0441 \pm 0.0059$ |
| | BBP | $0.7103 \pm 0.0508$ | $0.5377 \pm 0.0212$ | $0.5764 \pm 0.3161$ | $0.0368 \pm 0.0043$ |
| | SGLD | $0.6988 \pm 0.0540$ | $0.5271 \pm 0.0249$ | $0.7978 \pm 0.2380$ | $0.0414 \pm 0.0016$ |
| | Ensembles | $0.6763 \pm 0.0534$ | $0.5044 \pm 0.0232$ | $1.4618 \pm 0.8428$ | $0.0451 \pm 0.0033$ |
| GROVER | Deterministic | $0.5717 \pm 0.0320$ | $0.3957 \pm 0.0144$ | $0.6373 \pm 0.0281$ | $0.0446 \pm 0.0016$ |
| | MC Dropout | $0.5633 \pm 0.0349$ | $0.3900 \pm 0.0163$ | $0.5493 \pm 0.0500$ | $0.0441 \pm 0.0018$ |
| | SWAG | $0.5599 \pm 0.0340$ | $0.3851 \pm 0.0117$ | $0.5192 \pm 0.1369$ | $0.0440 \pm 0.0002$ |
| | BBP | $0.5638 \pm 0.0314$ | $0.4021 \pm 0.0088$ | $-0.0783 \pm 0.0532$ | $0.0232 \pm 0.0010$ |
| | SGLD | $0.5597 \pm 0.0409$ | $0.3958 \pm 0.0171$ | $-0.0915 \pm 0.0856$ | $0.0271 \pm 0.0006$ |
| | Ensembles | $0.5539 \pm 0.0314$ | $0.3833 \pm 0.0134$ | $0.4942 \pm 0.1247$ | $0.0447 \pm 0.0010$ |
| Uni-Mol | Deterministic | $0.5387 \pm 0.0439$ | $0.3824 \pm 0.0203$ | $-0.0690 \pm 0.1128$ | $0.0362 \pm 0.0024$ |
| | MC Dropout | $0.5367 \pm 0.0437$ | $0.3812 \pm 0.0204$ | $-0.0859 \pm 0.1088$ | $0.0355 \pm 0.0024$ |
| | SWAG | $0.5077 \pm 0.0446$ | $0.3598 \pm 0.0178$ | $-0.1168 \pm 0.1292$ | $0.0369 \pm 0.0018$ |
| | BBP | $0.5632 \pm 0.0509$ | $0.4139 \pm 0.0207$ | $-0.0682 \pm 0.0724$ | $0.0248 \pm 0.0039$ |
| | SGLD | $0.5238 \pm 0.0357$ | $0.3718 \pm 0.0160$ | $-0.1525 \pm 0.0618$ | $0.0265 \pm 0.0011$ |
| | Ensembles | $0.5255 \pm 0.0434$ | $0.3706 \pm 0.0187$ | $-0.1311 \pm 0.1122$ | $0.0347 \pm 0.0010$ |

Table 30: Test results on the randomly split QM7 dataset.

| | | RMSE | MAE | NLL | CE |
|---|---|---|---|---|---|
| DNN-rdkit | Deterministic | $107.9546 \pm 8.1501$ | $57.5611 \pm 4.8507$ | $5.5231 \pm 0.0677$ | $0.0449 \pm 0.0026$ |
| | MC Dropout | $106.5277 \pm 8.4035$ | $56.8402 \pm 4.4571$ | $4.9801 \pm 0.0805$ | $0.0434 \pm 0.0025$ |
| | SWAG | $105.3930 \pm 8.3626$ | $56.1532 \pm 3.9650$ | $4.9387 \pm 0.1316$ | $0.0429 \pm 0.0018$ |
| | BBP | $111.6410 \pm 8.9230$ | $65.9097 \pm 3.7377$ | $5.1715 \pm 0.2092$ | $0.0388 \pm 0.0020$ |
| | SGLD | $111.1149 \pm 7.8189$ | $75.5816 \pm 4.4419$ | $5.2277 \pm 0.0341$ | $0.0127 \pm 0.0004$ |
| | Ensembles | $104.0890 \pm 9.9660$ | $55.6554 \pm 4.1624$ | $5.0123 \pm 0.1890$ | $0.0438 \pm 0.0020$ |
| ChemBERTa | Deterministic | $97.4000 \pm 8.8093$ | $47.3906 \pm 3.6007$ | $4.5928 \pm 0.1364$ | $0.0402 \pm 0.0021$ |
| | MC Dropout | $97.1763 \pm 9.0430$ | $43.4817 \pm 4.8435$ | $4.3224 \pm 0.3558$ | $0.0458 \pm 0.0017$ |
| | SWAG | $97.5653 \pm 8.4876$ | $49.3942 \pm 4.0995$ | $4.7928 \pm 0.1495$ | $0.0397 \pm 0.0019$ |
| | BBP | $100.8003 \pm 8.9576$ | $53.0419 \pm 4.4522$ | $4.6938 \pm 0.2195$ | $0.0347 \pm 0.0027$ |
| | SGLD | $99.6487 \pm 7.9623$ | $54.8976 \pm 2.4214$ | $4.6314 \pm 0.1117$ | $0.0308 \pm 0.0015$ |
| | Ensembles | $97.5500 \pm 8.6923$ | $47.2016 \pm 3.7146$ | $4.5436 \pm 0.2688$ | $0.0392 \pm 0.0020$ |
| GROVER | Deterministic | $98.7533 \pm 9.8780$ | $42.7787 \pm 4.8753$ | $4.7424 \pm 0.3523$ | $0.0512 \pm 0.0003$ |
| | MC Dropout | $96.6699 \pm 9.6047$ | $42.6411 \pm 4.6449$ | $4.7324 \pm 0.3130$ | $0.0511 \pm 0.0003$ |
| | SWAG | $97.1293 \pm 10.0641$ | $43.3197 \pm 5.1696$ | $4.6304 \pm 0.2678$ | $0.0495 \pm 0.0008$ |
| | BBP | $99.0196 \pm 8.1011$ | $56.8246 \pm 3.9466$ | $4.9841 \pm 0.0630$ | $0.0223 \pm 0.0003$ |
| | SGLD | $101.0303 \pm 6.6971$ | $57.6752 \pm 3.7755$ | $4.9718 \pm 0.0556$ | $0.0237 \pm 0.0009$ |
| | Ensembles | $98.0631 \pm 9.7788$ | $42.8305 \pm 4.6214$ | $4.6874 \pm 0.3069$ | $0.0498 \pm 0.0006$ |
| Uni-Mol | Deterministic | $71.4475 \pm 2.3163$ | $36.7968 \pm 2.3826$ | $4.1470 \pm 0.1635$ | $0.0411 \pm 0.0032$ |
| | MC Dropout | $70.8934 \pm 2.2622$ | $35.5291 \pm 2.4115$ | $4.1425 \pm 0.1709$ | $0.0406 \pm 0.0032$ |
| | SWAG | $68.6536 \pm 1.3320$ | $29.1093 \pm 2.1596$ | $3.9460 \pm 0.0134$ | $0.0433 \pm 0.0011$ |
| | BBP | $73.0523 \pm 1.1094$ | $39.2194 \pm 1.8792$ | $4.5952 \pm 0.0996$ | $0.0297 \pm 0.0042$ |
| | SGLD | $71.2770 \pm 1.6947$ | $32.4820 \pm 3.3687$ | $4.4407 \pm 0.0538$ | $0.0321 \pm 0.0017$ |
| | Ensembles | $69.6610 \pm 0.9264$ | $33.8478 \pm 0.0577$ | $4.0436 \pm 0.0870$ | $0.0429 \pm 0.0020$ |

Table 31: Test results with frozen backbone model weights on BACE.

|  |  | ROC-AUC | ECE | NLL | BS |
|---|---|---|---|---|---|
| ChemBERTa | Deterministic | $0.6289 \pm 0.0090$ | $0.2096 \pm 0.0037$ | $0.7570 \pm 0.0073$ | $0.2751 \pm 0.0028$ |
|  | Temperature | $0.6289 \pm 0.0090$ | $0.1554 \pm 0.0103$ | $0.7134 \pm 0.0032$ | $0.2587 \pm 0.0014$ |
|  | Focal Loss | $0.6474 \pm 0.0051$ | $0.1964 \pm 0.0035$ | $0.7450 \pm 0.0023$ | $0.2741 \pm 0.0011$ |
|  | MC Dropout | $0.6515 \pm 0.0164$ | $0.1931 \pm 0.0045$ | $0.7393 \pm 0.0029$ | $0.2693 \pm 0.0013$ |
|  | SWAG | $0.6305 \pm 0.0081$ | $0.2045 \pm 0.0032$ | $0.7559 \pm 0.0077$ | $0.2746 \pm 0.0030$ |
|  | BBP | $0.5454 \pm 0.0619$ | $0.0998 \pm 0.0235$ | $0.7080 \pm 0.0087$ | $0.2573 \pm 0.0042$ |
|  | SGLD | $0.5217 \pm 0.0512$ | $0.0531 \pm 0.0114$ | $0.6941 \pm 0.0022$ | $0.2505 \pm 0.0011$ |
|  | Ensembles | $0.6308 \pm$ - | $0.1415 \pm$ - | $0.7081 \pm$ - | $0.2569 \pm$ - |
| GROVER | Deterministic | $0.7633 \pm 0.0116$ | $0.2083 \pm 0.0100$ | $0.6934 \pm 0.0070$ | $0.2514 \pm 0.0030$ |
|  | Temperature | $0.7635 \pm 0.0117$ | $0.1915 \pm 0.0109$ | $0.6826 \pm 0.0067$ | $0.2461 \pm 0.0029$ |
|  | Focal Loss | $0.7221 \pm 0.0343$ | $0.2125 \pm 0.0066$ | $0.7358 \pm 0.0048$ | $0.2710 \pm 0.0020$ |
|  | MC Dropout | $0.7672 \pm 0.0081$ | $0.2221 \pm 0.0111$ | $0.7037 \pm 0.0050$ | $0.2561 \pm 0.0020$ |
|  | SWAG | $0.7691 \pm 0.0096$ | $0.2232 \pm 0.0131$ | $0.7027 \pm 0.0036$ | $0.2559 \pm 0.0013$ |
|  | BBP | $0.7425 \pm 0.0089$ | $0.1946 \pm 0.0135$ | $0.7046 \pm 0.0038$ | $0.2565 \pm 0.0015$ |
|  | SGLD | $0.5485 \pm 0.0612$ | $0.0944 \pm 0.0227$ | $0.7007 \pm 0.0135$ | $0.2536 \pm 0.0067$ |
|  | Ensembles | $0.7826 \pm$ - | $0.2278 \pm$ - | $0.6807 \pm$ - | $0.2462 \pm$ - |
| Uni-Mol | Deterministic | $0.3713 \pm 0.1533$ | $0.1999 \pm 0.0357$ | $0.7544 \pm 0.0216$ | $0.2795 \pm 0.0101$ |
|  | Temperature | $0.3712 \pm 0.1534$ | $0.1811 \pm 0.0340$ | $0.7317 \pm 0.0149$ | $0.2689 \pm 0.0072$ |
|  | Focal Loss | $0.3512 \pm 0.1450$ | $0.2720 \pm 0.0414$ | $0.7977 \pm 0.0301$ | $0.2985 \pm 0.0131$ |
|  | MC Dropout | $0.3698 \pm 0.1569$ | $0.2052 \pm 0.0513$ | $0.7512 \pm 0.0203$ | $0.2780 \pm 0.0095$ |
|  | SWAG | $0.4297 \pm 0.1408$ | $0.1513 \pm 0.0237$ | $0.7330 \pm 0.0089$ | $0.2695 \pm 0.0042$ |
|  | BBP | $0.5958 \pm 0.0671$ | $0.1614 \pm 0.0191$ | $0.7361 \pm 0.0096$ | $0.2709 \pm 0.0045$ |
|  | SGLD | $0.7189 \pm 0.0291$ | $0.1414 \pm 0.0218$ | $0.6737 \pm 0.0091$ | $0.2405 \pm 0.0045$ |
|  | Ensembles | $0.6799 \pm$ - | $0.1543 \pm$ - | $0.7370 \pm$ - | $0.2714 \pm$ - |

Table 32: Test results with frozen backbone model weights on BBBP.

|  |  | ROC-AUC | ECE | NLL | BS |
|---|---|---|---|---|---|
| ChemBERTa | Deterministic | $0.6651 \pm 0.0075$ | $0.1025 \pm 0.0103$ | $0.6534 \pm 0.0072$ | $0.2308 \pm 0.0033$ |
|  | Temperature | $0.6651 \pm 0.0075$ | $0.1681 \pm 0.0189$ | $0.7366 \pm 0.0315$ | $0.2518 \pm 0.0073$ |
|  | Focal Loss | $0.6600 \pm 0.0082$ | $0.1342 \pm 0.0038$ | $0.6782 \pm 0.0017$ | $0.2430 \pm 0.0009$ |
|  | MC Dropout | $0.6592 \pm 0.0084$ | $0.1358 \pm 0.0159$ | $0.6813 \pm 0.0058$ | $0.2428 \pm 0.0030$ |
|  | SWAG | $0.6650 \pm 0.0079$ | $0.1068 \pm 0.0112$ | $0.6537 \pm 0.0074$ | $0.2310 \pm 0.0033$ |
|  | BBP | $0.6280 \pm 0.0074$ | $0.1684 \pm 0.0089$ | $0.7250 \pm 0.0082$ | $0.2645 \pm 0.0035$ |
|  | SGLD | $0.4822 \pm 0.0247$ | $0.0346 \pm 0.0091$ | $0.6945 \pm 0.0020$ | $0.2507 \pm 0.0010$ |
|  | Ensembles | $0.6545 \pm$ - | $0.1059 \pm$ - | $0.6539 \pm$ - | $0.2320 \pm$ - |
| GROVER | Deterministic | $0.6088 \pm 0.0124$ | $0.3048 \pm 0.0071$ | $0.9424 \pm 0.0027$ | $0.3303 \pm 0.0024$ |
|  | Temperature | $0.6083 \pm 0.0122$ | $0.2738 \pm 0.0073$ | $0.8674 \pm 0.0022$ | $0.3119 \pm 0.0019$ |
|  | Focal Loss | $0.5979 \pm 0.0167$ | $0.0867 \pm 0.0234$ | $0.6872 \pm 0.0071$ | $0.2469 \pm 0.0034$ |
|  | MC Dropout | $0.6089 \pm 0.0135$ | $0.3073 \pm 0.0114$ | $0.9372 \pm 0.0028$ | $0.3297 \pm 0.0021$ |
|  | SWAG | $0.6179 \pm 0.0112$ | $0.3087 \pm 0.0038$ | $0.9424 \pm 0.0042$ | $0.3300 \pm 0.0022$ |
|  | BBP | $0.6096 \pm 0.0069$ | $0.3023 \pm 0.0065$ | $0.9211 \pm 0.0038$ | $0.3271 \pm 0.0012$ |
|  | SGLD | $0.4628 \pm 0.0682$ | $0.0685 \pm 0.0306$ | $0.6979 \pm 0.0108$ | $0.2523 \pm 0.0053$ |
|  | Ensembles | $0.6107 \pm$ - | $0.2982 \pm$ - | $0.9338 \pm$ - | $0.3285 \pm$ - |
| Uni-Mol | Deterministic | $0.5325 \pm 0.0364$ | $0.3534 \pm 0.0074$ | $1.0741 \pm 0.0298$ | $0.3726 \pm 0.0064$ |
|  | Temperature | $0.5326 \pm 0.0366$ | $0.2867 \pm 0.0074$ | $0.9013 \pm 0.0181$ | $0.3297 \pm 0.0056$ |
|  | Focal Loss | $0.5405 \pm 0.0458$ | $0.0522 \pm 0.0042$ | $0.6912 \pm 0.0075$ | $0.2490 \pm 0.0037$ |
|  | MC Dropout | $0.5336 \pm 0.0397$ | $0.3449 \pm 0.0077$ | $1.0457 \pm 0.0290$ | $0.3666 \pm 0.0066$ |
|  | SWAG | $0.5592 \pm 0.0143$ | $0.3008 \pm 0.0042$ | $0.9277 \pm 0.0109$ | $0.3374 \pm 0.0033$ |
|  | BBP | $0.4626 \pm 0.0239$ | $0.2631 \pm 0.0447$ | $0.8784 \pm 0.0809$ | $0.3209 \pm 0.0243$ |
|  | SGLD | $0.4590 \pm 0.0456$ | $0.1206 \pm 0.0298$ | $0.7238 \pm 0.0136$ | $0.2646 \pm 0.0059$ |
|  | Ensembles | $0.5875 \pm$ - | $0.3370 \pm$ - | $1.0186 \pm$ - | $0.3614 \pm$ - |

Table 33: Test results with frozen backbone model weights on ClinTox.

| | | ROC-AUC | ECE | NLL | BS |
|---|---|---|---|---|---|
| ChemBERTa | Deterministic | $0.6344 \pm 0.2837$ | $0.2913 \pm 0.1651$ | $0.4713 \pm 0.2530$ | $0.1618 \pm 0.0952$ |
| | Temperature | $0.6344 \pm 0.2837$ | $0.2389 \pm 0.1655$ | $0.4157 \pm 0.2533$ | $0.1398 \pm 0.0964$ |
| | Focal Loss | $0.6369 \pm 0.2860$ | $0.3267 \pm 0.1183$ | $0.5099 \pm 0.2000$ | $0.1688 \pm 0.0859$ |
| | MC Dropout | $0.6689 \pm 0.2643$ | $0.2886 \pm 0.1675$ | $0.4706 \pm 0.2491$ | $0.1612 \pm 0.0934$ |
| | SWAG | $0.6911 \pm 0.2300$ | $0.2245 \pm 0.1211$ | $0.3728 \pm 0.1831$ | $0.1142 \pm 0.0607$ |
| | BBP | $0.7242 \pm 0.0410$ | $0.2659 \pm 0.1039$ | $0.4515 \pm 0.1399$ | $0.1379 \pm 0.0638$ |
| | SGLD | $0.5489 \pm 0.1696$ | $0.4339 \pm 0.0032$ | $0.6898 \pm 0.0048$ | $0.2483 \pm 0.0024$ |
| | Ensembles | $0.9734 \pm$ - | $0.2885 \pm$ - | $0.4085 \pm$ - | $0.1139 \pm$ - |
| GROVER | Deterministic | $0.6437 \pm 0.0235$ | $0.0304 \pm 0.0051$ | $0.2357 \pm 0.0041$ | $0.0599 \pm 0.0008$ |
| | Temperature | $0.6426 \pm 0.0240$ | $0.0179 \pm 0.0032$ | $0.2337 \pm 0.0050$ | $0.0594 \pm 0.0007$ |
| | Focal Loss | $0.6328 \pm 0.0220$ | $0.2116 \pm 0.0054$ | $0.3897 \pm 0.0064$ | $0.1109 \pm 0.0027$ |
| | MC Dropout | $0.6534 \pm 0.0201$ | $0.0222 \pm 0.0041$ | $0.2329 \pm 0.0029$ | $0.0594 \pm 0.0005$ |
| | SWAG | $0.6838 \pm 0.0124$ | $0.0268 \pm 0.0064$ | $0.2306 \pm 0.0031$ | $0.0589 \pm 0.0007$ |
| | BBP | $0.6019 \pm 0.0346$ | $0.0339 \pm 0.0028$ | $0.2422 \pm 0.0032$ | $0.0609 \pm 0.0005$ |
| | SGLD | $0.4601 \pm 0.0878$ | $0.4365 \pm 0.0355$ | $0.7050 \pm 0.0650$ | $0.2558 \pm 0.0321$ |
| | Ensembles | $0.7039 \pm$ - | $0.0348 \pm$ - | $0.2311 \pm$ - | $0.0591 \pm$ - |
| Uni-Mol | Deterministic | $0.5750 \pm 0.0185$ | $0.1620 \pm 0.0598$ | $0.3423 \pm 0.0569$ | $0.0916 \pm 0.0209$ |
| | Temperature | $0.5752 \pm 0.0182$ | $0.0713 \pm 0.0499$ | $0.2694 \pm 0.0319$ | $0.0678 \pm 0.0086$ |
| | Focal Loss | $0.5696 \pm 0.0199$ | $0.2605 \pm 0.0328$ | $0.4397 \pm 0.0359$ | $0.1290 \pm 0.0158$ |
| | MC Dropout | $0.5797 \pm 0.0259$ | $0.1704 \pm 0.0595$ | $0.3499 \pm 0.0577$ | $0.0944 \pm 0.0216$ |
| | SWAG | $0.5887 \pm 0.0228$ | $0.1125 \pm 0.0324$ | $0.2948 \pm 0.0250$ | $0.0741 \pm 0.0075$ |
| | BBP | $0.5899 \pm 0.0112$ | $0.2754 \pm 0.0510$ | $0.4608 \pm 0.0683$ | $0.1391 \pm 0.0313$ |
| | SGLD | $0.5257 \pm 0.0576$ | $0.4285 \pm 0.0442$ | $0.7081 \pm 0.0984$ | $0.2556 \pm 0.0463$ |
| | Ensembles | $0.5551 \pm$ - | $0.1449 \pm$ - | $0.3199 \pm$ - | $0.0811 \pm$ - |

Table 34: Test results with frozen backbone model weights on Tox21.

| | | ROC-AUC | ECE | NLL | BS |
|---|---|---|---|---|---|
| ChemBERTa | Deterministic | $0.6615 \pm 0.0039$ | $0.0402 \pm 0.0008$ | $0.2934 \pm 0.0004$ | $0.0826 \pm 0.0002$ |
| | Temperature | $0.6615 \pm 0.0039$ | $0.0307 \pm 0.0015$ | $0.2894 \pm 0.0008$ | $0.0817 \pm 0.0001$ |
| | Focal Loss | $0.6646 \pm 0.0026$ | $0.1183 \pm 0.0025$ | $0.3428 \pm 0.0021$ | $0.0949 \pm 0.0007$ |
| | MC Dropout | $0.6729 \pm 0.0048$ | $0.0315 \pm 0.0016$ | $0.2859 \pm 0.0007$ | $0.0807 \pm 0.0002$ |
| | SWAG | $0.6619 \pm 0.0038$ | $0.0397 \pm 0.0012$ | $0.2934 \pm 0.0005$ | $0.0826 \pm 0.0002$ |
| | BBP | $0.5426 \pm 0.0080$ | $0.1150 \pm 0.0079$ | $0.3629 \pm 0.0059$ | $0.1009 \pm 0.0019$ |
| | SGLD | $0.5062 \pm 0.0212$ | $0.3973 \pm 0.0005$ | $0.6903 \pm 0.0008$ | $0.2486 \pm 0.0004$ |
| | Ensembles | $0.6663 \pm$ - | $0.0383 \pm$ - | $0.2916 \pm$ - | $0.0822 \pm$ - |
| GROVER | Deterministic | $0.6897 \pm 0.0102$ | $0.0361 \pm 0.0002$ | $0.2838 \pm 0.0011$ | $0.0807 \pm 0.0003$ |
| | Temperature | $0.6897 \pm 0.0102$ | $0.0223 \pm 0.0017$ | $0.2789 \pm 0.0011$ | $0.0799 \pm 0.0003$ |
| | Focal Loss | $0.6632 \pm 0.0139$ | $0.1326 \pm 0.0023$ | $0.3593 \pm 0.0005$ | $0.0998 \pm 0.0002$ |
| | MC Dropout | $0.6874 \pm 0.0093$ | $0.0343 \pm 0.0004$ | $0.2842 \pm 0.0009$ | $0.0810 \pm 0.0002$ |
| | SWAG | $0.6923 \pm 0.0075$ | $0.0344 \pm 0.0002$ | $0.2830 \pm 0.0008$ | $0.0807 \pm 0.0002$ |
| | BBP | $0.6496 \pm 0.0040$ | $0.0364 \pm 0.0005$ | $0.2951 \pm 0.0003$ | $0.0836 \pm 0.0001$ |
| | SGLD | $0.5072 \pm 0.0266$ | $0.3898 \pm 0.0018$ | $0.6806 \pm 0.0024$ | $0.2438 \pm 0.0012$ |
| | Ensembles | $0.6911 \pm$ - | $0.0365 \pm$ - | $0.2839 \pm$ - | $0.0807 \pm$ - |
| Uni-Mol | Deterministic | $0.5702 \pm 0.0065$ | $0.0556 \pm 0.0008$ | $0.3352 \pm 0.0009$ | $0.0905 \pm 0.0002$ |
| | Temperature | $0.5702 \pm 0.0066$ | $0.0153 \pm 0.0012$ | $0.3036 \pm 0.0001$ | $0.0862 \pm 0.0000$ |
| | Focal Loss | $0.5620 \pm 0.0067$ | $0.0472 \pm 0.0030$ | $0.3183 \pm 0.0020$ | $0.0888 \pm 0.0004$ |
| | MC Dropout | $0.5696 \pm 0.0056$ | $0.0519 \pm 0.0004$ | $0.3297 \pm 0.0007$ | $0.0900 \pm 0.0002$ |
| | SWAG | $0.5936 \pm 0.0056$ | $0.0293 \pm 0.0011$ | $0.3070 \pm 0.0005$ | $0.0868 \pm 0.0001$ |
| | BBP | $0.5505 \pm 0.0184$ | $0.0472 \pm 0.0021$ | $0.3249 \pm 0.0021$ | $0.0896 \pm 0.0002$ |
| | SGLD | $0.4927 \pm 0.0089$ | $0.4022 \pm 0.0118$ | $0.7135 \pm 0.0216$ | $0.2599 \pm 0.0104$ |
| | Ensembles | $0.5866 \pm$ - | $0.0546 \pm$ - | $0.3335 \pm$ - | $0.0901 \pm$ - |

Table 35: Test results with frozen backbone model weights on ESOL.

| | | RMSE | MAE | NLL | CE |
|---|---|---|---|---|---|
| ChemBERTa | Deterministic | $1.9543 \pm 0.0830$ | $1.4348 \pm 0.0895$ | $1.2260 \pm 0.0454$ | $0.0090 \pm 0.0010$ |
| | MC Dropout | $1.8954 \pm 0.1132$ | $1.4051 \pm 0.1047$ | $1.1497 \pm 0.0815$ | $0.0080 \pm 0.0006$ |
| | SWAG | $1.9708 \pm 0.0865$ | $1.4372 \pm 0.0905$ | $1.2561 \pm 0.0495$ | $0.0105 \pm 0.0011$ |
| | BBP | $2.2450 \pm 0.0440$ | $1.7152 \pm 0.0346$ | $1.3560 \pm 0.0285$ | $0.0155 \pm 0.0026$ |
| | SGLD | $2.3491 \pm 0.0542$ | $1.7899 \pm 0.0472$ | $1.4684 \pm 0.0373$ | $0.0180 \pm 0.0055$ |
| | Ensembles | $1.8844 \pm$ - | $1.3699 \pm$ - | $1.1914 \pm$ - | $0.0082 \pm$ - |
| GROVER | Deterministic | $1.8533 \pm 0.1590$ | $1.4171 \pm 0.1281$ | $1.1335 \pm 0.0833$ | $0.0097 \pm 0.0024$ |
| | MC Dropout | $1.8629 \pm 0.1562$ | $1.4131 \pm 0.1234$ | $1.1444 \pm 0.0836$ | $0.0110 \pm 0.0031$ |
| | SWAG | $1.8059 \pm 0.1521$ | $1.3651 \pm 0.1102$ | $1.0930 \pm 0.0857$ | $0.0120 \pm 0.0029$ |
| | BBP | $1.9032 \pm 0.1318$ | $1.4502 \pm 0.1186$ | $1.3243 \pm 0.0225$ | $0.0088 \pm 0.0014$ |
| | SGLD | $2.4699 \pm 0.1371$ | $1.8858 \pm 0.1161$ | $1.3697 \pm 0.0510$ | $0.0285 \pm 0.0050$ |
| | Ensembles | $1.8185 \pm$ - | $1.3161 \pm$ - | $1.1063 \pm$ - | $0.0098 \pm$ - |
| Uni-Mol | Deterministic | $2.2126 \pm 0.0258$ | $1.7134 \pm 0.0482$ | $1.3168 \pm 0.0112$ | $0.0078 \pm 0.0054$ |
| | MC Dropout | $2.2155 \pm 0.0223$ | $1.7118 \pm 0.0426$ | $1.3169 \pm 0.0106$ | $0.0078 \pm 0.0053$ |
| | SWAG | $2.1809 \pm 0.0258$ | $1.6573 \pm 0.0294$ | $1.2883 \pm 0.0210$ | $0.0085 \pm 0.0035$ |
| | BBP | $2.1964 \pm 0.1226$ | $1.7223 \pm 0.0996$ | $1.3008 \pm 0.0625$ | $0.0053 \pm 0.0042$ |
| | SGLD | $2.3596 \pm 0.0599$ | $1.8831 \pm 0.0734$ | $1.4158 \pm 0.0808$ | $0.0139 \pm 0.0037$ |
| | Ensembles | $2.1476 \pm$ - | $1.6524 \pm$ - | $1.2669 \pm$ - | $0.0053 \pm$ - |

Table 36: Test results with frozen backbone model weights on FreeSolv.

|  |  | RMSE | MAE | NLL | CE |
|---|---|---|---|---|---|
| ChemBERTa | Deterministic | 3.7564 ± 0.0520 | 2.7933 ± 0.0770 | 2.0303 ± 0.0828 | 0.0378 ± 0.0078 |
|  | MC Dropout | 3.7861 ± 0.0542 | 2.8493 ± 0.0801 | 2.0393 ± 0.0773 | 0.0429 ± 0.0065 |
|  | SWAG | 3.7629 ± 0.0297 | 2.8017 ± 0.0563 | 2.0520 ± 0.0679 | 0.0395 ± 0.0062 |
|  | BBP | 4.5822 ± 0.0444 | 3.4921 ± 0.0636 | 2.3843 ± 0.0507 | 0.0669 ± 0.0037 |
|  | SGLD | 4.3996 ± 0.1248 | 3.3132 ± 0.1317 | 2.3018 ± 0.1113 | 0.0546 ± 0.0088 |
|  | Ensembles | 3.7094 ± - | 2.7309 ± - | 2.0012 ± - | 0.0370 ± - |
| GROVER | Deterministic | 4.2407 ± 0.3174 | 3.1254 ± 0.4023 | 2.0701 ± 0.1023 | 0.0233 ± 0.0141 |
|  | MC Dropout | 4.2045 ± 0.3199 | 3.1098 ± 0.3737 | 2.0422 ± 0.0929 | 0.0199 ± 0.0130 |
|  | SWAG | 4.1854 ± 0.2899 | 3.1039 ± 0.3888 | 2.0604 ± 0.1230 | 0.0251 ± 0.0127 |
|  | BBP | 4.4898 ± 0.2067 | 3.3533 ± 0.2186 | 2.0452 ± 0.0530 | 0.0378 ± 0.0136 |
|  | SGLD | 4.6731 ± 0.5142 | 3.5858 ± 0.5165 | 2.1826 ± 0.0714 | 0.0529 ± 0.0178 |
|  | Ensembles | 4.0551 ± - | 3.0166 ± - | 1.9352 ± - | 0.0245 ± - |
| Uni-Mol | Deterministic | 3.9034 ± 0.4092 | 2.8791 ± 0.3542 | 1.8779 ± 0.0965 | 0.0207 ± 0.0150 |
|  | MC Dropout | 3.9103 ± 0.4142 | 2.8836 ± 0.3612 | 1.8813 ± 0.0990 | 0.0209 ± 0.0151 |
|  | SWAG | 4.1746 ± 0.2749 | 3.1272 ± 0.2718 | 1.9582 ± 0.0786 | 0.0360 ± 0.0128 |
|  | BBP | 4.0001 ± 0.2701 | 3.0110 ± 0.2507 | 1.9486 ± 0.1219 | 0.0276 ± 0.0196 |
|  | SGLD | 4.1575 ± 0.0222 | 3.1164 ± 0.0627 | 2.0574 ± 0.1045 | 0.0321 ± 0.0066 |
|  | Ensembles | 4.0416 ± - | 2.9823 ± - | 1.9718 ± - | 0.0287 ± - |

Table 37: Test results with frozen backbone model weights on Lipophilicity.

|  |  | RMSE | MAE | NLL | CE |
|---|---|---|---|---|---|
| ChemBERTa | Deterministic | 0.9958 ± 0.0033 | 0.8083 ± 0.0047 | 0.4837 ± 0.0039 | 0.0016 ± 0.0001 |
|  | MC Dropout | 0.9985 ± 0.0064 | 0.8185 ± 0.0061 | 0.4885 ± 0.0063 | 0.0041 ± 0.0001 |
|  | SWAG | 0.9948 ± 0.0029 | 0.8070 ± 0.0043 | 0.4823 ± 0.0031 | 0.0016 ± 0.0001 |
|  | BBP | 1.0477 ± 0.0017 | 0.8578 ± 0.0018 | 0.5537 ± 0.0033 | 0.0025 ± 0.0001 |
|  | SGLD | 1.1189 ± 0.0033 | 0.9266 ± 0.0036 | 0.6225 ± 0.0038 | 0.0074 ± 0.0012 |
|  | Ensembles | 0.9837 ± - | 0.7981 ± - | 0.4729 ± - | 0.0016 ± - |
| GROVER | Deterministic | 0.9297 ± 0.0182 | 0.7568 ± 0.0165 | 0.4238 ± 0.0178 | 0.0019 ± 0.0002 |
|  | MC Dropout | 0.9366 ± 0.0146 | 0.7683 ± 0.0113 | 0.4348 ± 0.0148 | 0.0033 ± 0.0004 |
|  | SWAG | 0.9290 ± 0.0148 | 0.7621 ± 0.0104 | 0.4223 ± 0.0144 | 0.0034 ± 0.0003 |
|  | BBP | 0.9753 ± 0.0229 | 0.7950 ± 0.0192 | 0.6023 ± 0.0346 | 0.0025 ± 0.0007 |
|  | SGLD | 1.1203 ± 0.0238 | 0.9301 ± 0.0243 | 0.6134 ± 0.0206 | 0.0052 ± 0.0031 |
|  | Ensembles | 0.9160 ± - | 0.7455 ± - | 0.4134 ± - | 0.0019 ± - |
| Uni-Mol | Deterministic | 1.0917 ± 0.0210 | 0.9069 ± 0.0203 | 0.6446 ± 0.0189 | 0.0055 ± 0.0014 |
|  | MC Dropout | 1.0929 ± 0.0216 | 0.9080 ± 0.0208 | 0.6442 ± 0.0192 | 0.0054 ± 0.0014 |
|  | SWAG | 1.0822 ± 0.0114 | 0.8986 ± 0.0115 | 0.6144 ± 0.0121 | 0.0045 ± 0.0004 |
|  | BBP | 1.1003 ± 0.0198 | 0.9112 ± 0.0160 | 0.8454 ± 0.0580 | 0.0124 ± 0.0022 |
|  | SGLD | 1.1796 ± 0.0644 | 0.9567 ± 0.0190 | 0.6812 ± 0.0469 | 0.0135 ± 0.0051 |
|  | Ensembles | 1.0863 ± - | 0.9019 ± - | 0.6421 ± - | 0.0052 ± - |

Table 38: Test results with frozen backbone model weights on QM7.

|  |  | RMSE | MAE | NLL | CE |
|---|---|---|---|---|---|
| ChemBERTa | Deterministic | 137.8736 ± 2.2380 | 98.5749 ± 2.9618 | 5.4092 ± 0.0182 | 0.0049 ± 0.0001 |
|  | MC Dropout | 133.8345 ± 1.8327 | 97.7798 ± 2.3986 | 5.4049 ± 0.0117 | 0.0028 ± 0.0001 |
|  | SWAG | 138.2471 ± 2.2020 | 99.0168 ± 2.7806 | 5.4107 ± 0.0177 | 0.0048 ± 0.0001 |
|  | BBP | 141.0033 ± 1.2990 | 104.8384 ± 1.5366 | 5.4635 ± 0.0100 | 0.0026 ± 0.0005 |
|  | SGLD | 202.5163 ± 1.4235 | 165.5756 ± 1.1589 | 5.8131 ± 0.0087 | 0.0208 ± 0.0015 |
|  | Ensembles | 136.6747 ± - | 97.2405 ± - | 5.4066 ± - | 0.0045 ± - |
| GROVER | Deterministic | 134.4406 ± 3.7901 | 95.5997 ± 4.3313 | 5.4077 ± 0.0717 | 0.0058 ± 0.0023 |
|  | MC Dropout | 131.7638 ± 3.5591 | 93.2436 ± 4.3578 | 5.4005 ± 0.0761 | 0.0042 ± 0.0021 |
|  | SWAG | 130.2656 ± 2.2246 | 92.1086 ± 3.1537 | 5.3670 ± 0.0446 | 0.0047 ± 0.0015 |
|  | BBP | 137.9643 ± 6.4603 | 100.5829 ± 7.7898 | 5.8332 ± 0.1322 | 0.0055 ± 0.0030 |
|  | SGLD | 191.3884 ± 3.4342 | 154.4454 ± 1.8662 | 5.8488 ± 0.0227 | 0.0158 ± 0.0033 |
|  | Ensembles | 130.9773 ± - | 93.3082 ± - | 5.4027 ± - | 0.0051 ± - |
| Uni-Mol | Deterministic | 183.5096 ± 9.5881 | 148.3822 ± 9.8276 | 5.7908 ± 0.0353 | 0.0180 ± 0.0051 |
|  | MC Dropout | 183.7110 ± 9.1987 | 148.5660 ± 9.3875 | 5.7908 ± 0.0336 | 0.0178 ± 0.0048 |
|  | SWAG | 171.6691 ± 1.7586 | 137.1572 ± 2.1770 | 5.7147 ± 0.0078 | 0.0111 ± 0.0008 |
|  | BBP | 178.0057 ± 3.9910 | 142.9159 ± 3.6271 | 6.0873 ± 0.0082 | 0.0193 ± 0.0007 |
|  | SGLD | 226.7307 ± 21.2619 | 188.4545 ± 19.4483 | 6.0090 ± 0.1956 | 0.0458 ± 0.0322 |
|  | Ensembles | 175.6095 ± - | 140.6629 ± - | 5.7658 ± - | 0.0129 ± - |

