# OpenReview forum: "MUBen: Benchmarking the Uncertainty of Molecular Representation Models"
_NeurIPS.cc/2023/Workshop/AI4Science — NeurIPS2023-AI4Science Poster_

### Official Review · Reviewer_yNnc · 2023-10-11
**Review of MUBen**

**Rating:** 7
**Confidence:** 3

**Review:**

Summary:
This paper provides a comprehensive benchmark of uncertainty measures for a variety of SOTA molecular representation learning models on a variety of tasks. Their benchmarking is motivated by the necessity of uncertainty measures for better model robustness and estimation of data distributions.

Clarity:
This is a well written paper, with a clear experimental set up. They clearly define the uncertainty quantification methods used for each model and communicate the results of each.

Quality:
The experiments are very comprehensive, and provide a good quality assessment of each  uncertainty quantification method for each model. They also select a variety of backbone models ranging from language models to graph neural networks.

Strengths:
Provides a good quality assessment of a wide variety of uncertainty quantification methods for a good handful of SOTA molecule representation models.

Weaknesses:
It is difficult to generalize which uncertainty quantification method to use broadly in molecular representation learning from their paper. While they show that ensemble methods is best, it is unlikely to be adopted by the field due to high computation costs.

---

### Official Review · Reviewer_A6Dc · 2023-10-25
**benchmark for UQ of pre-trained molecular representation models**

**Rating:** 8
**Confidence:** 4

**Review:**

The paper presents a comprehensive study evaluating UQ methods for molecular pretraining models. The paper is well-written, and the topic is emerging. Indeed, there is a noticeable lack of benchmarks for measuring UQ in molecular pretraining models. The paper analyzes current SOTA methods, evaluating them through different backbone models and UQ method categories, thereby providing a clear view of the results. The authors have also provided a GitHub repo for convenient usage. One limitation is that the study compares only significant studies; extending the research to include more pretraining models would be beneficial.

---

### Meta-Review · Area_Chair_3LeD · 2023-10-26

**Recommendation:** Accept (Oral)
**Confidence:** 4

**Metareview:**

This paper presents a comprehensive benchmark for uncertainty quantification for various molecular representation models across various tasks. The reviewers agree this is well motivated, clearly written, with results that are of broad interest to the community. Recommendation: Oral.